# PrivacyRestore: Privacy-Preserving Inference in Large Language Models via Privacy Removal and Restoration

## Abstract

The widespread usage of online Large Language Models (LLMs) inference services has raised significant privacy concerns about the potential exposure of private information in user inputs to malicious eavesdroppers. Existing privacy protection methods for LLMs suffer from either insufficient privacy protection, performance degradation, or large inference time overhead. To address these limitations, we propose PrivacyRestore, a plug-and-play method to protect the privacy of user inputs during LLM inference. The server first trains restoration vectors for each privacy span and then release to clients. Privacy span is defined as a contiguous sequence of tokens within a text that contain private information. The client then aggregate restoration vectors of all privacy spans in the input into a single meta restoration vector which is later sent to the server side along with the input without privacy spans. The private information is restored via activation steering during inference. Furthermore, we prove that PrivacyRestore inherently prevents the linear growth of the privacy budget. We create three datasets, covering medical and legal domains, to evaluate the effectiveness of privacy preserving methods. The experimental results show that PrivacyRestore effectively protects private information and maintain acceptable levels of performance and inference overhead.

## 1 Introduction

Large Language Models (LLMs) have emerged as powerful tools in various domains, including healthcare (Chen et al., 2023; Xu et al., 2023), law (Wu et al.; Deng et al., 2023a), and finance (Wu et al., 2023; Xie et al., 2023). With the exception of a very small portion of users who have the resources and expertise to deploy LLMs locally, the vast majority of users access and interact with these powerful models through online inference services.

However, the widespread usage of online LLMs inference services has raised significant privacy concerns, especially regarding the potential risk of private information being leaked through user inputs when interacting with LLMs deployed on cloud platforms. User inputs often contain sensitive information such as details in medical records and legal cases. Potential threats may arise from eavesdropper attackers intercepting user queries during transmission to cloud platforms for malicious purposes. For example, in sensitive domains like medical diagnosis, if a user's input containing personal health information, such as "*I was previously diagnosed with HIV, and lately I've been experiencing fever and diarrhea...*" is disclosed, it may cause troubles to their life.

In this paper, we focus on protecting the private information contained in user inputs during LLM inference stage. In this setting, the client submits inputs to the server (also known as the service provider) and there is a risk that inputs might be disclosed by attackers. Current methods for protecting user inputs can be categorized into two categories: Secure Multi-Party Computation (SMPC) and Differential Privacy (DP). SMPC based methods (Hao et al., 2022b; Li et al., 2023a; Liang et al., 2024) utilize encryption protocols and algorithms to enable collaborative computation without revealing original data to others. However, SMPC methods have large inference time overhead, making them impractical for real-time applications. For example, running a single pass inference on the RoBERTa-Base (Liu et al., 2019) requires 168.43 seconds (Hao et al., 2022a). DP based methods (Feyisetan et al., 2020; 2019; Xu et al., 2020; Bo et al., 2021) introduce the definition of

$d_\chi$-privacy (Chatzikokolakis et al., 2013; Alvim et al., 2018) and apply a word-level text-to-text privatization on data locally before transmitting data to the server. Nevertheless, DP based methods inevitably degrade the performance of downstream tasks due to noise injection, which is known as the privacy-utility trade-off. Additionally, as text length increases, the performance degradation becomes pronounced, This phenomenon is known as the linear growth of the privacy budget Mattern et al. (2022b) in differential privacy. Hence, there is a need to develop privacy-preserving methods which can effectively safeguard the privacy of user inputs while maintaining high-quality outputs, without incurring prohibitive computational costs.

We propose PrivacyRestore which directly removes privacy spans in user inputs and restores private information via activation steering (Li et al., 2023c; Turner et al., 2023; Hernandez et al., 2023) during model inference. Our method is based on two key assumptions: **(a) Private information is confined within specific a contiguous sequence of tokens, termed "privacy span", rather than being dispersed throughout the entire input.** Privacy span is defined as a contiguous sequence of tokens within a text that contain private information. The removal or proper redaction of privacy spans significantly impedes unauthorized parties from reconstructing or inferring the underlying private information. For instance, if privacy spans "*HIV*" , "*fever*" and "*diarrhea*" are removed from a medical record "*I was previously diagnosed with HIV, and lately I've been experiencing fever and diarrhea...*", attackers can not recover any private information. **(b) In a particular domain, the number of potential privacy spans is limited and finite.** For example, in the application of medical diagnosis, privacy spans generally pertain to symptoms and disease names, and the number of possible symptoms and disease names is inherently limited. Moreover, although medical knowledge and terminology inevitably evolve, the core set of symptoms and diseases remains relatively stable and finite.

PrivacyRestore operates in two stages: the preparation stage and the inference stage. **In the preparation stage**, we first identifies the attention heads where the activation steering occurs. Second, each privacy span is encoded to a vector named restoration vector. This stage is conducted on the server side. Our method is plug-and-play, requiring only the restoration vectors to be trainable, while keeping the LLM frozen. Once training is complete, the users keep all restoration vectors on the client side. **In the inference stage**, the user construct a meta vector by first estimating the importance of each privacy span in the input and then calculating a weighted sum of the corresponding restoration vectors. The user then removes the privacy spans from the input and submits the remaining input along with the meta vector to the server. The server uses the meta restoration vector to restore the removed privacy spans through activation steering.

To prevent the leakage of privacy spans via reverse-engineering on the meta vector, the $d_\chi$-privacy mechanism (Feyisetan et al., 2020) is applied on the meta vector before transmission on the client side. $d_\chi$-privacy mechanism is a variant of the differential privacy mechanism (Dwork et al., 2016). By applying $d_\chi$-privacy to the meta restoration vector instead of words, our method inherently addresses the issue of linear growth of privacy budget (Mattern et al., 2022a) commonly encountered in $d_\chi$-privacy and other DP variants. Experimental results demonstrate that the proposed method effectively protects private information and maintains satisfactory performance and inference efficiency.

The contributions of our paper are summarized as follows,

- We propose a plug-and-play privacy protection method that removes privacy spans in the input and restores private information via activation steering during inference.

- We propose Attention-aware Weighted Aggregation to construct the meta vector and apply the $d_\chi$-privacy mechanism to the meta vector, inherently addressing the problem of the linear growth of privacy budget.

- We construct three datasets, covering the medical and legal fields, to evaluate our method. Experimental results demonstrate its capabilities of privacy protection. It also maintains acceptable performance and inference efficiency.

## 2 RELATED WORKS

In this section, we introduce the related works on user input protection methods, which are currently divided into two categories: SMPC-based methods and DP-based methods.

## 2.1 Secure Multi-Party Computation (SMPC)

Secure multi-party computation (SMPC) methods utilize multi-party encryption algorithms to enable collaborative computation among multiple parties while protecting the privacy of their data. However, most nonlinear operations in LLMs cannot directly support secure multi-party computation. To address this challenge, current SMPC methods focus on two optimization directions: model structure-oriented optimization and protocol-oriented optimization.

The model structure-oriented approach aims to replace SMPC-unfriendly nonlinear operations with SMPC-friendly alternatives. For instance, MPC-Former (Li et al., 2023a) approximates nonlinear operations in Transformer using polynomials and maintains performance through model distillation. MERGER (Liang et al., 2024) integrates previous techniques to natural language generation (NLG) tasks by bypassing embedded computation and reorganizing linear operations in Transformer modules, further enhancing computational efficiency and model performance. In contrast, the protocol-oriented approach focuses on designing efficient SMPC operators for nonlinear operations in LLMs while preserving the original model structure. Recent works Hao et al. (2022b); Liu & Liu (2023); Zheng et al. (2023b); Gupta et al. (2023) have improved the efficiency of nonlinear operations in privacy-preserving LLMs inference by utilizing various SMPC protocols, such as confusion circuit and function secret sharing.

Although SMPC-based methods can be applied to protect user inputs during model inference, they still suffer from large inference time overhead. For example, inference on the RoBERTa-Base model takes 168.43 seconds (Hao et al., 2022a), making current SMPC methods impractical for online LLM inference services.

## 2.2 Differential Privacy (DP)

Differential Privacy (DP), as introduced by Dwork et al. (2016), is designed to protect individual privacy by preventing attackers from identifying specific participants in a dataset. Several variants of DP have been developed to enhance privacy protection across various settings, adapting the core principles of DP to different types of data and threat models. Notable examples include Centralized Differential Privacy (CDP), Local Differential Privacy (LDP) and $d_\chi$-privacy.

CDP (Dwork et al., 2016) operates under the assumption that all data has been stored in a central repository. It guarantees that attackers cannot distinguish between any two adjacent repositories based on query results. In contrast, LDP (Duchi et al., 2013) provides a stronger guarantee, ensuring that attackers cannot distinguish between any two adjacent inputs. Mattern et al. (2022b) and Utpala et al. (2023) propose using paraphrasing techniques to achieve LDP on user inputs. The formal definitions of CDP and LDP are provided in Appendix D.

LDP allocates the same privacy budget $\epsilon$ to all adjacent input pairs, regardless of their similarity. Applying the same $\epsilon$ forces each user input to be indistinguishable from any other, which can negatively impact data utility. However, it is sufficient for privacy protection to make each user input indistinguishable only from its closer counterparts. To address this, $d_\chi$-privacy (Feyisetan et al., 2019), a relaxed version of LDP, incorporates metrics that measure the similarity between inputs, allowing for more flexible control over the privacy budget. $d_\chi$-privacy is defined as,

**Definition 2.1.** ($d_\chi$-privacy). *A randomized mechanism $\mathcal{M} : \mathcal{I} \to \mathcal{O}$ fulfills $\epsilon$-$d_\chi$-privacy if for all adjacent inputs $I, I' \in \mathcal{I}$ and all possible outputs $O \subset \mathcal{O}$,*

$$\mathbb{P}\left(\mathcal{M}(I) \in O\right) \leq \exp(\epsilon d_\chi(I, I'))\mathbb{P}\left(\mathcal{M}(I') \in O\right),$$

where $d_\chi$ is a distance function defined on $\mathcal{I}$. Recent works (Feyisetan et al., 2020; Xu et al., 2020; Li et al., 2023d; Qu et al., 2021) leverage $d_\chi$-privacy to safeguard user inputs during both inference and fine-tuning phases. However, as noted by Mattern et al. (2022b), all of the aforementioned DP variants suffer from the linear growth of the privacy budget. A larger privacy budget indicates weaker privacy protection, meaning that as the input length increases, the effectiveness of privacy protection diminishes.

## 3 THREAT MODEL

We consider a threat model involving two parties: a server that holds the LLM weights and a client holds user inputs containing privacy spans. **Privacy span is defined as a contiguous sequence of tokens within a text that contain private information.** The server provides services through an API, enabling the client to transmit inputs and receive responses while maintaining the confidentiality of the LLM weights. The server may be vulnerable to attacks by adversaries seeking to steal privacy information in user inputs. Our task is to protect these private spans in user inputs from being intercepted to the adversaries, even when the adversaries can attack the server directly.

## 4 METHODOLOGY

To protect privacy, PrivacyRestore transmit the input with privacy spans removed instead of the entire input text to the server. The information in the privacy spans is encrypted as a vector, which is then injected with noise and is also sent to the server. We propose to use activation steering (Li et al., 2023c; Turner et al., 2023; Hernandez et al., 2023) to restore privacy information. Activation steering methods modify the activations of a language model at inference time to predictably alter its behavior. Activation steering is widely used to truthfulness enhancement(Li et al., 2023c), LLMs detoxifying (Li et al., 2024), and sentiment modification (Turner et al., 2023). To our best knowledge, it is first attempt to use activation steering for privacy information restoration. We include the preliminaries of our methodology in Appendix C.

PrivacyRestore operates in two stages, i.e., the preparation stage and the inference stage.

(1) Preparation stage: This stage takes place on the server. We first identify the edited attention heads and train the restoration vectors for each privacy spans. After training, these vectors are released to the clients. The preparation stage is conducted offline, prior to the server beginning to offer its services.

(2) Inference stage: This stage involves collaboration between the client and server. The client constructs a meta vector which is later transmitted to the server along with the input where the privacy spans are removed. The server then performs inference on the input with privacy spans removed and restores those privacy spans via modifying activations.

An overview of PrivacyRestore is shown in Figure 1. Detailed descriptions of the preparation stage and the inference stage are provided in §4.1 and §4.2 respectively. The definitions of all notations used in this paper can be found in Appendix A.

### 4.1 PREPARETION STAGE

**Edited Heads Identification.**  As pointed by activation steering methods (Li et al., 2023c; Chen et al., 2024), modifying all attention heads in LLMs will degrade overall performance. Inspired by this, we aim to identify the attention heads most relevant to privacy spans.

As shown in upper part of Figure 1, we firstly utilize the probe technique (Alain & Bengio, 2016; Tenney et al., 2019; Belinkov, 2022) to identify the most relevant attention heads for each privacy span. $I_{all} = \{I_1, ..., I_m\}$ represents the user inputs in the training set where $m$ is the size of training set. Given a privacy span $s$, $Y_s = \{y_1, ..., y_m\}$ represents the corresponding labels, where $y_i = 1$ only if input $I_i$ contains privacy span $s$. For each user input $I_i$, we record the hidden state of last token on each attention head . We then train a binary classifier for each head, tailored to the privacy span $s$, as the probe. The probe takes the hidden state of last token as input and predicts whether the input contain the privacy span $s$. The probe is formulated as:

$$\mathcal{F}_h^s(\mathbf{u}_h) = \sigma(\theta_h^s \cdot \mathbf{u}_h), \tag{1}$$

where $\mathcal{F}_h^s(\cdot)$ is the probe of privacy token $s$ on head $h$, $\mathbf{u}_h$ is the hidden state of last token on head $h$, $\theta_h^s$ is parameters of the probe, and $\sigma(\cdot)$ indicates the sigmoid function. A probe $\mathcal{F}_h^s(\cdot)$ with higher accuracy indicates a stronger correlation between the head $h$ and the privacy span $s$. Therefore, we select the top $K$ attention heads with highest accuracies for each privacy span.

Subsequently, we introduce a **Top-K Heads Selector** to identify the common top-K heads set $\mathcal{H}_c$ from the individual top-K heads sets of each privacy span. Using different top-K head sets for

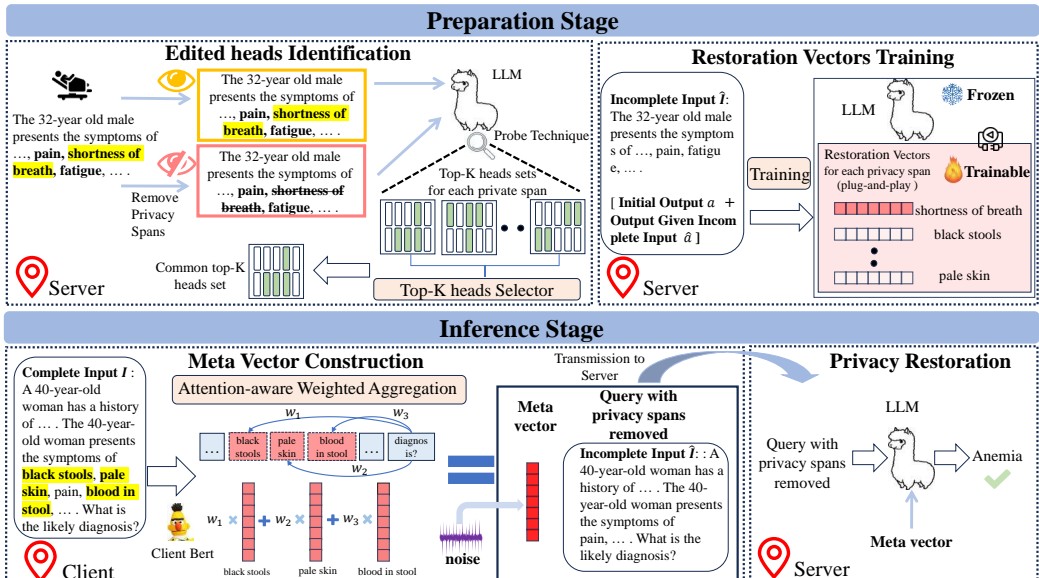

Figure 1: The PrivacyRestore consists of two stages. (1) **Preparation Stage.** This stage is operated on the server side only and is conducted offline before the server starts offering its services. This stage aims to identify the edited heads and train the restoration vectors. (2) **Inference Stage.** This stage involves the collaboration between the server and the client. The client need to construct a meta vector by computing a weighted sum of restoration vectors for all privacy spans in the input. A local lightweight model is used to estimate the weight of each privacy span. Then the client transmits the meta vector and the incomplete input with privacy spans removed to the server. Using the meta vector, the server restores the privacy information.

different privacy spans may suffer the risk of privacy leakage, as an attacker could infer the presence of a specific privacy span based on the characteristics of top-K heads set. Hence, we propose Top-K Heads Selector to combine all different top-K heads sets to construct a common top-K heads set $\mathcal{H}_c$ as the edited heads set. To achieve this, we calculate the average score of each head across all privacy spans, selecting the highest $K$ heads to construct the common set. A head receives a positive score if it appears in the top-K head set of a privacy span $s$. The score is related to the accuracy of probe associated with the head. Specifically, if the probe associated with this head yields higher accuracy, the score is higher. By iterating this process across all privacy spans, we can calculate the average score for each head. The detailed algorithm is described in Appendix G.

**Restoration Vectors Training.** After identifying the edited heads set, the next step is to train the restoration vectors for each privacy span on the server side. The training objective is to align the predictions given the input with privacy spans removed to be the same as the predictions given an intact input.

For each privacy span $s \in \mathcal{S}$, there is a trainable restoration vector $r_s^h$ for each head $h$ in the common top-K heads set $\mathcal{H}_c$. Restoration vectors of all privacy spans on all heads of $\mathcal{H}_c$ form the only trainable parameters $\Theta$ in our method. The LLM weights remain frozen. Our method is plug-and-play and parameter-efficient for training. We fine-tune these restoration vectors using ORPO loss proposed by Hong et al. (2024):

$$\text{ratio}(a|\hat{I};\Theta) = \frac{\mathbb{P}(a|\hat{I};\Theta)}{1 - \mathbb{P}(a|\hat{I};\Theta)}, \tag{2}$$

$$\mathcal{L}_{\text{ORPO}} = \sum_{\hat{I} \in \hat{I}_{all}} -\log \mathbb{P}(a|\hat{I};\Theta) - \lambda \log \sigma \left( \log \frac{\text{ratio}(a|\hat{I};\Theta)}{\text{ratio}(\hat{a}|\hat{I};\Theta)} \right), \tag{3}$$

where $\hat{I}$ denotes the input with privacy spans removed and $\hat{I}_{all} = \{\hat{I}_1, \cdots, \hat{I}_m\}$ represents the training set of incomplete inputs, $a$ is the initial output give the complete input, $\hat{a}$ is the output given

the incomplete input with privacy spans removed and $\lambda$ represents the coefficient. The ORPO loss encourages the model to generate the initial output $a$ rather than the output $\hat{a}$ given the incomplete input. After restoration vectors training, the server will release all restoration vectors to clients.

## 4.2 INFERENCE STAGE

**Meta Vector Construction.** In the inference stage, the client construct a meta vector which is later transmitted to the server along with the input with privacy spans removed, as shown in the lower left panel in Figure 1. These operations are conducted on the client side. Transmitting a single meta vector instead of multiple restoration vectors reduces the communication burden and the risk of data leakage. For instance, adversaries could easily know the number of privacy spans in the user input if multiple restoration vectors were sent.

However, equal weighted aggregation may weaken the influence of critical spans and amplify the effect of irrelevant ones. Therefore, we propose a novel method called **Attention-aware Weighted Aggregation (AWA)** which estimates a weight for each privacy span, and then take the weighted sum of restoration vectors as the aggregation result. This result is then added with noise for privacy protection and transmitted to the server. Considering the limitation of computing resource, we propose to utilize a lightweight model (e.g., BERT Devlin et al. (2019)) to estimate importance weights on the client side. For the privacy span $s$ in the user input $I$, the importance weight $w_s$ is calculated as the average attention received by $s$:

$$w_s \quad = \quad \frac{1}{n}\frac{1}{n_h}\sum_{t=1}^{n}\sum_{h=1}^{n_h}\text{Attn}_h(s, i_t), \tag{4}$$

where $n$ is the number of tokens in the input, $n_h$ is the number of attention heads in the lightweight model, $i_t$ is the $t$-th token of $I$, and $\text{Attn}_h(s, i_t)$ denotes the attention score of $i_t$ attending to the privacy span $s$. To simplify the problem, we first consider constructing the meta vector for a single head $h$. The meta vector $\mathcal{R}_h$ on head $h$ is obtained by computing the weighted sum of the restoration vector $r_s^h$ of each privacy span $s$ on head $h$, normalizing the summation, and adding noise $\mathcal{N}$. The process is formulated as follows,

$$Z_h \quad = \quad \frac{\sum_{s\in\mathcal{S}_I} w_s \cdot r_s^h}{||\sum_{s\in\mathcal{S}_I} w_s \cdot r_s^h||_2}, \tag{5}$$

$$\mathcal{R}_h \quad = \quad Z_h + \mathcal{N}, \tag{6}$$

where $\mathcal{S}_I$ denotes the set of privacy spans in the input $I$, $|\mathcal{S}_I|$ denotes the number of privacy spans and $Z_h$ represents the normalization of the weighted sum on head $h$. The injected noise $\mathcal{N}$ is sampling from the distribution $p(\mathcal{N}) \propto \exp(-\epsilon\|\mathcal{N}\|)$, according to Feyisetan et al. (2020), where $\epsilon$ is the privacy hyperparameter.

To construct the meta vector for multiple heads, we first concatenate the restoration vectors on multiple heads. Then, we apply the weighted summary, normalization, and noise addition to the concatenated vector, as we do for a single edited head. After construction, the meta vector, along with the input with the privacy spans removed, are transmitted to the server for inference.

**Privacy Restoration.** We utilizes the meta vector to restore the missing privacy spans during inference on the input with the privacy spans removed, as illustrated in the lower right part of Figure 1. This operation is conducted on the server side.

Following activation steering methods (Li et al., 2023c; Chen et al., 2024), we apply the meta vector to the outputs of the edited attention heads to restore the privacy spans. Let $\mathbf{u}_h$ represent the hidden state of the last token on head $h$ given the input with privacy spans removed, and $\mathcal{R}_h$ be the meta vector for head $h$, the hidden state of the last token on head $h$ after restoration $\bar{\mathbf{u}}_h$ is denoted as:

$$\bar{\mathbf{u}}_h = \mathbf{u}_h + ||\mathbf{u}_h||_2 \cdot \mathcal{R}_h, \ \forall h \in \mathcal{H}_c. \tag{7}$$

During inference, if a head belongs to the common top-K heads set $\mathcal{H}_c$, its hidden state should be modified using Eq 7.

## 5 ANALYSIS OF PRIVACY BUDGET

In this section, we analyze the privacy budget of DP variants and PrivacyRestore.

**Theorem 5.1.** *The DP variants including CDP, LDP and $d_\chi$-privacy are constrained by a privacy budget that grows linearly with the length of the protected text.*

The detail proof of Theorem 5.1 is presented in Appendix E. As the length of the protected text increases, the growing privacy budget makes these DP variants more vulnerable to adversarial attacks, thus compromising their robustness. We also provide empirical evidence demonstrating the linear growth problem of $d_\chi$-privacy in Section 6.3. We implement two types of attack, i.e., prompt injection attack (Perez & Ribeiro, 2022; Suo, 2024) and attribute inference attack (Li et al., 2022), across three privacy-preserving datasets. As shown in Figure 3(a) and 3(b), attack performance increases with the length of the protected text, highlighting the linear growth problem of the privacy budget in $d_\chi$-privacy.

**Theorem 5.2.** *PrivacyRestore fulfills $d_\chi$-privacy and provides a privacy budget of $\epsilon||Z' - Z||$, where $\epsilon$ denotes privacy hyperparameter, and $Z'$ and $Z$ represent any pair of normalized weighted sums of restoration vectors concatenated across all edited heads. The privacy budget of PrivacyRestore is independent of the length of protected text.*

The detail proof of Theorem 5.2 is provided in Appendix F. The $\epsilon$ is privacy hyperparameter, independent of the length of protected text. The $||Z' - Z||$ represents the distance between two vectors, also independent of the length of protected text. Since the privacy budget in PrivacyRestore is independent of the length of the protected text, our method effectively protects privacy even with longer protected text, inherently addressing the linear growth issue in DP variants. We also provide empirical evidence to support the theorem. We present the attack performance of PrivacyRestore across varying protected text lengths, as detailed in Section 6.3.

## 6 EXPERIMENTS

### 6.1 EXPERIMENTS SETUP

**Datasets.** We evaluate our method in medical and legal domains. However, existing benchmarks, such as DDXPlus (Tchango et al., 2022) and NLICE (Al-Ars et al., 2023) for medical diagnosis, and SLJA (Deng et al., 2023b) for legal judgement, do not specify privacy spans in the input. To address this gap, we leveraged GPT-3.5 (Ouyang et al., 2022) to classify symptoms in DDXPlus/NLICE and case details in SLJA into sensitive and non-sensitive categories, treating the sensitive data as privacy spans. The classification prompt is shown in Appendix O.1. Based on classification results, we curated three privacy-preserving datasets: **Pri-DDXPlus**, **Pri-NLICE** and **Pri-SLJA**, with 149, 64 and 142 types of privacy spans, respectively. The process of dataset construction and statistical information can be found in Appendix B.

**Metrics.** The evaluation access both performance and inference efficiency. For performance evaluation, we employ **MC1** and **MC2** (Lin et al., 2021) to measure the model's accuracy in selecting the correct answer among 4 options. Each sample in Pri-DDXPlus, Pri-NLICE, and Pri-SLJA is assigned with 4 options, including one correct and three incorrect options. The detailed calculation of MC1 and MC2 is outlined in Appendix J. We also evaluate the model's generation ability using **ROUGE-L** and **LLM-Judge (LLM-J)**(Zheng et al., 2023a). For ROUGE-L, the reference text is the initial output produced by the backbone LLM. As ROUGE-L primarily focuses on n-gram overlap between generated text and reference texts, which may not fully capture the semantic meaning or overall quality of the generated content, we further use a LLM (i.e., GPT-3.5) to assess the quality of outputs considering relevance, clarity, and accuracy. The assessment prompt is shown in Appendix O.3. The LLM-J score ranges from 1 to 10, with higher scores indicating better quality. For inference efficiency, we use **Throughput (TP)**, defined as the number of tokens generated per second, to evaluate the inference efficiency.

**Compared Methods.** To demonstrate the effectiveness of our method, we compare our model with following baselines. $d_\chi$-**privacy.** The client applies $d_\chi$-privacy (Feyisetan et al., 2020) mechanism on the entire input, which injects noise into tokens' embedding and find the nearest tokens to replace the initial tokens. $d_\chi$-**privacy on privacy spans.** The client employs $d_\chi$-privacy (Feyisetan et al., 2020) mechanism on privacy spans in the input, rather than the entire input. **Paraphrase.** According to

| Datasets | Methods | MC1 ↑ | MC2 ↑ | ROUGE-L ↑ | LLM-J ↑ | TP ↑ |
|---|---|---|---|---|---|---|
| Pri-DDXPlus | $d_\chi$-privacy | 28.79±0.02 | 30.26±0.01 | 17.97±0.00 | 1.17±0.00 | **37.45±0.01** |
| | $d_\chi$-privacy on privacy spans | 44.71±0.29 | 42.36±0.00 | **29.17±0.04** | 3.31±0.00 | 33.21±0.00 |
| | Paraphrase | 27.92±0.56 | 28.56±0.07 | 18.04±0.01 | 1.23±0.00 | 35.42±0.67 |
| | PrivacyRestore | **62.97±0.00** | **60.19±0.00** | 27.24±0.26 | **4.47±0.00** | 26.09±0.08 |
| Pri-NLICE | $d_\chi$-privacy | 29.08±0.00 | 29.72±0.00 | 15.68±0.02 | 1.41±0.00 | **38.30±0.00** |
| | $d_\chi$-privacy on privacy spans | 30.00±0.09 | 31.46±0.00 | 22.97±0.00 | 3.01±0.00 | 35.73±0.57 |
| | Paraphrase | 28.46±0.02 | 29.15±0.03 | 16.15±0.01 | 1.62±0.00 | 37.22±0.07 |
| | PrivacyRestore | **62.23±1.70** | **57.94±0.09** | **24.42±0.81** | 3.67±0.01 | 32.33±0.01 |
| Pri-SLJA | $d_\chi$-privacy | 16.66±0.37 | 17.57±0.04 | 23.35±0.00 | 2.08±0.00 | **36.83±0.03** |
| | $d_\chi$-privacy on privacy spans | 24.23±1.69 | 26.63±0.67 | **40.10±0.00** | 4.54±0.00 | 36.16±0.00 |
| | Paraphrase | 16.21±0.02 | 17.52±0.02 | 24.90±0.01 | 2.07±0.01 | 31.31±0.05 |
| | PrivacyRestore | **35.47±1.48** | **35.41±0.64** | 37.56±0.06 | **5.25±0.00** | 30.73±0.04 |

Table 1: Comparison of the performance and the inference efficiency between PrivacyRestore and other baselines across three privacy-preserving datasets. All experiments are conducted over 3 runs, with the average results and variances reported. The best results are highlighted in **bold**.

Mattern et al. (2022b); Utpala et al. (2023), clients can use generative models to paraphrase original inputs, achieving effects similar to DP.

**Implementation Details.** We use Llama2-chat-7b (Touvron et al., 2023) as the LLM backbone on the server side, and BERT-base (Devlin et al., 2019) on the client side for weight estimation, as described in Section 4.2. For fair comparison, we utilize flan-t5-base model (Chung et al., 2024) on the client side for paraphrasing in the Paraphrase baseline as its model size is comparable to that of BERT-base. During restoration vector training, the LLM parameters remain fixed, and we train the restoration vectors for 5 epochs with a batch size of 1. The optimal number of edited heads $K$ is 175 for Pri-DDXPlus/Pri-SLJA and 125 for Pri-NLICE. The search process is shown in Section I. To evaluate the generation capabilities, we utilize GPT-3.5 to assess the generated outputs. The prompts are detailed in Appendix O.2. To evaluate inference efficiency, we use the greedy search decoding strategy and set the max generation length to 256 during generation.

**Settings of Privacy Hyperparameters.** The hyperparameters related to privacy protection strength are $\epsilon$ for $d_\chi$-privacy (on privacy spans) and PrivacyRestore, and $\tau$ for paraphrase. For fair comparison, we ensure all methods under the same privacy budget. We show the calculation process of determining values of $\epsilon$ and $\tau$ for different methods on different datasets in Appendix H. The values of $\epsilon$ and $\tau$ for different privacy-preserving methods are shown in Table 5 in the Appendix.

## 6.2 MAIN RESULTS

As shown in Table 1, we evaluate the performance and inference efficiency of PrivacyRestore and other compared methods across three privacy-preserving datasets. Compared to $d_\chi$-privacy and paraphrase, $d_\chi$-privacy on privacy spans solely apply $d_\chi$-privacy mechanism to those privacy spans and achieves higher scores in MC1/2, ROUGE-L and LLM-J. The possible reason for this is that both $d_\chi$-privacy and paraphrase operate on the entire user input, instead of specific privacy spans. Injecting noise into the entire input creates larger disturbances during inference compared to only corrupting a limited number of privacy spans.

PrivacyRestore achieves best scores in MC1/2 and LLM-J compared to other privacy-preserving methods. In terms of the ROUGE-L evaluation metric, PrivacyRestore achieve the best result in Pri-NLICE while ranking second in the other two datasets. This discrepancy likely stems from ROUGE-L's dependence on n-gram overlap between the reference text and the generated output, which does not fully reflect the quality of generated outputs. As demonstrated by the examples in Figure 6 and Appendix N, PrivacyRestore often generates outputs with different sentence structures while still providing accurate answers. Consequently, our method achieves slightly lower ROUGE-L scores but significantly higher LLM-J scores compared to $d_\chi$-privacy on privacy spans. Furthermore, the ROUGE-L metric displays larger variance than the LLM-J metric, potentially due to its sensitivity to expression rather than the underlying meaning of the generated output.

As for inference efficiency, $d_\chi$-privacy achieves the highest throughput. In contrast, $d_\chi$-privacy on privacy spans requires prior identification of privacy spans, while paraphrase necessitates rephrasing

the user input on the client side, leading to delays. PrivacyRestore also requires additional time for the prior identification and removal of privacy spans, along with constructing the meta vector on the client side. However, its throughput can reach nearly 70% on Pri-DDXPlus and 80% on Pri-NLICE and Pri-SLJA, relative to the best results.

## 6.3 EMPIRICAL PRIVACY PROTECTION RESULTS

We not only provide a theoretical privacy proof of our method in Section 5, but also implement two attack methods to empirically evaluate the privacy protection capability of our approach. PrivacyRestore sends only the meta vector and the incomplete input with privacy spans removed. It is less likely for adversaries to infer privacy spans from incomplete input. Adversaries can only attack by intercepting the meta vector and inferring the corresponding privacy spans from it. Therefore, we implement the embedding inverse attack (Li et al., 2023b; Morris et al., 2023) and attribute inference attack (Li et al., 2022), both commonly used methods for attacking embeddings. For $d_\chi$-privacy, $d_\chi$-privacy and paraphrase methods, we obtain the hidden state of the last token in the last layer in Llama2-chat-7b as the embeddings. The embedding inverse attack utilizes ROUGE-L as the evaluation metric, while the attribute inference attack employs F1 to assess attack performance. Detailed description about both attack methods and evaluation metrics can be found in Appendix M.

**Different Privacy Hyperparameter** $\epsilon$. We evaluate the attack performance of the embedding inverse attack and attribute inference attack for PrivacyRestore and other baselines. Additionally, we present the attack results without any privacy protection, which serves as the upper bound of attack performance. The values of $\epsilon$ on x-axis in Figure 2 represent the values used in PrivacyRestore. Equivalent values of $\epsilon$ and $\tau$ for other baselines are provided in the Appendix H to ensure the same privacy budget. As shown in Figure 2, although the privacy budget is the same, PrivacyRestore demonstrates better privacy protection performance, evidenced by its

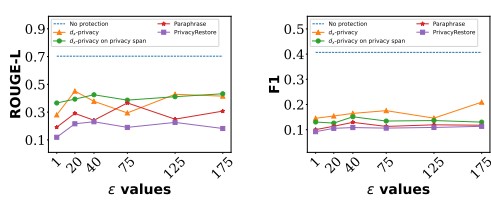

(a) Embedding Inverse Attack   (b) Attribute Inference Attack

Figure 2: Results of all methods under embedding inverse attack and attribute inference attack under different privacy hyperparameters $\epsilon$ on three datasets.

lower ROUGE-L and F1 scores. All privacy-preserving methods effectively protect privacy compared to the upper bound of no protection.

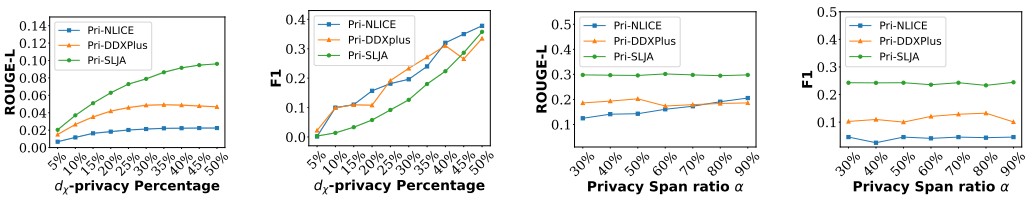

(a) Prompt Injection Attack   (b) Attribute Inference Attack   (c) Embedding Inverse Attack   (d) Attribute Inference Attack

Figure 3: (a) and (b) present the results of $d_\chi$-privacy method under the prompt injection attack and attribute inference attack under varying $d_\chi$-privacy percentages across three privacy-preserving datasets. (c) and (d) show the results of PrivacyRestore for the embedding inverse attack and attribute inference attack under different privacy span ratios $\alpha$ on the same three datasets.

**Different $d_\chi$-privacy Percentage for $d_\chi$-privacy.** We randomly select a proportion of tokens in user input to protect, denoted as the $d_\chi$-privacy percentage. A larger $d_\chi$-privacy percentage indicates a larger number of tokens being protected and a longer protected text. As illustrated in Figure 3(a) and 3(b), both prompt injection attack and attribute inference attack exhibit increased attack performance with larger $d_\chi$-privacy percentages, as reflected by the increase of ROUGE-L and F1 scores. These experimental results demonstrate the linear growth problem of the privacy budget in $d_\chi$-privacy, as proved in Section 5.1. Implementation details of attacks are presented in Appendix L.1.

**Different Privacy Span Ratio $\alpha$ for PrivacyRestore.** We randomly select a proportion of privacy spans to protect, denoted as $\alpha$, where a larger $\alpha$ indicates a larger number of privacy spans being protected and longer protected text. As shown in Figure 3(c) and 3(d), the ROUGE-L scores of embedding inverse attack are stable across different $\alpha$ values on three datasets. Similarly, the F1 scores of attribute inference attack also keep stable. The length of the protected text does not impact the privacy protection capability of PrivacyRestore. The stable performance in both attack settings provides empirical evidence that our method inherently addresses the linear growth problem of privacy budget in these DP variants, as proved in Section 5.2. The trend exhibits slight fluctuations on both the Pri-NLICE and Pri-DDXPlus datasets. A detailed analysis of these fluctuations is provided in Appendix L.2.

## 6.4 ABLATION STUDIES

| Datasets | Methods | MC1 ↑ | MC2 ↑ | ROUGE-L ↑ | LLM-J ↑ | TP ↑ |
|----------|---------|-------|-------|-----------|---------|------|
| Pri-DDXPlus | Equal Weighted Aggregation | 53.84 | 51.12 | 26.32 | 4.29 | **26.35** |
| | Attention-aware Weighted Aggregation | **62.97** | **60.19** | **27.24** | **4.47** | 26.09 |
| Pri-NLICE | Equal Weighted Aggregation | 46.92 | 45.89 | 22.78 | 3.12 | **32.75** |
| | Attention-aware Weighted Aggregation | **62.23** | **57.94** | **24.42** | **3.67** | 32.33 |
| Pri-SLJA | Equal Weighted Aggregation | 30.88 | 30.70 | 30.96 | 4.10 | **31.00** |
| | Attention-aware Weighted Aggregation | **35.47** | **35.41** | **37.56** | **5.25** | 30.73 |

Table 2: Comparison of the performance and the inference efficiency between Equal Weighted Aggregation and Attention-aware Weighted Aggregation. The best results are highlighted in **bold**.

In order to verify the effectiveness of Attention-aware Weighted Aggregation (AWA) component, we compare the performance and the inference efficiency between equal weighted aggregation and attention-aware weighted aggregation. Different from attention-aware weighted aggregation, equal weighted aggregation computes the meta vector by simply summing up all restoration vectors.

As shown in Table 2, the MC1, MC2, ROUGE-L, and LLM-J scores of equal weighted aggregation are all lower than those of attention-aware weighted aggregation, indicating that simply summing all restoration vectors equally degrades performance. This degradation is primarily due to the equal weights diluting the influence of critical spans while amplifying the effect of irrelevant ones. In terms of inference efficiency, the throughput difference between Attention-Aware Weighted Aggregation and Equal Weighted Aggregation is negligible. This suggests that the weight computation, as defined in Eq 4, is efficient and does not significantly impact overall throughput.

## 6.5 ANALYSIS OF HYPERPARAMETER AND LLM BACKBONE

We analyze the performance of PrivacyRestore using different numbers of edited heads $K$. In addition, we analyze the performance of PrivacyRestore using a different LLM backbone (i.e., Llama-13b-chat). Due to space limitation, we put the analysis in the Appendix I and Appendix K.

## 7 CONCLUSION

We propose PrivacyRestore which protects the privacy within user inputs during inference in online LLM inference services. PrivacyRestore achieves privacy protection by directly removing privacy spans in the user input and then restoring these privacy spans via activation steering. PrivacyRestore provides a practical and efficient solution for protecting privacy while maintaining satisfactory performance and inference efficiency. We demonstrate that PrivacyRestore inherently addresses the linear growth problem of the privacy budget found in differential privacy variants. We curate three privacy-preserving datasets covering medical and legal fields, and PrivacyRestore achieves strong performance and inference efficiency across all datasets. Additionally, we implemented two types of attacks, and the experimental results demonstrate PrivacyRestore's robust privacy protection capabilities.

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

# A  NOTATIONS

Here we present all notations used in our paper in Table 3.

# B  DATASETS

## B.1  CONSTRUCTION PROCESS

We used GPT-3.5 (Ouyang et al., 2022) to classify symptoms in DDXPlus and NLICE, as well as case details in SLJA, into five levels ranging from non-sensitive to highly sensitive. The assessment prompt template is shown in Appendix O.1. A higher level indicates that the symptom or case detail is more sensitive. We define all symptoms and case details with a sensitivity level greater than 3 as privacy spans.

We assign each sample a correct answer along with three randomly selected incorrect options. For DDXPlus and NLICE, we randomly select three diagnosis results to combine with the correct diagnosis as the choices. In the SLJA dataset, we randomly select three legal judgments to pair with the correct one as the options.

The initial dataset is extensive, and we observed that for most samples, removing all privacy spans often yields outputs similar to those obtained when privacy spans are provided. Privacy preserving for these samples is meaningless because users can directly hide those privacy spans and obtain approximate result outputs. In real-world scenarios, sensitive privacy spans often play a crucial role in medical diagnoses and legal judgments, making privacy preservation highly valuable. Our dataset is designed to benchmark various privacy-preserving methods and must include samples where privacy spans are crucial for generating outputs. We utilize the KL divergences to measure the importance scores of samples. We calculate the KL divergence between the model output distributions with and without the privacy symptoms included. A higher KL divergence indicates that the absence of sensitive privacy spans may lead to different or incorrect outputs. We selected only samples with high KL divergence to construct the privacy-preserving datasets. As a result, we curated three privacy-preserving datasets: Pri-DDXPlus and Pri-NLICE for medical diagnosis, and Pri-SLJA for legal judgment.

## B.2  STATISTICAL INFORMATION

We show the statistics of the obtained Pri-DDXPlus, Pri-NLICE and Pri-SLJA datasets in Table 4. We tally the number of user inputs, privacy span types, and answer types. We also compute the average occurrence of privacy spans per instance. In Pri-DDXPlus and Pri-NLICE, the privacy spans are the symptoms, and the answers are the diagnoses. In Pri-SLJA, the privacy spans are the case details, and the answers are the legal judgments.

Pri-DDXPlus commonly contains more instances and more privacy symptoms types compared to Pri-NLICE and Pri-SLJA. Each sample in Pri-DDXPlus contains an average of six privacy symptoms, while samples in Pri-NLICE have an average of four privacy spans, and samples in Pri-SLJA have an average of three privacy spans.

# C  PRELIMINARIES FOR METHODOLOGY

The $d_\chi$-privacy protection and activation steering technique are two crucial components of our method. Here, we provide an illustration of these techniques for better understanding of our method.

## C.1  $d_\chi$-PRIVACY PROTECTION

$d_\chi$-privacy (Feyisetan et al., 2019) is a variant of the differential privacy mechanism designed to protect privacy by incorporating a distance measure into the privacy budget. Its detailed definition is provided in Definition 2.1. Typically, to implement the $d_\chi$-privacy mechanism, noise is added to the

| Notations | Definitions |
|---|---|
| $s$ | A single privacy span. |
| $\mathcal{S}$ | All possible privacy spans. |
| $\mathcal{S}_I$ | All privacy spans in user input $I$. |
| $h$ | A single edited head. |
| $\mathcal{H}_c$ | The common top-K heads set. |
| $\mathcal{H}_a$ | The set of all heads. |
| $\mathcal{H}_k^s$ | The top-K heads set of the privacy span $s$. |
| $L_h$ | The score list of the head $h$ across all privacy spans. |
| $K$ | The number of selected edited heads. |
| $\mathcal{F}_h^s$ | The probe of privacy span $s$ on head $h$. |
| $\theta_h^s$ | The parameters of the probe $\mathcal{F}_h^s$. |
| $\sigma$ | The sigmoid function. |
| $\mathbf{u}_h$ | The output hidden state on head $h$. |
| $\bar{\mathbf{u}}_h$ | The output hidden state after restoration on head $h$. |
| $r_s^h$ | The restoration vector for privacy span $s$ on head $h$. |
| $\Theta$ | All restoration vectors for all privacy spans on all edited heads. |
| $\lambda$ | The tradeoff hyperparameter of ORPO loss. |
| $w_s$ | The weight of privacy span $s$. |
| $n$ | The number of tokens in the user input. |
| $n_h$ | The number of heads in the lightweight model. |
| $\text{Attn}_h(x, y)$ | The attention score of $y$ attending to $x$ on head $h$. |
| $Z_h$ | The normalized weighted sum of restoration vectors on head $h$. |
| $Z_h, Z_h'$ | Any two normalized weighted sums. |
| $\mathcal{R}_h$ | The meta vector on head $h$. |
| $\mathcal{N}$ | The added noise on the normalized weighted sums for meta vector construction. |
| $\|x\|_2$ | The $l_2$-normalization of $x$ |
| $I_{all} = \{I_1, ..., I_m\}$ | All user inputs in the training set. |
| $Y_s = \{y_1, ..., y_m\}$ | The labels indicating whether the corresponding input contains $s$. |
| $m$ | The size of training set. |
| $I, I'$ | Any two user inputs. |
| $I = \{i_1, ..., i_n\}$ | The tokens of the input $I$. |
| $i_t$ | The $t$-th token of the user input $I$. |
| $\{e_1, ..., e_n\}$ | The token embeddings of the input $I$. |
| $O$ | The possible output sets for $I$. |
| $O = \{o_1, ..., o_n\}$ | The possible output sets for tokens of the input $I$, also represented as $O$. |
| $\hat{I}$ | The user input with all privacy spans removed. |
| $\hat{I}_{all} = \{\hat{I}_1, ..., \hat{I}_m\}$ | All user inputs with privacy spans removed in the training set. |
| $a$ | The initial output given the complete input $I$. |
| $\hat{a}$ | The output given the incomplete input with privacy spans removed $\hat{I}$. |
| $Q, Q'$ | Any two queries to the database. |
| $Q = \{q_1, ..., q_n\}$ | The sub-queries of the query $Q$. |
| $G$ | The possible query result sets for $Q$. |
| $G = \{g_1, ..., g_n\}$ | The possible result sets for sub-queries of the query $Q$, also represented-as $G$. |
| $\epsilon$ | The privacy hyperparameter. |
| $\tau$ | The generation temperature. |
| $\delta$ | The privacy hyperparameter. |
| $n_{ps}$ | The number of tokens associated with the privacy spans in the input. |
| $\alpha$ | The proportion of privacy spans selected for protection. |
| $d_\chi$ | Any distance function used by $d_\chi$-privacy. |
| $d_e$ | The prior value of distance between token embeddings. |
| $d_z$ | The prior value of distance between normalized weighted sums. |

Table 3: Definitions of all notations used in our paper.

| Datasets | Dataset Split | User inputs | Privacy Span Type | Avg. Privacy Spans |
|---|---|---|---|---|
| | All | 7759 | 149 | 5.95 |
| Pri-DDXPlus | Train | 5901 | 149 | 6.03 |
| | Dev | 309 | 60 | 5.37 |
| | Test | 1549 | 78 | 5.77 |
| | All | 4062 | 64 | 4.49 |
| Pri-NLICE | Train | 3282 | 64 | 4.55 |
| | Dev | 130 | 58 | 4.25 |
| | Test | 650 | 64 | 4.24 |
| | All | 3901 | 142 | 2.67 |
| Pri-SLJA | Train | 3117 | 142 | 2.56 |
| | Dev | 130 | 95 | 3.21 |
| | Test | 654 | 142 | 3.09 |

Table 4: The statistics of Pri-DDXPlus and Pri-NLICE. Average privacy symptoms indicate the average privacy spans occur in one query.

initial embedding or vector for privacy protection, as follows:

$$\mathcal{R} = Z + \mathcal{N}, \tag{8}$$

$$\mathbb{P}(\mathcal{N}) \propto \exp(-\epsilon||\mathcal{N}||), \tag{9}$$

where $Z$ is the protected embedding/vector, $\mathcal{N}$ is the added noise, $\mathcal{R}$ is the protected results and $\epsilon$ is the privacy parameter of the mechanism. According to Feyisetan et al. (2019), in order to sampling the noise $\mathcal{N}$ from the distribution, we can compute as the following:

$$\mathbf{v} \in \{v \in \mathbb{R}^n : ||v|| = 1\} \tag{10}$$

$$\mathbb{P}(\mathbf{l}) \propto \frac{\mathbf{l}^{n-1}e^{-\epsilon\mathbf{l}}}{\Gamma(n)\epsilon^{-n}}, \tag{11}$$

$$\mathcal{N} = \mathbf{l} \cdot \mathbf{v}, \tag{12}$$

where $n$ is the size of the embedding/vector and $\epsilon$ is the privacy parameter.

### C.2 ACTIVATION STEERING TECHNIQUE

Activation steering methods (Li et al., 2023c; Turner et al., 2023; Hernandez et al., 2023) control the behavior of LLM by modifying their activations during the inference stage, without incurring training costs. It serves as a crucial part of our methodology to restore privacy information during LLM inference. Typically, the attention mechanism (Vaswani et al., 2017) in LLM is responsible for capturing contextual information, and it can be expressed as:

$$q = W_q \cdot \mathbf{i}, \tag{13}$$

$$\mathbf{u} = \text{Softmax}(\frac{q \cdot K^T}{\sqrt{d_k}}) \cdot V, \tag{14}$$

where $\mathbf{i}$ is the input hidden state, $\mathbf{u}$ is the output hidden state, $W_q$ is the query weight matrix, $K$ is the key of the context and $V$ is the value of the context and $d_k$ is the dimension of the key. Activation steering methods add some steering vectors into the output hidden state and in our methos we add the restoration meta vector into the output hidden state to restore privacy information, which can be expressed as:

$$\mathbf{u} = \mathbf{u} + \mathcal{R}, \tag{15}$$

where $\mathcal{R}$ is the steering/restoration meta vector.

## D FORMAL DEFINITIONS OF CDP AND LDP

### D.1 CENTRALIZED DIFFERENTIAL PRIVACY (CDP)

CDP protects individual privacy only **after data has been aggregated in a central repository**, which is defined as,

**Definition D.1.** *(CDP). A randomized mechanism $\mathcal{M} : \mathcal{Q} \to \mathcal{G}$ fulfills $(\epsilon, \delta)$-differential privacy if for all adjacent queries $Q, Q' \in \mathcal{Q}$ and all possible query results $G \subset \mathcal{G}$,*

$$\mathbb{P}\left(\mathcal{M}(Q) \in G\right) \leq \exp(\epsilon)\mathbb{P}\left(\mathcal{M}(Q') \in G\right) + \delta.$$

CDP (Dwork et al., 2016) ensures that adversaries cannot distinguish between $Q$ and $Q'$ based on $G$ due to the similar probabilities, meaning the query results are probabilistically indistinguishable. This prevents adversaries from inferring characteristics about the repository based on multiple queries and their results.

## D.2 LOCAL DIFFERENTIAL PRIVACY (LDP)

However, privacy risks can also emerge during the data collection process itself, as attackers may intercept user inputs while they are being transmitted to the central repository. LDP (Duchi et al., 2013) protect user inputs **during transmission process** by ensuring that attackers cannot distinguish between any two adjacent inputs, which is defined as,

**Definition D.2.** *(LDP). A randomized mechanism $\mathcal{M} : \mathcal{I} \to \mathcal{O}$ fulfills $(\epsilon, \delta)$-differential privacy if for all adjacent inputs $I, I' \in \mathcal{I}$ and all possible outputs $O \subset \mathcal{O}$,*

$$\mathbb{P}\left(\mathcal{M}(I) \in O\right) \leq \exp(\epsilon)\mathbb{P}\left(\mathcal{M}(I') \in O\right) + \delta.$$

The mechanism $\mathcal{A}$ processes the user input before transmitting it. LDP ensures that, even if attackers intercept $O$, they cannot distinguish between the initial user input $I$ and the adjacent one $I'$.

## E PROOF OF THEOREM 5.1

**Proof of $d_\chi$-privacy**. As shown in Definition 2.1, if the input length is 1, indicating a single token $i_1$, $d_\chi$ can be:

$$\mathbb{P}\left(\mathcal{M}(i_1) \in o_1\right) \leq \exp(\epsilon d_\chi(i_1, i_1'))\mathbb{P}\left(\mathcal{M}(i_1') \in o_1\right),$$

where $o_1$ is the possible output set for $\mathcal{M}(i_1)$ and the privacy budget is $\epsilon d_\chi(i_1, i_1')$. When the input becomes the sequential tokens $I = \{i_1, i_2, ..., i_n\}$ with corresponding output sets $O = \{o_1, ..., o_n\}$, the LCP for the sequence of length $n$ is:

$$\begin{aligned}
\mathbb{P}\left(\mathcal{M}(I) \in O\right) &= \mathbb{P}\left(\mathcal{M}(i_1) \in o_1\right) \cdot \mathbb{P}\left(\mathcal{M}(i_2) \in o_2\right) \cdot ... \cdot \mathbb{P}\left(\mathcal{M}(i_n) \in o_n\right) \\
&\leq [\exp(\epsilon d_\chi(i_1, i_1'))\mathbb{P}\left(\mathcal{M}(i_1') \in o_1\right)] \cdot ... \cdot [\exp(\epsilon d_\chi(i_n, i_n'))\mathbb{P}\left(\mathcal{M}(i_n') \in o_n\right)] \\
&= \exp[\epsilon(d_\chi(i_1, i_1') + ... + d_\chi(i_n, i_n'))]\mathbb{P}\left(\mathcal{M}(I') \in O\right) \\
&= \exp[\epsilon \sum_{j=0}^{n} d_\chi(i_j, i_j')]\mathbb{P}\left(\mathcal{M}(I') \in O\right),
\end{aligned}$$

where the privacy budget is $\epsilon \sum_{j=1}^{n} d_\chi(i_j, i_j')$. Commonly, we use the Euclidean distance as the $d_\chi$ function and obviously $\sum_{j=1}^{n} d_\chi(i_j, i_j') \propto n$. Therefore, the privacy budget of LDP grows linearly with the length $n$.

**Proof of LDP**. As stated in Definition D.2, if the length of input is 1, corresponding to a single token $i_1$, LDP can be expressed as:

$$\mathbb{P}\left(\mathcal{M}(i_1) \in o_1\right) \leq \exp(\epsilon)\mathbb{P}\left(\mathcal{M}(i_1') \in o_1\right) + \delta,$$

where $o_1$ is the possible output set of $\mathcal{M}(i_1)$ and the privacy budget is controlled by $(\epsilon, \delta)$. Considering the sequence tokens $I = \{i_1, i_2, ..., i_n\}$ and corresponding output sets $O = \{o_1, o_2, ..., o_n\}$, the CDP for the sequence of length $n$ can be written as:

$$\begin{aligned}
\mathbb{P}\left(\mathcal{M}(I) \in O\right) &= \mathbb{P}\left(\mathcal{M}(i_1) \in o_1\right) \cdot \mathbb{P}\left(\mathcal{M}(i_2) \in o_2\right) \cdot ... \cdot \mathbb{P}\left(\mathcal{M}(i_n) \in o_n\right) \\
&\leq [\exp(\epsilon)\mathbb{P}\left(\mathcal{M}(i_1') \in o_1\right) + \delta] \cdot ... \cdot [\exp(\epsilon)\mathbb{P}\left(\mathcal{M}(i_n') \in o_n\right) + \delta] \\
&= \exp(n\epsilon)\mathbb{P}\left(\mathcal{M}(I') \in O\right) + \delta \cdot \sum_{i=1}^{n} \prod_{j!=i} \mathbb{P}\left(\mathcal{M}(i_j') \in o_j\right) + \delta^2 \cdot ...,
\end{aligned}$$

where $\delta$ is typically considered a very small value. When $\delta$ approaches 0, we consider only the first two terms and then,

$$\mathbb{P}\left(\mathcal{M}(I) \in O\right) \leq \exp(n\epsilon)\mathbb{P}\left(\mathcal{M}(I') \in O\right) + \delta \cdot \sum_{i=1}^{n} \prod_{j!=i} \mathbb{P}\left(\mathcal{M}(i'_j) \in o_j\right),$$

which indicating the privacy budget becomes $\left(n\epsilon, \delta \cdot \sum_{i=1}^{n} \prod_{j!=i} \mathbb{P}\left(\mathcal{M}(i'_j) \in o_j\right)\right)$, according to the Definition of LDP in Section D.2 . The first term $n\epsilon$ obviously grows linearly with the length $n$. The second term can be view as $\delta$ multiplied by $\sum_{i=1}^{n} \prod_{j!=i} \mathbb{P}\left(\mathcal{M}(i'_j) \in o_j\right)$. The second term summarizes $n$ multiplicative terms, each bounded within $(0, 1)$. The second term can be approximately considered to grow linearly with $n$. Therefore, the privacy budget of CDP also grows linearly with the length $n$.

**Proof of CDP**. The definition of CDP, as shown in Section D.1, is similar to LDP, with the only difference being that CDP applies to the user query $Q = \{q_1, q_2, ..., q_n\}$ rather than the text input $I = \{i_1, i_2, ..., i_n\}$. If we consider the token $i_n$ as a sub-query $q_n$, then similarly the definition of CDP for sequential sub-queries $Q = \{q_1, q_2, ..., q_n\}$ can be:

$$\mathbb{P}\left(\mathcal{M}(Q) \in G\right) \leq \exp(n\epsilon)\mathbb{P}\left(\mathcal{M}(Q') \in G\right) + \delta \cdot \sum_{i=1}^{n} \prod_{j!=i} \mathbb{P}\left(\mathcal{M}(q'_j) \in g_j\right),$$

where $G = \{g_1, g_2, ..., g_n\}$ are the possible output sets for sequence inputs $Q$. The privacy budget is $\left(n\epsilon, \delta \cdot \sum_{i=1}^{n} \prod_{j!=i} \mathbb{P}\left(\mathcal{M}(q'_j) \in g_j\right)\right)$ and also grows linearly with the length $n$.

# F    PROOF OF THEOREM 5.2

As shown in Figure 1, during the inference stage, only the meta vector and the query with privacy spans removed are transmitted from the client to the server. The meta vector holds information about all privacy spans and could be vulnerable to interception by adversaries who may attempt to reverse-engineer these spans.

PrivacyRestore protects the meta vector by adding noise $\mathcal{N}$ which is sampling from the distribution $p(\mathcal{N}) \propto \exp(-\epsilon\|\mathcal{N}\|)$, before transmission, as shown in Eq 6. Firstly, to simply the question, we only consider the situation of only one edited head $h$. Assume $Z_h$ represents the normalized weighted sum of restoration vectors of all privacy spans on head $h$ without adding noise, $\mathcal{R}_h$ denotes the vector on head $h$ after adding noise, as shown in Eq 5 and 6. The process of adding noise can be represented by $\mathcal{M}$. Then, the possibility that $Z_h$ becomes $\mathcal{R}_h$ after adding noise $\mathcal{N}$ is

$$\begin{aligned}
\mathbb{P}(\mathcal{M}(Z_h) = \mathcal{R}_h) &= \mathbb{P}(Z_h + \mathcal{N} = \mathcal{R}_h) \\
&= \mathbb{P}(\mathcal{N} = \mathcal{R}_h - Z_h) \\
&= \exp(-\epsilon\|\mathcal{R}_h - Z_h\|).
\end{aligned}$$

Then for any normalized weighted sums on head $h$, $Z_h$ and $Z'_h$, we have

$$\begin{aligned}
\frac{\mathbb{P}[\mathcal{M}(Z_h) = \mathcal{R}_h]}{\mathbb{P}[\mathcal{M}(Z'_h) = \mathcal{R}_h]} &= \frac{\exp(-\epsilon\|\mathcal{R}_h - Z_h\|)}{\exp(-\epsilon\|\mathcal{R}_h - Z'_h\|)} \\
&= \exp(\epsilon(\|\mathcal{R}_h - Z'_h\| - \|\mathcal{R}_h - Z_h\|)) \\
&\leq \exp(\epsilon\|Z'_h - Z_h\|).
\end{aligned}$$

According to the definition of $d_\chi$-privacy in Section 2.1, the mechanism $\mathcal{M}$ satisfies $d_\chi$-privacy. In other words, by adding noise $\mathcal{N}$, adversaries cannot infer $Z_h$ from $\mathcal{R}_h$ even if $\mathcal{R}_h$ is intercepted. Moreover, the privacy budget of our methods is $\epsilon\|Z'_h - Z_h\|$.

Then for multiple edited heads, the only difference is to concatenated all restoration vectors $\mathcal{R}_h$ to form a single vector, as shown in Section 4.2. Then, the concatenated vector is also added with noise to protect privacy and the privacy budget of our methods become $\epsilon\|Z' - Z\|$, where $Z'$ and $Z$ represent any pair of normalized weighted sums of restoration vectors concatenated across all edited heads. It is independent of the input length $n$ and depends on the hyperparameter $\epsilon$ the term $\|Z' - Z\|$. Therefore PrivacyRestore fulfills $d_\chi$-privacy and provides a privacy budget $\epsilon\|Z' - Z\|$ which is independent of the input length and inherently addresses the problem of the linear growth of privacy budget.

## G  TOP-K HEADS SELECTOR ALGORITHM

In the section, we present the detail implementation of Top-K Heads Selector, as shown in Algorithm 1. Firstly, we initialize an empty score list $L_h$ for each head. Secondly, each privacy span $s$ has its corresponding top-K heads set $\mathcal{H}_k^s$. For each head $h$ in $\mathcal{H}_k^s$, we append $\text{Score}(h, \mathcal{H}_k^s)$ into its score list $L_h$. $\text{Score}(h, \mathcal{H}_k^s)$ is defined as the rank of head $h$ among $\mathcal{H}_k^s$ in ascending order based on the accuracy of the probe associated with the head $h$ and privacy span $s$. Thirdly, we calculate the average value of each score list $L_h$ as the score of the corresponding head $h$. Finally, we sort all heads in the LLM by the scores and pick up top-K heads as the common top-K heads set $\mathcal{H}_c$.

---

**Algorithm 1** Top-K Heads Selector

---

**Input:** $\mathcal{S}$ is the set of privacy spans; $\mathcal{H}_a$ is the set of all heads; $\mathcal{H}_k^s$ is the top-K heads set of the privacy span $s$; $\text{Score}(h, \mathcal{H}_k^s)$ return the rank of head $h$ among $\mathcal{H}_k^s$ in ascending order based on the accuracy of the probe associated with the head $h$ and privacy span $s$. The score of the head with lowest accuracy is 1. The score of the head with highest accuracy is $K$.

1: Initialize an empty score list $L_h = [\ ]$ for each head $h$ in $\mathcal{H}_a$.
2: **for** $s$ in $\mathcal{S}$ **do**
3:     **for** $h$ in $\mathcal{H}_k^s$ **do**
4:         Append $\text{Score}(h, \mathcal{H}_k^s)$ into $L_h$.
5:     **end for**
6: **end for**
7: **for** $h$ in $\mathcal{H}_a$ **do**
7:     $\text{score}_h = \text{average}(L_h)$
8: **end for**
9: Sort $\mathcal{H}_a$ according to $\text{score}_h$ and select top $K$ heads to obtain **common top-K heads set** $\mathcal{H}_c$.

**Output:** $\mathcal{H}_c$ is the common top-K heads set.

---

## H  SETTINGS OF PRIVACY HYPERPARAMETER $\epsilon$

As demonstrated in Appendix E, the privacy budget of $d_\chi$-privacy is $\epsilon n d_e$ and the privacy budget of $d\chi$-privacy on privacy spans is $\epsilon n_{sp} d_e$, where $n$ represents the length of user inputs, $n_{sp}$ denotes the length of privacy spans, $d_e$ denotes the average distance between word embeddings and $\epsilon$ is the privacy hyperparameter. In addition, as pointed by Mattern et al. (2022b); Utpala et al. (2023), the privacy budget of paraphrase method is $2n/\tau$, where $\tau$ is the generation temperature used during paraphrasing, and $n$ represents the average length of user inputs. To ensure same privacy budget for fair comparison, we need to determine the values of different hyperparameters for different methods on different datasets.

The privacy budget of PrivacyRestore is $\epsilon d_z$, according to Appendix F, where $d_z$ denotes the average distance between normalized weighted sums, computed as $||Z_h' - Z_h||$. We set the privacy hyperparameter $\epsilon$ to 75.00.

To achieve the same privacy budget when using other privacy-preserving methods, we first analyze the distribution of users inputs lengths $n$, privacy spans lengths $n_{sp}$, distances between word embeddings $d_e$ and distances between normalized weighted sums $d_z$ across three privacy-preserving datasets in Figure 4. The average values of these distributions are used as prior estimates for the corresponding parameters. We then compute the corresponding $\epsilon$ for $d_\chi$-privacy (on privacy spans) and $\tau$ for paraphrase across different datasets. The privacy hyperparameter $\epsilon$ and $\tau$ for different baselines across three privacy-preserving datasets are shown in Table 5.

## I  HYPERPARAMETER ANALYSIS

We evaluate the performance of our methods using different numbers of edited heads, $K$, across the development sets of three privacy-preserving datasets. For simplicity, we compute MC2 to represent classification performance, LLM-J to measure generation performance, and TP to indicate inference efficiency. As shown in Table 6, according to the MC2 score, the optimal value of $K$ is 175 for the Pri-DDXPlus and Pri-SLJA datasets, and 125 for the Pri-NLICE dataset. The performance

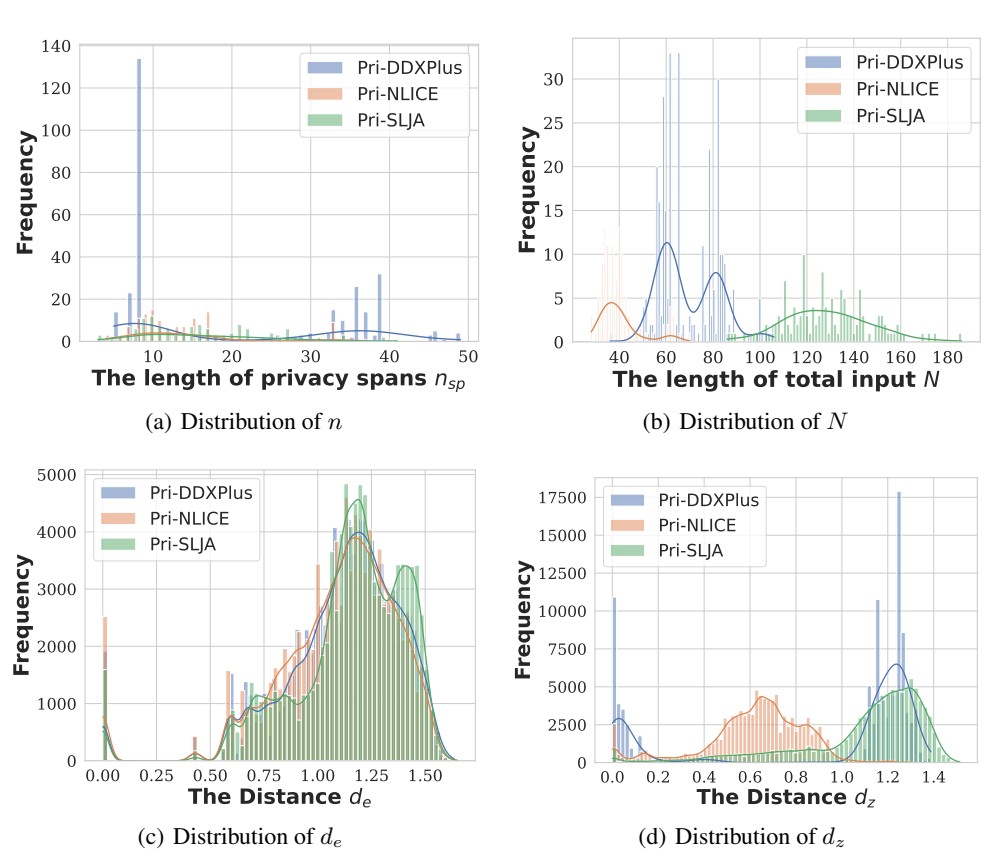

(a) Distribution of $n$

(b) Distribution of $N$

(c) Distribution of $d_e$

(d) Distribution of $d_z$

Figure 4: The distributions of privacy span lengths $n_{sp}$, total input lengths $n$, distances between word embeddings $d_e$, and distances between normalized weighted sums $d_z$ on the dev set across all three privacy-preserving datasets are analyzed. The average values from these distributions are used as prior estimates for the corresponding parameters.

| | $d_\chi$-privacy | | | $d_\chi$-privacy on privacy spans | | | Paraphrase | | PrivacyRestore | | Total Privacy Budget |
|---|---|---|---|---|---|---|---|---|---|---|---|
| Datasets | $n$ | $d_e$ | $\epsilon$ | $n_{sp}$ | $d_e$ | $\epsilon$ | $n$ | $\tau$ | $d_z$ | $\epsilon$ | |
| Pri-DDXPlus | 69.63 | 1.12 | **0.87** | 20.28 | 1.12 | **3.00** | 69.63 | **2.04** | 0.91 | **75.00** | **68.25** |
| Pri-NLICE | 40.14 | 1.09 | **1.06** | 14.08 | 1.09 | **3.03** | 40.14 | **1.72** | 0.62 | **75.00** | **46.50** |
| Pri-SLJA | 129.76 | 1.15 | **0.54** | 16.33 | 1.15 | **4.35** | 129.76 | **3.17** | 1.09 | **75.00** | **81.75** |

Table 5: The settings of privacy hyperparameters for different baselines across all privacy-preserving datasets.

degradation as $K$ increases can be attributed to the cumulative effect of multiple edited heads. As more heads are modified, the activations progressively deviate from their initial values, potentially compromising the LLM's general capabilities. Moreover, throughput increases with larger $K$ because we need to inject the meta vector for each head in $\mathcal{H}_c$ using Eq 7 on the server. Consequently, more heads indicate more injections, which increases the inference time on the server.

| Datasets | Metrics | $K = 75$ | $K = 100$ | $K = 125$ | $K = 150$ | $K = 175$ | $K = 200$ |
|---|---|---|---|---|---|---|---|
| Pri-DDXplus | MC2 ↑ | 52.20 | 56.17 | 59.39 | 58.96 | **62.95** | 62.64 |
| | LLM-J ↑ | 4.51 | 4.38 | 4.45 | 4.33 | **4.71** | 4.55 |
| | TP ↑ | **24.31** | 21.51 | 19.72 | 20.07 | 22.68 | 21.91 |
| Pri-NLICE | MC2 ↑ | 37.15 | 51.01 | **58.97** | 51.89 | 58.11 | 58.45 |
| | LLM-J ↑ | 3.27 | 3.66 | **3.80** | 3.44 | 3.40 | 3.62 |
| | TP ↑ | **20.05** | 19.14 | 18.23 | 16.08 | 15.89 | 15.48 |
| Pri-SLJA | MC2 ↑ | 28.75 | 30.65 | 35.07 | 32.41 | **35.13** | 32.08 |
| | LLM-J ↑ | 5.21 | **5.41** | 5.00 | 5.33 | 5.15 | 5.28 |
| | TP ↑ | **36.28** | 35.25 | 34.62 | 32.97 | 30.51 | 29.87 |

Table 6: The performance of PrivacyRestore on the development set using various numbers of edited heads $K$. MC2 reflects classification capability, while LLM-J indicates generation performance. The TP assesses inference efficiency. We report results across three datasets to identify the optimal $K$ for each datasets. The best results are highlighted in bold.

## J    CALCULATION PROCESS OF MC1 AND MC2

We evaluate the model's classification ability using two metrics: **MC1** and **MC2** (Lin et al., 2021). We assign each sample in Pri-DDXPlus, Pri-NLICE and Pri-SLJA with four options, including one correct answer and three incorrect ones. The details of calculation process is as follows:

**Calculation of MC1.** For each user input, we select the option with the highest probability as the model's choice. MC1 is defined as the model's accuracy, which is calculated as the proportion of correctly answered inputs.

**Calculation of MC2.** For each user input, we compute the normalized probability of the correct answer among the four options. The average of these normalized probabilities across all inputs is calculated as the MC2 score.

## K    VARYING LLM BACKBONE

We evaluate the performance of PrivacyRestore and other privacy-preserving baselines on a larger model, Llama-13b-chat. As shown in Figure 5, PrivacyRestore outperforms the other baselines in terms of both MC2 and LLM-J values across all three privacy-preserving datasets. Notably, the performance of all privacy-preserving methods on the larger model, Llama-13b-chat, is worse than on the smaller model, Llama-7b-chat. This suggests that as model size increases, the model becomes more sensitive to the injected disturbances introduced by these privacy-preserving methods, leading to performance degradation.

## L    SUPPLEMENTS OF EMPIRICAL PRIVACY PROTECTION

We present empirical evidence of the privacy protection capabilities of $d_\chi$-privacy and PrivacyRestore by implementing various attacks on these privacy-preserving methods. Lower attack performance indicates stronger privacy protection provided by these methods.

### L.1    EMPIRICAL PRIVACY PROTECTION OF $d_\chi$-PRIVACY

As presented by Feyisetan et al. (2019; 2020), the $d_\chi$-privacy mechanism protects input by injecting noise into the token embeddings and replacing the original tokens with their nearest neighbors. To attack the garbled text input, we implement two types of attacks: prompt injection attack (Suo, 2024)

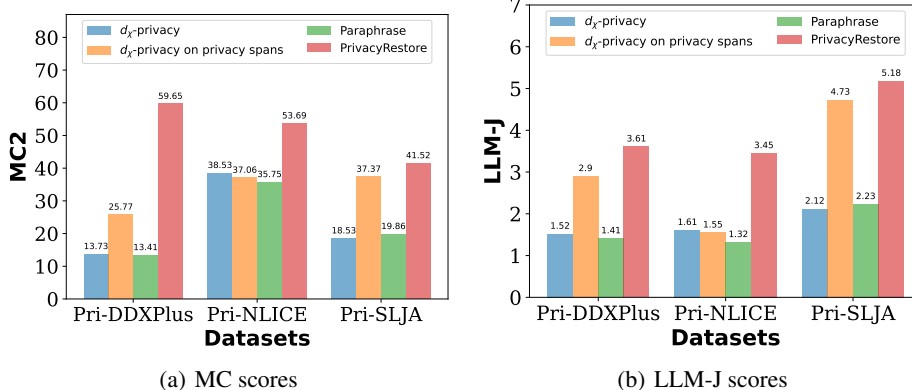

(a) MC scores              (b) LLM-J scores

Figure 5: The MC2 and LLM-J results of PrivacyRestore and other privacy-preserving baselines on larger model, Llama-13b-chat, across three datasets.

and attribute inference attack (Li et al., 2022), both commonly used for attacking text inputs. In prompt injection attack, additional instructions are added before and after the garbled text input, prompting the model to output the original text. The attack's performance is measured by calculating the ROUGE-L score between the generated text and the original input. Attribute inference attack performs classification on the garbled text, where the target labels are the token IDs of the original input. The attack's performance is evaluated using the classification F1 score. The details of these attack methods are presented in Appendix M.

### L.2 EMPIRICAL PRIVACY PROTECTION OF PRIVACYRESTORE

As stated in Section 6.3, the attack object of PrivacyRestore is the meta vector and we implement two types of attack: the embedding inverse attack (Li et al., 2023b; Morris et al., 2023) and attribute inference attack (Li et al., 2022), both commonly used methods for attacking embeddings. Embedding inverse attack utilizes the generative model to generate the privacy spans from the meta vector. Embedding inverse attack measure the attack performance by computing the ROUGE-L between the generate output and the privacy spans. Attribute inference attack performs classification on the meta vector and the target labels are the token IDS of the privacy spans. Attribute inference attack compute the F1 score to evaluate the attack performance. The details of these attack methods are presented in Appendix M.

As shown in Figure 3(c) and 3(d), the ROUGE-L score for the embedding inverse attack remains nearly stable across different $\alpha$ values in the Pri-SLJA and Pri-DDXPlus datasets. What's a little strange is the ROUGE-L score in the Pri-NLICE dataset shows a slight increase. The possible reason is that higher ratio indicating more privacy spans consider and resulting longer reference string when compute the ROUGE-L score. Since ROUGE-L measures the overlap between the generated output and the reference string, a longer reference string may slightly boost the score. The F1 score for the attribute inference attack remains stable across all three datasets. The stable performance in both attack scenarios provides empirical support for Theorem 5.2.

## M  DETAILED IMPLEMENTATION OF ATTACK METHODS

### M.1  PROMPT INJECTION ATTACK

$d_\chi$-privacy injects noise into the original user inputs and transmits the garbled inputs to the server to protect the privacy spans. We employ prompt injection attack to recover the initial question from the garbled inputs. Following Perez & Ribeiro (2022); Suo (2024), we insert additional instructions before and after the user inputs to prompt the model to output the original user input rather than the normal response. The template for the additional instructions is provided in Appendix O.4.

We set the maximum generation length for the prompt injection attack to 256 tokens. To evaluate the attack's performance, we calculate the ROUGE-L score between the generated output and the

original user input. A higher ROUGE-L score indicates greater overlap between the recovered text and the original input, signifying more successful attack results.

## M.2 ATTRIBUTE INFERENCE ATTACK

Attribute inference attack attempts to steal user inputs by performing classification on the garbled inputs, where the target labels correspond to the token IDs of the original inputs. Since each input contains multiple tokens, this classification task is naturally a multi-label classification problem. Following Li et al. (2022), we utilize a multi-layer perceptron (MLP) model as the classifier. The input dimension is 4096 and the output dimension is the size of the whole vocabulary size. To evaluate the attack's performance, we calculate the F1 score of the classification, where a higher F1 score indicates a more successful attack. The attack targets can include garbled text from $d_\chi$-privacy, paraphrased text, or the meta vector from PrivacyRestore. The implementation details for text and vectors may vary slightly.

**Attribute inference attack on meta vector**. To attack the meta vector from PrivacyRestore, we can directly use a fully-connected layer to transform the meta vector's dimension from $128 \times K$ to the classifier's input dimension 4096. We then perform classification on the transformed meta vector.

**Attribute inference attack on text**. For garbled text from $d_\chi$-privacy or paraphrased text, we first transform the text into a vector representation. We utilize Llama2-chat-7b to process the text input and obtain the last token's hidden state as the vector representation. Classification is then performed on this hidden state.

## M.3 EMBEDDING INVERSE ATTACK

Different from attribute inference attack, embedding inverse attack steal the user inputs through the generative model to generate the original user inputs. We utilize the GPT-2 model (Radford et al., 2019) as the generative model and set the maximum generation length to 256. We finetune the GPT-2 model on the training set for 20 epoch using the learning rate of 1e-5. To evaluate the attack's performance, we compute the ROUGE-L score between the generated output of the GPT-2 model and the original user input, where higher scores indicate better attack effectiveness. Similar to the attribute inference attack, the implementation of the embedding inverse attack differs between text and meta vectors.

**Embedding inverse attack on meta vector**. We use a fully-connected layer to transform the meta vector's dimension to the dimension of hidden state of GPT-2 model. Then we directly input the transformed meta vector as the input embedding.

**Embedding inverse attack on text**. We use Llama-2-chat-7b to process the text input and extract the last token's hidden state as the vector representation. This vector is then transformed by a fully connected layer to match the hidden state dimension of the GPT-2 model. Finally, it is fed into the GPT-2 model as the input embedding for subsequent generation.

## N EXAMPLE OUTPUTS OF PRIVACYRESTORE

We provide some example outputs of our method in Figure 6. As shown in these examples, applying $d_\chi$-privacy to privacy spans results in outputs with higher ROUGE-L scores but lower LLM-J scores compared to our method. After analyzing these outputs in detail, the high ROUGE-L scores from $d_\chi$-privacy on privacy spans likely result from a greater overlap with the initial output. However, the overlapping sections consist mainly of meaningless sentence structures and lack diagnostic information. Moreover, the final diagnosis is incorrect, leading to lower LLM-J scores. In contrast, PrivacyRestore generates outputs with a different structure but provides the same, correct diagnosis. As a result, our method achieves slightly lower ROUGE-L scores but significantly higher LLM-J scores compared to $d_\chi$-privacy on privacy spans.

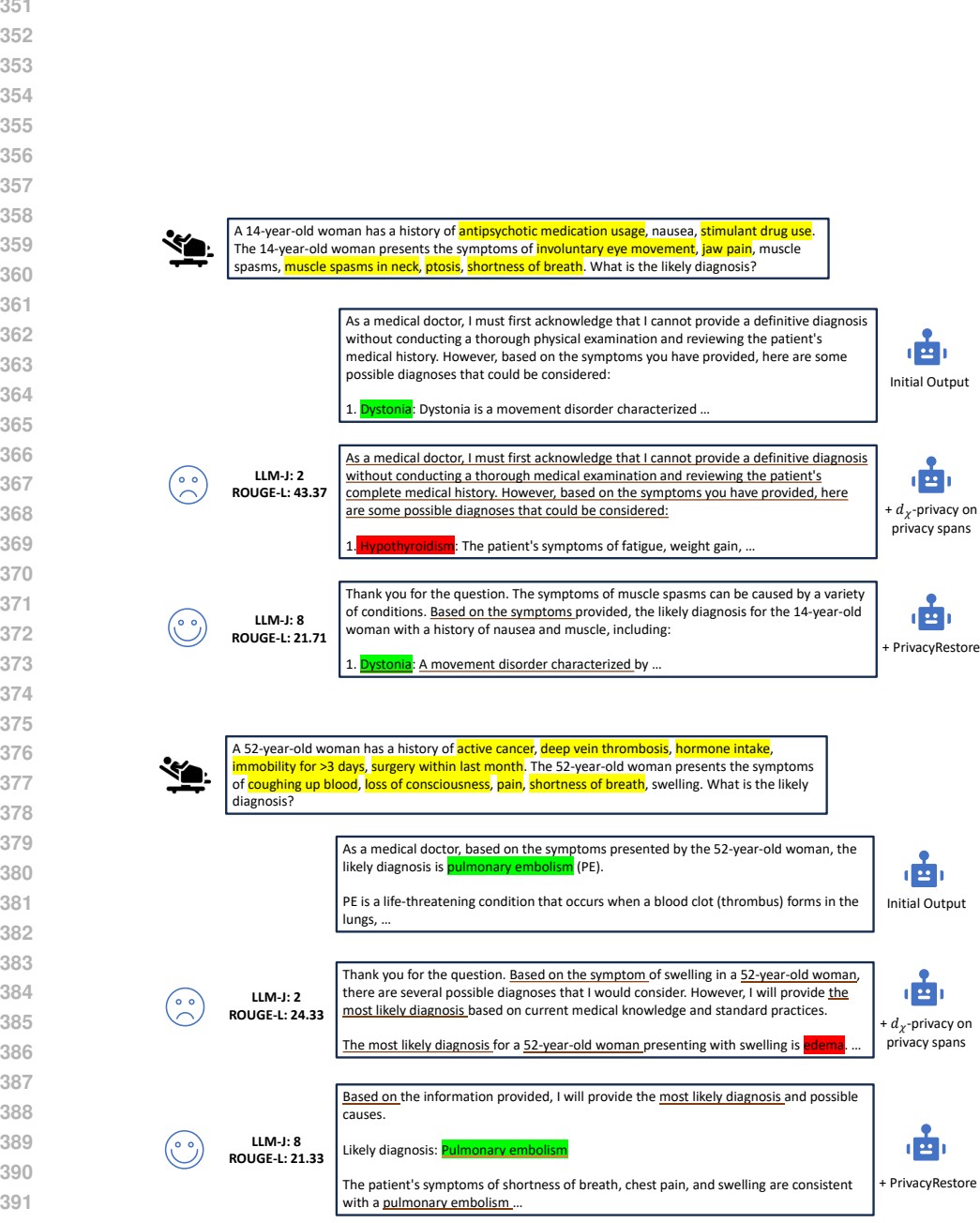

Figure 6: Example Outputs of PrivacyRestore and $d_\chi$ on privacy spans in Pri-DDXPlus dataset. Text highlighted with a yellow background represents the privacy spans in user inputs. Text highlighted with a green background indicates the correct diagnosis. Text highlighted with a red background denotes the incorrect diagnosis. Underscored text marks sections that overlap with the initial output.

## O    PROMPT TEMPLATE DETAILS

### O.1    CLASSIFICATION OF PRIVACY SPANS.

#### O.1.1    MEDICAL DATASETS (PRI-DDXPLUS/PRI-NLICE).

The prompt template is for GPT and used to classify symptoms in Pri-DDXPlus/Pri-NLICE dataset into sensitive and non-sensitive categories. GPT grades the symptoms on a scale of one to five based on sensitivity, with levels greater than three considered private spans in the Pri-DDXPlus/Pri-NLICE dataset.

> You are a privacy expert with extensive medical knowledge. Given a potential symptom or antecedent, please categorize symptom or antecedent into five grades based on the level of privacy they entail.
>
> Utilize the classification guidelines provided below to categorize the symptom or antecedent:
>   Degrade 1: public information, symptoms that are common, widely known, and do not reveal any personal or sensitive information. Examples include sneezing, headache, or minor injuries.
>   Degrade 2: non-sensitive personal information, symptoms that may be personal but not necessarily sensitive or revealing. These may include common illnesses like cold or flu, allergies, or minor digestive issues.
>   Degrade 3: potentially sensitive information, symptoms that could be indicative of underlying health conditions but are not immediately sensitive or stigmatizing. Examples include chronic conditions like diabetes, hypertension, or asthma.
>   Degrade 4: sensitive personal information, symptoms that may be stigmatizing or have social implications if disclosed publicly. This could include mental health issues like depression or anxiety, reproductive health concerns, or substance abuse.
>   Degrade 5: highly sensitive information, symptoms that are highly personal, stigmatizing, or potentially life-altering if disclosed publicly. This category includes sexually transmitted infections, HIV/AIDS, certain types of cancer, or rare and serious medical condition.
> Output your evaluation in the following format:
>
> #thereason: your analysis here [note: this should be step-by-step analysis following the steps: (a) Give an overview of the given symptom or antecedent. (b) Evaluate the degree of the given symptom or antecedent according to classification guidelines.]
> #thescore: your degrade here. [note: provide your degrade in based on your analysis and the classification criteria. Please note that only the degrade is required to be written in here]
> Symptom: **[INSERT YOUR SYMPTOMS HERE].**

#### O.1.2    LEGAL DATASET (PRI-SLJA).

The prompt template is for GPT and used to classify the case details in Pri-SLJA dataset into sensitive and non-sensitive categories. GPT grades the symptoms on a scale of one to five based on sensitivity, with levels greater than three considered private spans in the Pri-SLJA dataset.

> You are a privacy expert with extensive legal knowledge, please categorize the given case element into five levels based on the degree of privacy involved.
>
> Utilize the classification guidelines provided below to categorize the case element:
>   Degrade 1: public legal information: Information that is publicly available and does not reveal personal or sensitive legal details. Examples include general legal concepts, publicly available case laws, or minor regulatory matters.
>   Degrade 2: non-sensitive personal legal information: Legal information that may involve a specific individual but is not particularly sensitive or revealing. Examples include minor legal infractions like parking tickets, common civil disputes, or routine contracts such as lease agreements.
>   Degrade 3: potentially sensitive legal information: Legal information that could imply more personal aspects but is not immediately sensitive or damaging. Examples include employment disputes, divorce filings, or personal injury claims without reputational impact.
>   Degrade 4: sensitive personal legal information: Information that could be stigmatizing or have significant social, financial, or reputational consequences. Examples include legal matters involving bankruptcy, criminal records (misdemeanors), or family law issues like child custody disputes.
>   Degrade 5: highly sensitive legal information: Legal details that are highly personal, stigmatizing, or potentially life-altering if disclosed. This includes serious criminal cases (e.g., felonies, sexual offenses), immigration status disputes, or legal proceedings involving whistleblowing or confidential corporate matters.
> Output your evaluation in the following format:
> #thereason: your analysis here [note: this should be step-by-step analysis following the steps: (a) Give an overview of the given case element. (b) Evaluate the degree of the given case element according to classification guidelines.]
> #thescore: your degrade here. [note: provide your degrade in based on your analysis and the classification criteria. Please note that only the degrade is required to be written in here]
> Case element: **[INSERT CASE ELEMENTS HERE].**

## O.2 GENERATION PROMPTS.

### O.2.1 MEDICAL DATASETS (PRI-DDXPLUS/PRI-NLICE).

The prompt template is for Llama model and used during model generation for the Pri-DDXPlus/Pri-NLICE datasets.

> [INST] <<SYS>>
> You are a medical doctor answering real-world medical entrance exam questions. Based on your understanding of basic and clinical science, medical knowledge, and mechanisms underlying health, disease, patient care, and modes of therapy, answer the following diagnosis question. Base your answer on the current and standard practices referenced in medical guidelines.
> <</SYS>>
>
> **[INSERT USER INPUT HERE]** [/INST]

### O.2.2 LEGAL DATASET (PRI-SLJA).

The prompt template is for Llama model and used during model generation for the Pri-SLJA datasets.

> [INST] <<SYS>>
> You are the world's best legal judgment machine, tasked with rendering a verdict in a legal case. Below are four paragraphs describing the case, though some information may be missing.
> Subject refers to the individual with criminal responsibility who committed the crime;
> Subjective Element consists of intention and negligence, representing the psychological state of the subject concerning the criminal behavior and its outcomes;
> Object refers to the social relationships protected by criminal law that have been infringed upon by the criminal acts;
> Objective Elements include the specific details of the crime, such as time, location, actions, and consequences.
> Using these four components, generate a judgment to sentence the subject of the crime.
> <</SYS>>
>
> **[INSERT USER INPUT HERE]** [/INST]

## O.3 EVALUATION OF GENERATED OUTPUT.

### O.3.1 MEDICAL DATASETS (PRI-DDXPLUS/PRI-NLICE).

The prompt template is for GPT and evaluates the quality of generated output based on relevance, clarity, and accuracy, for Pri-DDXPlus/Pri-NLICE dataset. Scores range from 1 to 10, with higher values indicating better output.

> You are a neutral legal evaluator. You will be provided with a legal question containing the description of a case and the AI assistant's judgment of the case.
> Your task is to assess whether the AI assistant's judgment is useful and accurately points out the crime. Consider factors such as the relevance, clarity, and accuracy of the judgment.
> Rate the response on a scale from 1 to 10, where 1 indicates poor quality and 10 indicates excellent quality.
> Start your answer with the rating score by strictly following the format: `[[rating]]`. After assigning a rating, provide a brief explanation for your score, justifying your evaluation.
> [User Question]
> **[INSERT QUESTION HERE].**
>
> [The Start of Assistant Response]
> **[INSERT GENERATED OUTPUT HERE].**
> [The End of Assistant Response]
>
> Rating:

### O.3.2 LEGAL DATASET (PRI-SLJA).

The prompt template is for GPT and evaluates the quality of generated output based on relevance, clarity, and accuracy, for Pri-SLJA dataset. Scores range from 1 to 10, with higher values indicating better output.

> You are a neutral medical evaluator. You will be provided with a medical question from a patient and the AI assistant's response to the question. Your task is to assess whether the AI assistant's response is useful and accurately addresses the diagnosis. Consider factors such as the relevance, clarity, and accuracy of the information provided.
> Rate the response on a scale from 1 to 10, where 1 indicates poor quality and 10 indicates excellent quality.
> Start your answer with the rating score by strictly following the format: `[[rating]]`. After assigning a rating, provide a brief explanation for your score, justifying your evaluation."
> [User Question]
> **[INSERT QUESTION HERE].**
>
> [The Start of Assistant Response]
> **[INSERT GENERATED OUTPUT HERE].**
> [The End of Assistant Response]
>
> Rating:

## O.4 PROMPT INJECTION ATTACK.

The prompt template is for Llama model and used to carry out a prompt injection attack, translating the garbled text back into the original text.

> INST]<<SYS>>
> You are a text generating machine. Given input with garbled characters, output normal text without garbled characters. ONLY generate translated normal text.
> <</SYS>>
> **[INSERT USER INPUT HERE]**
>
> Translated normal text:[/INST]

