# OpenReview forum: "PrivacyRestore: Privacy-Preserving Inference in Large Language Models via Privacy Removal and Restoration"
_ICLR.cc/2025/Conference — ICLR 2025 Conference Withdrawn Submission_

### Official Review · Reviewer_iVJz · 2024-11-01

**Soundness:** 2
**Presentation:** 2
**Contribution:** 3
**Rating:** 5
**Confidence:** 3

**Summary:**

This work operates in a setting where the user wants to off-load inference computation to a server while preserve the privacy of its input data. The authors propose a framework that trains restoration vectors during the preparation stage, then have the clients transmit meta vectors with masked inputs to the server, which will be used for inference by the server.

**Strengths:**

1. The paper focuses on an important problem of privacy-preserving inference of large language models
2. I believe having a hybrid privacy setting where the input is a mixture of private and public information is more practical. In particular, the idea of a privacy span is interesting.

**Weaknesses:**

1. There are a lot of moving parts in the framework that make it difficult to understand and follow. Rather than only elaborating on the technical aspects of each step, the authors should also discuss a running example, similar to the example user input used in Figure 1, to help give more intuition about what each step is doing.
2. It would be beneficial if the authors could provide a background section on the attention mechanism and activation steering methods prior to introducing the methodology.
3. In the preparation stage, the server supposedly has access to user inputs in the training set. But during inference, the framework is providing privacy to another set of user inputs? So the authors are not concerned about the privacy of this user input training set?
4. The assumptions needed for the textual data, which are stated in the introduction, are a bit strong. For example, in Figure 1, the user input in the training set is nearly lexically identical compared to the inference user inputs. So, do we also have to assume that the server has access to this training set? All of these assumptions may severely limit the scope of the work to a small set of datasets.
5. Additionally, the client needs a local lightweight model for the inference stage. This feels like it defeats the motivation of this work: users offload ML computations to a server.
6. Due to weaknesses (3), (4), and (5), I’m unsure what real-world application this framework falls under. Especially with (5), if a client needs to perform ML computations, why would one use this framework over federated learning? The client can just run inference locally with their lightweight model trained via Federated Learning + client-level DP-SGD, which achieves LDP–a stronger version of dx-privacy.

**Questions:**

See weakness (2), (4), and (6) for questions.

---

> ### Author Response · Authors · 2024-11-20
> **Authors' Response (1/2)**
>
> We sincerely thank you for your feedback and suggestions! We address all your concerns below:
>
> **To be mentioned, we have updated our submitted paper, please download the latest one, thank you.**
>
> >W1. the authors should also discuss a running example.
> >W2. Provide a background section on the attention mechanism and activation steering methods prior to introducing the methodology.
>
> We have included some running examples result in Appendix N and we will also add more explations for the present examples.
> Due the space limit, we do not include the background section in our submitted paper. We have included the background section in Appendix C in our new updated version.
>
>
> >W3. In the preparation stage, the server supposedly has access to user inputs in the training set. But during inference, the framework is providing privacy to another set of user inputs? So the authors are not concerned about the privacy of this user input training set?
>
> Firstly, protecting the privacy of the training set is not our main focus and our method is mainly used to protect user privacy during inference.
> Secondly, and more importantly, **the training set used in our framework does not raise any privacy concerns**.
> Your concerns about privacy protection is possibly because our approach includes an offline training process and does not apply privacy protection for the training set.
> However, this concern is redundant, as the key point you overlooked is that the training sets we need to use—DDXplus[1], NLICE[2], and SLJA[3]—are all public datasets and pose no privacy risks.
> **Our training process does not require private datasets; public datasets are sufficient**.
> These public datasets all have underwent a deprivatization process before being released.
> Many LLMs have included these datasets during the pretraining/Instruct tuning phase.
>
> [1] Tchango, et al. "Ddxplus: A new dataset for automatic medical diagnosis". NeurIPS 2022.
> [2] Al-Ars, et al. "NLICE: Synthetic medical record generation for effective primary healthcare differential diagnosis". BIBE 2023.
> [3] Deng, et al. "Syllogistic reasoning for legal judgment analysis". EMNLP 2023.
>
>
> >W4. The assumptions needed for the textual data are a bit strong. For example, in Figure 1, the user input in the training set is nearly lexically identical compared to the inference user inputs ...
>
> You may misunderstand the example we given in Figure 1.
> We present the use input example during inference whose format is lexically similar to the public training dataset for better understanding.
> **However, the format of user input in our method is not restricted to any specific format**.
> It is important to emphasize that we propose a realistic setting where the user determines their own privacy by themselves, removes sensitive information from the raw input, and formats the input accordingly.
> The setting is meaningful and satisfies the requirement "information self-determination right" [1] and this setting is widely advocated by the public [2,3].
> However, **if the user can not determine privacy by themselves and the format of input during inference is varying, our method still can work well by be combined with the "text sanitizaiton" technique** [4, 5].
> The text sanitization technique can automatically identify the privacy spans from the varying-format inputs and our method can be built up on the identification results.
> Our method is orthogonal to these text sanitizaiton techniques.
> We also have conducted the experiments to evaluate the effectiveness of our method to address the situation when the user inputs are varying format, as shown in Q3 of General Response.
> The experimental results in Q3 of General Response demonstrate that our method is effective in handling real-world scenarios where user input varies in format.
>
> [1] Mattern, J., et al. "The limits of word-level differential privacy". NAACL 2022.
> [2] Utpala, S., et al. "Locally differentially private document generation using zero-shot prompting". EMNLP 2023.
> [3] McSherry, F., et al. "Mechanism design via differential privacy". FOCS 2007.
> [4] Chen, Y., et al. "Hide and seek (HAS): A lightweight framework for prompt privacy protection". arXiv 2023.
> [5] Kan, Z., et al. "Protecting user privacy in remote conversational systems: A privacy-preserving framework based on text sanitization". arXiv 2023.

---

> ### Author Response · Authors · 2024-11-20
> **Authors' Response (2/2)**
>
> >W5. Additionally, the client needs a local lightweight model for the inference stage. This feels like it defeats the motivation of this work: users offload ML computations to a server.
>
> You have misunderstood our motivation, which is to enable users to offload **most** ML computations to a server while **retaining a small portion (1%, nearly negligible) locally to ensure privacy protection**.
> Our proposed framework employs a highly lightweight model on the client side, **comprising approximately 1% of the total size of the server-side model**.
> The little computaions can bring significant performance improvement and privacy protection.
> All other methods to protect user privacy are always introduced computations in the client.
> For example, SMPC-based methods [1,2] all need to require encryption and decryption computations in the client and Encryption and decryption computations are more time-consuming and resource-intensive compared to our method.
> Paraphrase methods [3,4] need to deploy a paraphrase model in the client to paraphrase the user inputs before transmission.
> Indeed, retaining a small portion of ML computation is both acceptable and essential, and it provides significant privacy protection benefits.
>
> [1] Liang, Z., et al. "Merge: Fast private text generation". AAAI 2024.
> [2] Li, D., et al. "MPCFormer: Fast, performant and private transformer inference with MPC". ICLR 2024.
> [3] Mattern, J., et al. "The limits of word level differential privacy". NAACL 2022.
> [4] Utpala, S., et al. "Locally differentially private document generation using zero shot prompting". EMNLP 2023.
>
>
>
> >W6. If a client needs to perform ML computations, why would one use this framework over federated learning? The client can just run inference locally with their lightweight model trained via Federated Learning + client-level DP-SGD, which achieves LDP–a stronger version of dx-privacy.
>
> Firstly, all ML computations occurring on the client are just inference using a lightweight Bert model,
> which is used to capturing the importance of each privacy span. **The inference cost for this lightweight Bert model on the client side is negligible**.
> Secondly, if we follow the federated learning framework as you mentioned in the review, the server should train a lightweight generative model, such as GPT2-medium, and send the model to the client for local inference.
> However, the results are very poor. The reason for this is that **the capacity of the lightweight model (e.g. GPT2-small) for reasoning and inference is limited compared to larger LLMs** (e.g., Llama-7b-chat).
> Therefore, in our method, we use a large LLM on the server (Llama-7b-chat) for inference while the lightweight model on the client (Bert-base) is only used to capture the importance of each privacy span.
> This approach fully leverages the strong reasoning and inference capabilities of the LLM and also achieves the goal of privacy protection.
> For comparison, we also conduct additional experiments in which we train a lightweight model (GPT2-small) for local inference and evaluate its performance, as shown in the following table.
> (Additionally, we choose GPT2-small because its parameter count of 124M is comparable to that of BERT, which has 110M parameters. Both models can be considered lightweight due to their relatively small sizes.)
>
> | |MC1|MC2|RL|LL-J
> |-|-|-|-|-|
> Train GPT-2 and infer locally |45.19 $_{\pm0.13}$|45.23 $_{\pm0.09}$|25.98 $_{\pm0.32}$|2.90 $_{\pm0.02}$
> PrivacyRestore |62.97 $_{\pm0.00} (\uparrow 17.78)$|60.19 $_{\pm0.00} (\uparrow 14.96)$|27.24 $_{\pm0.26} (\uparrow 1.26)$|4.47 $_{\pm0.00} (\uparrow 1.57)$
>
> ---
> We deeply appreciate the reviewer’s efforts.
>
> **We are looking forward to your reply. If you have any further questions, we are happy to dicuss them!**

---

> ### Author Response · Authors · 2024-11-26
> **Kind Remind**
>
> Dear reviewer: This is a polite reminder that we are awaiting your feedback on our rebuttal. To assist you, we have highlighted the key points for your convenience. We would greatly appreciate it if you could discuss them with us at your earliest convenience.

---

> > ### Comment · Reviewer_iVJz · 2024-11-27
> > **Reponse by Reviewer iVJz**
> >
> > I thank the authors for their response.
> >
> > > our approach includes an offline training process and does not apply privacy protection for the training set… is that the training sets we need to use—DDXplus[1], NLICE[2], and SLJA[3]—are all public datasets and pose no privacy risks.
> >
> > So I think this raises my other concern that the assumptions needed for the textual data are a bit strong. So, in your example, you use DDXplus for the offline training process, but then you use your curated privacy-preserving dataset Pri-DDXPlus for inference. If that’s the case, then I feel like there’s a huge assumption that the offline training set needs to be lexically similar to the inference data in order for this to work well. Please correct me if I’m misunderstanding this. If you have experimental results that show the performance is strong even when the lexical formatting of the data is very different between the offline training and inference datasets, then please let me know.

---

> > > ### Author Response · Authors · 2024-11-28
> > > **Further Authors' Response (1/2)**
> > >
> > > We sincerely thank you for your timely response and your efforts in reviewing our paper.
> > > Below, we have addressed your additional concern by adding more experiments and explanations:
> > > > ... If that’s the case, then I feel like there’s a huge assumption that the offline training set needs to be lexically similar to the inference data in order for this to work well ...
> > > If you have experimental results that show the performance is strong even when the lexical formatting of the data is very different between the offline training and inference datasets, then please let me know.
> > >
> > > Thank you for your thoughtful consideration. However, **our method is indeed robust, even when the lexical formatting of the test data differs significantly from the offline training set.**
> > > Firstly, the reason why the training set is lexically similar to the test set is just that both **the training set and test set are from the same source dataset** (DDXPlus, NLICE and SLJA datasets).
> > > Nevertheless, in fact, **our method does not require the test set (inference data) to be lexically similar to the offline training set**.
> > > We have evaluated our methods when given different lexical format inference data as follows:
> > > Here, we consider two scenarios based on whether users have the ability to determine privacy spans by themselves, a concept known as 'Self-determination' [1,2,3].
> > >
> > > **1. When the users can determine privacy by themselves:**
> > > As we explained in the General Q3 of General Response,
> > > the self-determination of the users indicate that they can determine the privacy by themselves and find the closest one in the pre-defined privacy spans set to construct the meta vector.
> > > This is also the main setting of our paper.
> > > In this situation, the lexical expression of privacy spans (such as symptoms or diseases) is limited and restricted to the predefined set.
> > > Then the lexical difference mainly refers to the format of user questions, which can also be named as "question template".
> > > In order to make the test set lexically different to the training set, **we create some question templates that differ from the template used in the initial Pri-DDXPlus training set**, as follows:
> > > ```
> > > [Initial Question Template of Pri-DDXPlus Train set]:
> > > A {fake_age}-year-old {fake_sex} has a history of {medical_history}. The {age}-year-old {sex_text} presents the symptoms of {medical_symptom}. What is the likely diagnosis?
> > >
> > > [Question Template Variant 1 of Pri-DDXPlus Test set]:
> > > The patient is a {fake_age}-year-old individual of {fake_sex}, with a background of {medical_history}, is showing symptoms of {medical_symptom}. What might be the diagnosis?
> > >
> > > [Question Template Variant 2 of Pri-DDXPlus Test set]:
> > > Recently, I've been experiencing symptoms such as {medical_symptom}, which have been quite uncomfortable. I also have a medical history of {medical_history}. I’m {fake_age}-years-old and identify as {fake_sex}. Based on this, what condition do you think I might have?
> > >
> > > [Question Template Variant 3 of Pri-DDXPlus Test set]:
> > > Lately, I've been dealing with symptoms like {medical_symptom}, which have been quite troubling. I'm a {fake_age}-year-old {fake_sex}. Additionally, I have a history of {medical_history}. Given this information, what condition do you think I may have?
> > > ```
> > > It is worth mentioning that Template Variant 1 is the question template which **is described in the third-person perspective**, which is the same as the Initial Question Template of training set.
> > > Template Variant 2 and Template Variant 3 are **described from the first-person perspective**, which **is more common when the users use the LLM services**.
> > > We use the three question template variants above to transform the test set and evaluate our method.
> > > The following experimental results demonstrate that our method still works well, even when the lexical format of the test set differs from that of the offline training set:
> > >
> > > |Pri-DDXPlus |MC1 |MC2 |ROUGE-L |LLM-J|
> > > |--|--|--|--|--|
> > > |No Restoraion (lower bound) | 33.57 | 32.49 | 25.19 | 3.21 |
> > > |PrivacyRestore Using the same question template as the train set | 62.97 $(\uparrow 29.40)$ | 60.19 $(\uparrow 27.70)$ | 27.24 $(\uparrow 2.05)$ | 4.47 $(\uparrow 1.26)$ |
> > > |PrivacyRestore Using Question Template Variant 1 | 59.91 $(\uparrow 26.34)$ | 57.32 $(\uparrow 24.83)$ | 26.30 $(\uparrow 1.11)$ | 4.50 $(\uparrow 1.29)$ |
> > > |PrivacyRestore Using Question Template Variant 2 | 63.07 $(\uparrow 29.50)$ | 59.77 $(\uparrow 27.28)$ | 27.01 $(\uparrow 1.82)$ | 4.91 $(\uparrow 1.70)$ |
> > > |PrivacyRestore Using Question Template Variant 3 | 64.36 $(\uparrow 30.79)$ | 61.26 $(\uparrow 28.77)$ | 26.59 $(\uparrow 1.40)$ | 4.52 $(\uparrow 1.31)$ |

---

> > > ### Author Response · Authors · 2024-11-28
> > > **Further Authors' Response (2/2)**
> > >
> > > As shown in the table above, even when the question format (including different person perspectives in the expression and different question phrasings) differs significantly from the training set,
> > > **our method still outperforms the No Restoration baseline (lower bound)**.
> > > Although changing the question template leads to a decrease in performance, the decline is minimal, strongly demonstrating that our method is robust to variations in the lexical format of the test set.
> > > Moreover, we observe **a slight performance improvement when using the first-person perspective** in the test set's lexical format (Variants 2 and 3), both in MC1/2 and LLM-J.
> > > This may be attributed to **the LLM's preference for first-person questions**.
> > >
> > >
> > > **2. When the users can not determine privacy by themselves:**
> > > We also consider the scenario where users cannot determine privacy and then **both the question template and the lexical expression of privacy spans can be varied**.
> > > In order to transform the test set with varying question templates and lexical expressions of privacy spans, we **use GPT-4 to paraphrase the initial questions**,
> > > ensuring that the meaning remains unchanged while only the lexical format is modified. Here we provide an example as follows:
> > > ```
> > > [Inital Question Format]:
> > > A {fake_age}-year-old {fake_sex} has a history of {medical_history}. The {age}-year-old {sex_text} presents the symptoms of {medical_symptom}. What is the likely diagnosis?
> > >
> > > [Question before transforming]:
> > > A 10-year-old woman has a history of antipsychotic medication usage, nausea, stimulant drug use. The 10-year-old woman presents the symptoms of involuntary eye movement, jaw pain, muscle spasms, muscle spasms in neck, ptosis, shortness of breath. What is the likely diagnosis?
> > >
> > > [Question after transforming using GPT-4 (Varying in format)]:
> > > I'm a 10-year-old girl and I've been taking antipsychotic medication for a while now. I've been feeling really nauseous, and I also used a stimulant drug recently. Lately, I've been having these strange symptoms like my eyes moving involuntarily, and my jaw hurts a lot. I've also been getting muscle spasms in my neck and elsewhere, and my eyelids droop sometimes. On top of that, I've been feeling short of breath. What could be causing all of these symptoms?
> > > ```
> > > And then we evaluate our method on the transformed test set and the experiment results of our method are as follows:
> > > |Pri-DDXPlus |MC1 |MC2 |ROUGE-L |LLM-J|
> > > |--|--|--|--|--|
> > > |No Restoraion (lower bound) | 33.57 | 32.49 | 25.19 | 3.21 |
> > > |PrivacyRestore Using the same question template as the train set | 62.97 $(\uparrow 29.40)$ | 60.19 $(\uparrow 27.70)$ | 27.24 $(\uparrow 2.05)$ | 4.47 $(\uparrow 1.26)$ |
> > > |PrivacyRestore After question transforming using GPT-4 | 62.87 $(\uparrow 29.30)$ | 59.97 $(\uparrow 27.48)$ | 26.27 $(\uparrow 1.08)$ | 4.28 $(\uparrow 1.07)$ |
> > >
> > > The results above show that **our method still performs effectively when the test set, transformed by the GPT-4 model, is lexically very different from the offline training set.**
> > > The improvement over the No Restoration baseline is substantial.
> > > For more implementation details, please refer to General Q3 in the General Response.
> > >
> > > ---
> > >
> > > We sincerely appreciate your efforts and your timely reply to us.
> > >
> > > **We very look forward to your further feedback on our paper. If you have any additional concerns, we would be happy to discuss them!**

---

> > > > ### Comment · Reviewer_iVJz · 2024-11-29
> > > > **Response by Revewier iVJz**
> > > >
> > > > I thank the authors for their response. They have addressed some of my concerns, and I have raised my score accordingly. The framework introduced by the authors has a lot of interesting parts, but it's also challenging to explain the framework clearly. In its current state, the paper still needs some work for a clearer presentation, which I know is hard to do with only 10 pages for the main paper.

---

> ### Author Response · Authors · 2024-11-30
> **Further Authors' Response (1/1)**
>
> We sincerely thank you for your timely response and your efforts in reviewing our paper.
> It is truly inspiring to see you raise your score!
>
> Firstly, we **promise to incorporating all rebuttal content into the paper and including any material that cannot be accommodated within the 10-page limit in the appendix**.
>
> Secondly, as you mentioned, we only have addressed some of the concerns, and **we are very eager to hear about any other remaining issues**.
> We will try our best to conduct additional experiments and provide further explanations to address these remaining concerns during the remaining time of the rebuttal period.
> If we can resolve most of your concerns, we very hope to earn your positive score (maybe marginally above the threshold), which would be very inspiring and motivating for us.
>
> ---
> Anyway, we really appreciate your effort and timely feedback which is very helpful for us.
> We sincerely thank you once again and we are deeply grateful.
> **We very look forward to your further feedback on our paper or any other concerns. If you have additional questions, we would be happy to discuss them and try our best to solve your remaining concerns.**
> Thank you very much once again!

---

### Official Review · Reviewer_MqBb · 2024-11-03

**Soundness:** 3
**Presentation:** 2
**Contribution:** 3
**Rating:** 5
**Confidence:** 4

**Summary:**

The paper proposes a method ('PrivacyRestore') to protect private information in user inputs that are sent to a cloud LLM service for generating responses during inference.

As preparation, their solution first identifies task-specific contiguous sections, so-called "privacy spans", that represent privacy-sensitive information.
Next, for each privacy span, the attention heads in the LLM are identified which are most affected when that privacy span is removed (redacted) from the input texts. Based on this, a subset of the common top K most influenced attention heads is selected across all identified privacy spans.
Now the server trains so-called "restoration vectors" for each privacy span and each of the top K attention heads, which can be used to correct the output of the attentions head when the privacy span is redacted in the input text.
The restoration vectors are then sent to all clients.

During inference, if a client wants to use the LLM service, it first determines all privacy spans in its input text and computes a normalized, weighted sum of the corresponding restoration vectors for each of the top K attention heads. The weights are determined using a separate, light-weight proxy model (BERT) that is run locally on the input text. For additional privacy protection, the summed vectors are perturbed with Gaussian noise to achieve d_X-privacy, a generalization of differential privacy.
The client the redacts the privacy spans from the input text and sends the redacted text together with the noisy, so-called meta vectors, to the server.
The server runs the redacted input text through the LLM while using the meta vectors to correct the outputs of the affected attention heads, and sends the generated text back to the user.

The authors derive three datasets for conducting privacy-relevant evaluations. They propose a variety of experiments involving suitable baselines and interesting attacks.
However, some theoretical issues and questions remain that in my opinion must be clarified, which is why the current results should be taken with a grain of salt.

**Strengths:**

Comparisons against d_X-privacy token-perturbation methods on the entire input or only privacy spans, and against paraphrasing, provides interesting baselines in the experiments.

I also like the proposed attacks for the experiments on empirical privacy protection. The proposed prompt injection attack to output the original text is a good additional idea to check how well sensitive information is actually protected. On the other hand, it seems the inversion attacks could be strengthened (cf. weaknesses).

Overall, I think the approach could work in practice, albeit with the limitation of how well sensitive information can be covered through a limited set of privacy spans that must be defined in advance. (cf. weaknesses: typos in names of diseases, etc.).
More importantly, given the issues in the Theorems 5.1 and particularly 5.2, I am not confident that the experiment results use the correct privacy budget for the proposed method.

**Weaknesses:**

Theorem 5.1:

- What is the context/what assumptions do you make how DP methods are utilized here? The way you formulate Theorem 5.1, it reads as if it applies to any LDP/CDP/d_X privacy mechanism. However, that makes no sense since DP can be applied in other use cases apart from protecting text, and even for text, it depends on how the DP mechanism is applied to a text to make conclusions about the required privacy budget. So you should specify the actual scope to which your Theorem 5.1 applies to.


Theorem 5.2:

- Since you transmit meta vectors R_h=Z_h+noise for each affected/'edited' attention head, shouldn't the privacy budget also scale with the number of affected attention heads |H_c| ? Note that this also would affect your experiment results: You state in L397-398 that there are 175 or 125 affected attention heads, which would significantly increase your total required privacy budget!
- Can you provide an upper bound for the ||Z_h'-Z_h||? If they are normalized unit vectors and the norm is the L2 norm, 2 might work?

L71-L73: What happens if there is a typo in name of disease or symptom? What if new symptoms are discovered for a certain disease? What if a disease or symptom is paraphrased in an unexpected way? It seems, such examples cannot easily be covered by privacy spans and hence often remain unprotected, so the assumption about limited number of required privacy spans appears overly optimistic to me.

Restoration vectors are determined on target LLM (Llama2) per attention head, but the associated weights to combine these restoration vectors into the meta vector are determined from another proxy model (BERT) with _different_ attention heads.
How does this fit together (e.g., due to possible deviations in these models' attention scores), especially since the weights are averages across the proxy model's attention heads, but then still applied individually to the per-head restoration vectors?

**Questions:**

Please introduce or explain special terms and concepts *before* using them:
- L019: What is a "privacy span"? (Yes, it is in the next sentence, but these tiny details make the paper hard to read.) Can you reorder/rephrase to describe this concept first?
- L025: You talk about "linear growth of the privacy budget", which seems to imply that some form of differential privacy is employed. However, you never mentioned differential privacy as part of your solution until that point.
- L213: What is "*the* Top-K Heads Selector"? This only becomes clearer after reading the paragraph. Instead, maybe start the section explaining that you subsequently "will introduce *a* Top-K Heads Selector whose aim is to ...".
- L269: "\lambda represents *the* coefficient" <- Which coefficient? What does it do? Where did you introduce it?

L068-L070: What do you want to say here? "if privacy spans [...] are removed [...] attackers are able to recover private information" *despite* having removed the privacy spans? Or is this an (intentionally bad) example where the spans are too short (covering one word only)? In this case, it may help to explicitly distinguish proper vs. degenerate spans (longer contiguous sequences vs. single words).

L205-210: I find the description of 'probes' a bit too complex. Do we need the formula and knowledge of its internals like the parameters? Or could this be stated more simply as "a probe is a binary classifier consisting of a dense layer with a sigmoid activation" (or similar)?

L253: (typo) "server" side

L269: In "a is the initial output give[n?] the complete input", what output do you mean? Output of some attention head, some other part of the LLM, the entire LLM (softmax probabilities over the token space)?

L252-272: (Related to the previous question.) Where and how would the restoration vectors be applied to actually restore something (if no meta vector was produced)? You just described how to compute/train them, but not how and where they would be applied.

L294 (and elsewhere): You talk about "meta vector R_h on head h", so there is not just one meta vector, but several meta vectors, one for each affected/'edited' attention head h \in H_c? Then I would assume that not just one, but all of these meta vectors are computed by the client and sent to the server. Elsewhere (e.g., L241, L277, and others), you often refer to 'the' or 'a' meta vector in singular form. Note that this is important also for determining the overall privacy budget, cf. weakness on Theorem 5.2.

L345-355: You state the number of "_types_ of privacy spans".
- What is the meaning of a 'type'? Can you give one or two examples?
- How many actual privacy spans (not 'types') did you identify for each dataset?
- What is the impact of the number of spans on the computational overhead? As I understand the approach, the overhead grows with the number of privacy spans, since inference requires clients to compute attention weights by running a separate proxy model (BERT) for each possible privacy span.

Minor question: Why do you name the affected attention heads "edited" instead of just "affected attention heads"? As I understand the paper, the heads that are most affected by the removal of the privacy spans are left unmodified (i.e., *no* editing is done to them), and by applying the meta vector you only modify the _output_ of those attention heads. (Besides, you already have another name for them, e.g., you write "common top-K heads set H_c" in L316.)

It may help to expand preliminaries (could add into related work section) on concepts that are critical to the approach proposed in the paper, to make the paper more self-contained and to make it easier for readers that do not yet know the cited papers where these preliminary concepts have been introduced:
- Add preliminaries on _inference steering_, including restoration vectors.
- Might help to recap attention mechanism/heads (since they are a critical component used for the probes and activation steering).
- Alternative: Could split "Related Work" (what other approaches that you are *not* building upon exist, such as SMPC, but also d_X privacy perturbations or paraphrasing that you use as baseline) and add separate "Preliminaries" (background on building blocks that are critical for your solution, like DP, inference steering, attention mechanism details) section. (I am aware of space constraints; one idea is that you could shorten elsewhere, e.g., you write rather frequently, in bold, what privacy spans are.)

Section 6.3:
- You use an embedding inversion attack, but I didn't fully understand how exactly you implement it (even after checking Appendix L). My guess is: The attacker tries (using a finetuned GPT-2) to predict the original text (i.e., the privacy spans) given the meta vectors? But then a stronger attack model (that I still think is plausible) could provide the attacker some context around the privacy spans, e.g., the incomplete queries, in addition to the meta vectors. Then the attacker could train a model to predict the privacy span given the context (infilling/masked language modeling) *and* the noisy meta vectors, which could narrow down the predictions to more suitable ones.
- In the "different d_X-privacy Percentage" experiments, you vary the percentage of protected tokens, but I'd assume that you leave the overall privacy budget the same? For clarity, I suggest stating this assumption explicitly.
- The blue curve (Pri-NLICE) in Fig 3.(c) seems to go up linearly and almost double -- which seems to contradict your expectations. Could you explain the observed behavior, and how it deviates from or still is in line with your expectations? Could more protected privacy spans actually lead to more information leakage, as more private information is encoded in the meta vectors that are used for the inversion attack?

---

> ### Author Response · Authors · 2024-11-20
> **Authors' Response (1/6)**
>
> We sincerely thank you for your feedback and suggestions! We address all your concerns below:
>
> **To be mentioned, we have updated our submitted paper, please download the latest one, thank you.**
>
> >W1. Theorem 5.1: What is the context/what assumptions do you make how DP methods are utilized in Theorem 5.1?
>
> Actually, we have pointed out Theorem 5.1 is applicable to when applying any LDP/CDP/$d_\chi
> $-privacy mechanism to protecting text in Line 324 and the theorem itself.
> Theorem 5.1 itself is ended by "the protected text".
> Even so, more emphasis on the context of protecting text will be added in the expression of Theorem 5.1.
> To our best knowledge, the mainstream way of applying these DP methods to protect text is adding noise to the token embeddings and converting the noisy embeddings back to tokens, as proposed by [1].
> And this paradigm is adopted in all following works, [2,3,4,5]. The way of applying DP to text in Theorem 5.1 is the default and mainstream paradigm.
> We have not investigated other paradigms to apply DP to text.
> I guess that you may errorly regrad applying DP-SGD during language model training as one way to apply DP to text.
> In fact, DP-SGD applies DP to the gradients, not the text, and aims to prevent the model from memorizing the training data during training process rather than protect text inputs during inference.
> We would appreciate it if you could provide relevant works that propose new paradigms for applying DP to text, so that we can refine the definition of Theorem 5.1.
>
> [1] Feyisetan, O., et al. "Leveraging hierarchical representations for preserving privacy and utility in text". ICDM 2019.
> [2] Xu, Z., et al. "A differentially private text perturbation method using regularized Mahalanobis metric". Proceedings of the Second Workshop on Privacy in NLP, 2020.
> [3] Qu, C., et al. "Natural language understanding with privacy-preserving BERT". CIKM 2021.
> [4] Mattern, J., et al. "The limits of word-level differential privacy". NAACL 2022.
> [5] Li, Y., et al. "Privacy-preserving prompt tuning for large language model services".
>
> >W2.1. Theorem 5.2: Shouldn't the privacy budget also scale with the number of affected attention heads |H_c|?
>
> Thank you for your careful inspection; However, there may be some misunderstanding regarding how noise is added.
> In our presentation on constructing the meta vector (Lines 299-303), we simplified the explanation by considering only one edited head to make it easier to understand.
> As shown in the equation $\mathcal{R}_h=Z_h+\mathcal{N}$, we add noise to the normalized weighted sum of the restoration vectors only for that single head $h$.
> **In fact, when multiple attention heads need to be edited, we firstly concatenate the weighted sums of these heads into a single vector.**
> Then, the subsequent steps are the same as for a single edited head: we apply normalization and add noise to the concatenated vector to obtain the meta vector.
> Therefore, after concatenation, the client transmits a single meta vector, whether for one edited head or multiple heads.
> This is also why we always use 'the' or 'a' meta vector in singular form (e.g., L241, L277, and others). We will include this explanation in our paper to reduce misunderstandings.
> As for the question of whether the privacy budget scales with the number of edited attention heads $|H_c|$, the answer is definitively not.
> **The reason is that, after concatenation, the client adds noise only once to the concatenated result and transmits a single vector.** The concatenation operation prevents adding noise multiple times and transmitting multiple vectors, which would incur additional privacy costs that scale with $|H_c|$. Therefore, through concatenation, we can decoupling the privacy budget with the number of edited heads $|H_c|$.
> However, although we have introduced concatenation for preventing the growing privacy budget with the number of edited heads, $|H_c|$ can also influence the utility of our methods.
> **The reason is that with more number of edited heads, the concanated meta vector will include more information togather with more noise.**
> For example, when editing one head, only a 128-dimensional noise vector is added.
> But when $|H_c|$ heads are edited, the noise vector's dimension increases to 128 $\times$ $|H_c|$.
> Intuitively, the reason why more noise is needed is that with more edited heads, the client can transform more information to restore the privacy spans.
> To prevent information leakage, more noise is essential. Therefore, the number of edited heads $|H_c|$ causes a trade-off problem because more edited heads will bring more information and more noise.
> So we regard the number of edited heads as a hyperparameter and conduct the hyperparameter analysis on the number of edited heads to pick up the best number, as shown in Appendix I.
> Here we name the number of edited heads as $K(=|H_c|)$.

---

> > ### Comment · Reviewer_MqBb · 2024-11-27
> >
> > W1: Yes, while you state it is related to text, you still do not mention how exactly the DP variants in Thm 5.1 are applied to texts. You could apply it to tokens/words or token/word embeddings, in which case the privacy budget would grow with the text length, but you could also apply those DP variants, e.g., to embeddings of entire texts (cf. e.g., papers "DP-VAE" by Weggenmann et al. or "DP-BART" by Igamberdiev et al.), in which case the privacy budget would be constant and _not_ grow. And there may be other ways, too. So the conditions of Thm. 5.1, as presented in the paper, are still unclear. If you make assumptions ("mainstream way"), state them explicitly. I wasn't referring to DP-SGD.
> >
> > W2.1. That makes more sense. The more precise (correct) formulation that you explain here in the comments should indeed have been included in the paper. While I agree that aiming for a simple explanation is good in general, I would argue that it should still be correct and not confuse the reader by leaving out the important details how the simply-explained idea is extended to the overall idea.

---

> ### Author Response · Authors · 2024-11-20
> **Authors' Response (2/6)**
>
> >W2.2. Theorem 5.2: Can you provide an upper bound for the ||Z_h'-Z_h||?
>
> The theoretical upper bound of the distance $||Z_h^{\prime}-Z_h||$ is obviously 2 for the normalization operation.
> However, in practice, the theoretical upper bound is too loose to be used to estimate the privacy budget.
> Therefore, we plot the distribution of the distance in subfigure (d) of Figure 4 and compute the mean value of distance as the prior estimates.
>
> >W3. L71-L73: What happens if there are typos/ new symptoms/ diseases or symptoms are paraphrased?
>
> It is an interesting question.
> First of all, the main claim of our paper is that **the user can determine what constitutes privacy and select the simliar predefined privacy spans themselves**.
> In that case, no typo/paraphrase problem will happen. Even if new symptoms are discovered, the user can select the most similar predefined symptoms as the substitute.
> We have demonstrated that the actual distribution of privacy spans is long-tailed, which means most privacy spans focus on minority categories, as shown in Q2 in General Reponse.
> And we can easily predefine those mostly frequent privacy spans. If the left privacy spans (symptoms or disease) actually appear, the user can select the similar one from the predefined set as the substitute.
> We have conducted the experiment that if the predefined set can not cover all privacy spans, our method still shows superior performance, as shown in Q2 in General Reponse.
> Then we also further consider the situation when the user can not find privacy by themselves. Since our method is orthogonal to those 'text sanitization' methods [1,2], which are used to detect and remove private information.
> **We can combine our method with these text sanitization methods easily by using the text sanitization** to detect and remove all privacy spans and restorate these privacy spans using our method to ensure the inference utility. We add more experiment results of combining our method with these text sanitizaiton methods to shown that our method still work even user can not fine privacy. The detailed experiment results and more explanation please refer to Q3 in General Reponse.
>
> [1] Chen, Y., et al. "Hide and Seek (HAS): A lightweight framework for prompt privacy protection".
> [2] Kan, Z., et al. "Protecting user privacy in remote conversational systems: A privacy-preserving framework based on text sanitization".
>
> >W4. How does BERT and Llama2 fit together (e.g., due to possible deviations in these models' attention scores), especially since the weights are averages across the proxy model's attention heads, but then still applied individually to the per-head restoration vectors?
>
> Actually, we use the attention weights computed by BERT model to assess the importance of different privacy spans.
> Higher attention weights mean more important privacy spans.
> The question you raised is interesting: whether the attention scores obtained by BERT are similar to those generated by the target LLM (Llama2).
> **Intuitively, the answer is yes, because attention weights are determined by the semantic meaning of the privacy spans.**
> If the privacy spans carry important semantic meaning, the attention weights will consistently be high across these language models (LMs).
> We also conduct additional experiments to validate this by computing the BERT attention weight distribution (averaged across all heads) and the inital Llama2 attention weight distribution (per head) for privacy spans in user queries.
> We then calculate the KL divergence to assess the similarity between these distributions.
> Specifically, for each query, we compute the KL divergence between the averaged BERT attention weights and the per-head Llama2 attention weights.
> Finally, we report the mean KL divergence and the characteristics of the KL divergence distribution.
>
> |            | Mean value | Quantile 25% | Quantile 50% | Quantile 75% |
> |------------|------------|--------------|--------------|--------------|
> | KL divergence | 0.1070     | 0.0541       | 0.0934       | 0.1433       |
>
> As shown in the table, for most queries, the KL divergence is low, and for nearly half of the queries, it is below 0.1.
> **This strongly suggests that the deviations in attention scores between the models are very small**, and the average attention weights obtained from BERT are applicable to each head in the Llama2 model for reflecting the importance of different privacy spans.

---

> ### Author Response · Authors · 2024-11-20
> **Authors' Response (3/6)**
>
> >Q1. L269: "\lambda ..." Which coefficient? What does it do? Where did you introduce it?
>
> The coefficient is one part of the ORPO loss, which was proposed by previous work [1].
> It is used to balance the LLM's ability to produce the correct output $a$ and to distinguish between the correct output $a$ and the false outputs $\hat{a}$.
> The design of the ORPO loss is not our contribution, and we do not explain it in great detail.
> We will add a more thorough explanation of the ORPO loss in the appendix to make the paper easier to read.
>
> [1] Hong, J., et al. "ORPO: Monolithic preference optimization without reference model".
>
> >Q2. L068-L070: What do you want to say here? "if privacy spans [...] are removed [...] attackers are able to recover private information" ?
>
> Thank you for your careful inspection.
> There was a typo here; what we want to say is that **if privacy spans are removed, the attacker cannot recover private information**, as most of the sensitive data is contained within these privacy spans.
>
> >Q3. L205-210: The description of 'probes' a bit too complex.
> >Minor question: The usage of affected attention heads "edited" instead of just "affected attention heads".
>
> The reason why we choose the term "probe" is that the term is always used in the the previous works[1, 2, 3] and we just continue to follow these previous works.
> A probe is a classifier designed to identify the heads most closely related to these privacy spans. We have illustrated the definition of the probe in the formula 1, Line 209-210.
> For the item "edited", it also has been used in previous work [2,3,4]. As illustrated in the Related Work part of [2], not only the weights of LLM can be edited, but the activations (outputs of those attention heads) can also be edited.
> And our method built upon this category of "activation editing", therefore, we choose the "edited" name rather than "affected".
> Even though, we will addtional explanation of this term into our appendix for better reading experience.
>
> [1] Belinkov, Y. "Probing classifiers: Promises, shortcomings, and advances". Computational Linguistics, 48(1), 207–219, 2022.
> [2] Li, K., et al. "Inference-time intervention: Eliciting truthful answers from a language model". NeurIPS 2024.
> [3] Chen, Z., et al. "Truth forest: Toward multi-scale truthfulness in large language models through intervention without tuning". AAAI 2024.
> [4] Hernandez, E., et al. "Inspecting and editing knowledge representations in language models". COLM 2024.
>
> >Q4. L269: In "a is the initial output given the complete input", what output do you mean?
>
> Here, $a$ is the generated **text** response from LLM when given the complete input with privacy spans.
> And $\hat{a}$ is the generated **text** response from LLM when given the incomplete input without privacy spans.
> By using $a$ and  $\hat{a}$, ORPO loss (the loss we used in training process) will encourage the model to generate the initial text response $a$ rather than $\hat{a}$, even given the incomplete input without privacy spans, as shown in Line 270-272.
> Here we provide an example:
> ```
> [Complete input]:
> I'm a 51-year-old woman with a history of chronic pancreatitis, diabetes, and obesity. My family has a history of pancreatic cancer. Lately, I've been experiencing a persistent cough, frequent diarrhea, nausea, abdominal pain, pale stools, dark urine, and I've noticed that I've been losing weight unintentionally. I'm concerned about these symptoms and wondering if they could be related to my existing medical conditions or if there might be something else going on.
>
> [Incomplete input without privacy spans]:
> A 51-year-old woman presents the symptoms of cough, nausea. What is the likely diagnosis?
>
> [The output given complete input]:
> Based on the symptoms and medical history provided, the likely diagnosis for the 51-year-old woman is pancreatic cancer.\n\nPancreatic cancer is a common cause of abdominal pain, weight loss, and gastrointestinal symptoms, including cough, diarrhea, nausea, and pale or dark urine. ...
>
> [The output give incomplete input]:
> Thank you for the question. Based on the symptoms provided, the likely diagnosis for a 51-year-old woman presenting with cough and nausea is a respiratory tract infection, such as bronchitis or pneumonia.\n\nBronchitis is an inflammation of the bronchial tubes ....
> ```

---

> > ### Comment · Reviewer_MqBb · 2024-11-27
> >
> > Thanks for the explanations/glad if it helped improve the paper.

---

> ### Author Response · Authors · 2024-11-20
> **Authors' Response (4/6)**
>
> >Q5: L252-272: (Related to the previous question.) Where and how would the restoration vectors be applied to actually restore something?
>
> In fact, Section 4.2 (inference stage) has explained how and where the meta vector is applied.
> Firstly, if the user input contains privacy spans, then the client will compute the meta vector using the restoration vectors corresponding to these privacy spans.
> The detailed process of computing the meta vector have been illustrated in the "Meta Vector Construction" part of Section 4.2.
> The meta vector always can be produced unless the user query contains not privacy spans.
> Then the privacy protection is meaningless and the plain text of user query can be transmitted to the server directly.
> However, this is extremely rare because most realistic user queries will contain one or more privacy.
> After the client produces the meta vector and transmits it to the server, the server utilizes the meta vector to restore something.
> The detailed process is presented in the "Privacy Restoration" part of Section 4.2.
> Briefly, the method involves adding the meta vector to the attention head output to control the inference of the LLM, which is the paradigm used in previous activation steering works [1,2].
>
> [1] Li, K., et al. "Inference-time intervention: Eliciting truthful answers from a language model". NeurIPS 2024.
> [2] Chen, Z., et al. "Truth forest: Toward multi-scale truthfulness in large language models through intervention without tuning". AAAI 2024.
>
> >Q6. L294 (and elsewhere): You talk about "meta vector R_h on head h", so there is not just one meta vector, but several meta vectors, one for each affected/'edited' attention head h \in H_c? ... you often refer to 'the' or 'a' meta vector in singular form. Note that this is important also for determining the overall privacy budget, cf. weakness on Theorem 5.2.-
>
> The reason why we use 'the' and 'a' is that we concanate all vectors to form a single vector before weighted addition, normalization, and adding noise.
> For the question concerning the privacy budget of our method, please refer to the answer of Q2.
>
> >Q7.1. L345-355: You state the number of "types of privacy spans". What is the meaning of a 'type'? Can you give one or two examples?
>
> Here we present an example to illustrate what is the meaning of a 'type'.
> In the following examples, both "feeling quite moody" and "mood swing" are related to Mood Instability.
> Therefore, **Mood Instability** is regarded as the type of these slightly different expressions.
> ```
> [Question A]:
> I'm with a history of using oral antibiotics ... Along with that, I've been feeling quite moody and irritable, ... what the likely diagnosis might be?
> [Question B]:
> I'm a man who recently got vaccinated, and now I'm experiencing a cough, fever, mood swings, stuffy nose, and pain. What could be causing these symptoms?
> ```
>
> >Q7.2. L345-355: How many actual privacy spans for each dataset?
>
> We have included the average number of privacy spans in each sentence of three datasets in Table 4. We also add more information about total number of privacy spans for each dataset as follows:
>
> | Datasets  | Pri-DDXPlus | Pri-NLICE | Pri-SLJA |
> |------------|------------|--------------|--------------|
> | Number of privacy spans| 46152 | 18246 | 10425       |
>
> >Q7.3. L345-355: What is the impact of the number of spans on the computational overhead?
>
> First of all, we only use a lightweight model (BERT) to compute the attention weights and the time cost of compute weight is relately very small compared to the time cost of the LLM inference in the server.
> Secondly, the BERT model architecture is not autoregressive; regardless of the number of privacy spans in the user's question, the BERT model requires only one forward pass.
> Finally, we also compute the time cost of the BERT model in the client and the LLM model in the server in the following table and prove the time cost of BERT is relately small (only **12.83%**).
>
> | | Time cost of the Client | Time cost of the Server | Total time Cost |
> |------------|------------|--------------|--------------|
> | Latency(ms)| 88.52 | 601.46  | 689.98       |
> | Proportion| 12.83% | 87.17%  | 100%      |

---

> ### Author Response · Authors · 2024-11-20
> **Authors' Response (5/6)**
>
> >Q8.1. Section 6.3: You use an embedding inversion attack ... Then the attacker could train a model to predict the privacy span given the context (infilling/masked language modeling) and the noisy meta vectors, which could narrow down the predictions to more suitable ones.
>
> Actually, we implement the embedding inverse attack just following the previous work [1].
> Convert the sanitized text input into embedding representation (the meta vector) and recover the privacy by utilizing finetuned GPT-2 model according to the meta vector.
> We have provided all the implementation details in Appendix M, and your guess is correct.
>
> The question raised by you is interesting. The input of the attack model (GPT-2 model) can be both the meta vector and the incomplete query.
> And we add additional attack experiments by training a GPT-2 model to recover privacy according to the meta vector and the incomplete query.
> To be concrete, we convert the incomplete query into serveral token embeddings and concanate these these token embeddings with the meta vector as the input of the attack model (GPT-2).
> The attack performance (ROUGE-L) is as follows, with the better performances highlighted in **bold**:
>
> | Only meta vector              | ε=1    | ε=20   | ε=40  | ε=75   | ε=125  | ε=175  |
> |------------------|------|------|-----|------|------|------|
> | Pri-DDXPlus      | 0.1190 | **0.2160** | 0.2306 | 0.1901 |**0.2259**| 0.1821 |
> | Pri-NLICE        | 0.1312 | 0.2286 | 0.2103 |**0.2268**| 0.2160 |**0.2083**|
> | Pri-SLJA         | 0.1730 | 0.2803 |**0.2965**| **0.2995** |**0.2964**|**0.2921**|
>
> | Meta vector + incomplete query         | ε=1    | ε=20   | ε=40  | ε=75   | ε=125  | ε=175  |
> |------------------|------|------|-----|------|------|------|
> | Pri-DDXPlus      | **0.1478** | 0.1893 | **0.2346** | **0.1951** | 0.1965 | **0.2423** |
> | Pri-NLICE        | **0.1426** | **0.2317** | **0.2187** | 0.2252 | **0.2190** | 0.1939 |
> | Pri-SLJA         | **0.1987** | **0.2901** | 0.2953 | 0.2686 | 0.2882 | 0.2758 |
>
> As the above table shows, the improvement by combining meta vector and the incomplete query is limited, the reason is that the content of incomplete query is not relevant to the privacy and not helpful for attack performance.
> These experiment results also sidely demonstrate our Assumption 1 in Line 64-66: attack can not recover private information from the sanitized input.
> Even for certain values of $\epsilon$, adding an incomplete query can harm attack performance, as the additional incomplete query is not directly related to the private information.
> This distracts the attention of the attack model (GPT-2), leading to degraded performance.
> In a word, no matter adding the incomplete query or not, the attack performances remains consistently low across different $\epsilon$, which demonstrates the effectiveness of our method in privacy protection.
>
> [1] Li, H., et al. "Sentence embedding leaks more information than you expect: Generative embedding inversion attack to recover the whole sentence". ACL 2023.
>
> >Q8.2. Section 6.3: In the "different d_X-privacy Percentage" experiments, you vary the percentage of protected tokens, but I'd assume that you leave the overall privacy budget the same?
>
> Indeed you may confuse the "different $d_\chi$-privacy Percentage" experiment with the main experiemnt of Section 6.2 and Table 1.
> For the main experiment, we want to prove that our method demonstrates superior performance compared to other privacy-perserving baselines under the same strength of privacy protection.
> Therefore, we need to control the overall privacy budget the same. However, for the "different $d_\chi$-privacy Percentage" experiment, we aim to demonstrate that the privacy budget of traditional $d_\chi$-privacy protection increases with the percentage of protected tokens, which weakens the strength of privacy protection.
> The goal of this experiment is not to make fair comparions among different privacy protection methods and the assumption of keeping the same privacy budget is not applicabel here.
> **In fact, this experiment focuses on testing the strength of privacy protection with different percentages of protected tokens** and the privacy budget is always not the same.

---

> ### Author Response · Authors · 2024-11-20
> **Authors' Response (6/6)**
>
> >Q8.3. Section 6.3: Why the blue curve (Pri-NLICE) goes up linearly in Fig 3.(c)? Could more protected privacy spans actually lead to more information leakage?
>
> Thank you for your careful observation. Indeed, we also have observed this strange phenomenon and give the related explantions in Line 1277-1284 in Appendix L.2.
> Here we briefly summarize again.
> **The abnormal rise of the attack performance of embedding inverse attack in Pri-NLICE is because the used metric ROUGE-L iteself rather than more information leakage.**
> Since ROUGE-L measures the overlap between the output generated by the attacker (using the GPT-2 model) and the reference text (privacy spans), more privacy spans result in a longer reference text, leading to greater overlap and a slight increase in ROUGE-L.
> This increase is caused by the longer reference text rather than the better attack performance.
> Indeed, the improvement is little, rising from 0.12 (30%) to 0.19 (90%), a difference of only 0.07.
> The other experiment results of embedding inverse attack on the other two datasets and attribute inference attack on all three datasets are all stable, strongly demonstrating that more privacy spans will not lead to more information leakage if protected by our method.
>
> ---
> We sincerely thank the reviewer for their efforts in review our paper.
> **We are looking forward to your reply. If you have any further questions, we are happy to dicuss them!**

---

> > ### Comment · Reviewer_MqBb · 2024-11-27
> >
> > Q8.3. I think the observation is a valuable insight and should be discussed also in the paper. Would it make more sense to normalize the ROGUE-L metric according to the length of the reference text to reduce the impact of the reference text length (and hence of the number of privacy spans) on the attack score?

---

> ### Author Response · Authors · 2024-11-26
> **Kind Remind**
>
> Dear reviewer: This is a polite reminder that we are awaiting your feedback on our rebuttal. To assist you, we have highlighted the key points for your convenience. We would greatly appreciate it if you could discuss them with us at your earliest convenience.

---

> ### Comment · Reviewer_MqBb · 2024-11-27
>
> W2.2. Differential privacy in general is understood as a worst-case guarantee that is related to the _sensitivity_ of its inputs, which here would be the worst-case (i.e., _maximum_) distance between embedding vectors. By using $d_X$-privacy (as introduced in Def. 2.1) instead of normal differential privacy, you can avoid the need for defining a sensitivity value. However, you still cannot simply take the average or median distance (as you state in L1065) between embedding vectors $d_e$ or $d_z$ and multiply that average with $\epsilon$ to get the budget. If you perturb one embedding vector, the privacy budget still is $\epsilon$ and not $\text{some-average-distance}\cdot\epsilon$.
>
> W3. Sure, but even the best NER / "text sanitization" methods may miss some texts that relate to a privacy-sensitive concept, or maybe the user of your methods has overseen a privacy-sensitive concept and forgot to model it in the text sanitization method. If you rely on those classical text sanitization methods, why even bother applying DP and sending the perturbed meta vectors instead of just masking the privacy spans?
>
> W4. Thanks for the update and computing the KL divergences. I can see it makes some sense intuitively.

---

> > ### Author Response · Authors · 2024-11-29
> > **Further Authors' Response (2/3)**
> >
> > > **The definition and sufficiency of privacy spans. (1/2)**
> > Sure, but even the best NER / "text sanitization" methods may miss some texts that relate to a privacy-sensitive concept, or maybe the user of your methods has overseen a privacy-sensitive concept and forgot to model it in the text sanitization method.
> >
> > Thank you for your thoughtful concern about the effectiveness of the best temporary text sanitization method.
> > It is true that even the best sanitization methods cannot achieve 100% accuracy in identifying privacy spans.
> > However, **the gap is very small**. We have evaluated the identification of privacy spans using the text sanitization method [1,2] and **achieved an accuracy of nearly 99%**, which is sufficient for real-world applications.
> > For detailed implementation and experiment results, please refer to the Step 2 of General Q3 in general response.
> > Here we only repeat the experiment results again:
> >
> > | |Precision|Recall |F1|
> > |-|-|-|-|
> > Privacy Spans Identification|99.16 $_{\pm 0.21}$|98.78 $_{\pm 0.31}$|99.66 $_{\pm 0.27}$
> >
> > Furthermore, we also implement both Embedding Inverse Attack and Attribute Inference Attack to evaluate the privacy protection performance of text sanitization, as follows:
> >
> >
> > | |Embedding Inverse Attack (ROUGE-L )|Attribute Inference Attack (F1)
> > |-|-|-
> > No Protection Results|0.4069|0.7028
> > Sanitized Results|0.0606 $(\downarrow 0.3463)$|0.0705 $(\downarrow 0.6323)$
> >
> > The results above show that the current text sanitization method **is sufficient to identify and protect most privacy spans in real-world scenarios**, even though the identification accuracy is 99% rather than 100%.
> >
> > [1] Chen, Y., et al. "Hide and seek (HAS): A lightweight framework for prompt privacy protection".
> > [2] Kan, Z., et al. "Protecting user privacy in remote conversational systems: A privacy-preserving framework based on text sanitization".
> >
> > >**The definition and sufficiency of privacy spans. (2/2)**
> > If you rely on those classical text sanitization methods, why even bother applying DP and sending the perturbed meta vectors instead of just masking the privacy spans?
> >
> > You may have overlooked an important point: these text sanitization methods only identify and remove the privacy spans from the user's question **without any restoraion of inference utility**.
> > This inevitably harms the utility of the generated output of the server LLM, as the sanitized question lacks the information contained in the removed privacy spans.
> > This is why we propose PrivacyRestore, which **aims to restore the utility of the generated output while ensuring privacy protection**.
> > Specifically, we construct meta vectors, apply DP to protect these vectors, and then use the perturbed meta vectors to restore the utility of the generated output during LLM inference on the server.
> > Moreover, it is emphasized that our method is orthogonal to those text sanitization methods and the main goal of our method is to **restore the utility of generation** while protecting privacy.
> > Here, we also add additional experiments to evaluate the performance of only using text sanitization, as well as combining text sanitization and PrivacyRestore. The experiments results are as follows:
> >
> > | |MC1|MC2|ROUGE-L|LLM-J
> > |-|-|-|-|-
> > No Restoration(lower bound)|33.57|32.49|25.19|3.21
> > Only use sanitization(without PrivacyRestore)|29.63 $({\downarrow3.94})$|30.85 $({\downarrow1.64})$|25.45 $({\uparrow0.26})$|3.46 $({\uparrow0.25})$
> > Sanitization + PrivacyRestore|62.87 $({\uparrow29.30})$|59.97 $({\uparrow27.48})$|26.47 $({\uparrow1.28})$|4.28 $({\uparrow1.07})$
> >
> > It is obvious that our method **effectively restores utility and output performance** compared to using text sanitization alone without PrivacyRestore.
> > Moreover, using only text sanitization methods leads to performance degradation compared to the No Restoration baseline (lower bound),
> > which assumes that all privacy spans have been removed from the question and that non-privacy parts are unaffected.
> > This phenomenon may result from **text sanitization slightly affecting the non-privacy parts of the question**, leading to a decrease in performance.

---

> ### Comment · Reviewer_MqBb · 2024-11-27
>
> Thanks for providing further details, esp. on the overhead by running the local BERT model.
>
> Q7.1. If you talk about "privacy types" in the paper, please also explain in the paper what they are.
>
> Q6 (and related to W2.1). I still think this is at least confusing, if not incorrect in certain places. For instance, Eq. 6 clearly states that noise is applied to the meta vector _for each_ head, while you explained in your first author response that in fact you concatenate the meta vectors of all heads first, and then apply noise only once to the concatenated vector. This is not what you express with the formula in the paper, and you didn't mention anything about concatenating the vectors first in the paper. As explained, this is a critical piece of information since it impacts the overall privacy budget, and hence, crucial to explain _in_ the paper (as you promised in your first response).

---

> ### Comment · Reviewer_MqBb · 2024-11-27
>
> Q8.1. Thanks for the extra experiment on the inversion attack in the short time. The attack using GPT2 with extra context information is a good start. I think the attack can still be improved, since as you state, the extra information may confuse GPT2's attention scores, so in fact _it's the employed model_ (GPT2 with simple injection of the meta vector as an input embedding) _that may not quite be up for the task_. The Pri-SLJA dataset has the longest input lengths according to Fig. 4b), which would support that the GPT2 model is confused by the large amount of input tokens next to the meta vector, hence this may explain why the attack with context information only improves in two cases (eps=1 and 20) for that long-input dataset. The other datasets with shorter inputs show more improvements.
> A clever attacker who intercepts the sanitized inputs and meta vectors might employ activation steering exactly as you describe in Sec. 4.2 -- _but for the inversion attack model_ which he trains/finetunes to generate the text redacted from the privacy spans. This way, the model might be less confused by the extra information, because you inject the meta vectors right at the correct place without having to rely on some attention mechanism figuring things out by itself (and getting distracted by additional tokens). In summary, as said it's a good start, but inconclusive without trying more sophisticated attacks.
>
> Q8.2. Fair enough. I didn't understand it from the current text in the paper, so I think it would be helpful to state that more explicitly in the paper.

---

> > ### Author Response · Authors · 2024-11-29
> > **Further Authors' Response (3/3)**
> >
> > > **Evaluation using potentially stronger attacker models.**
> > Q8.1. ... The attack using GPT2 with extra context information is a good start. I think the attack can still be improved, since as you state, the extra information may confuse GPT2's attention scores, ... A clever attacker who intercepts the sanitized inputs and meta vectors might employ activation steering exactly as you describe in Sec. 4.2 -- but for the inversion attack model which he trains/finetunes to generate the text redacted from the privacy spans. ... In summary, as said it's a good start, but inconclusive without trying more sophisticated attacks.
> >
> > Thank you for your thoughtful and detailed feedback on our work.
> > We greatly appreciate your recognition of the additional experiment on the inversion attack and your insightful suggestions regarding potential improvements of our attack model.
> > Considering that a clever attack may exploit the extra information in a way that does not confuse the GPT-2 model, we have implemented two additional attack methods:
> > 1. The first attack method is that the attackers **may use the intercepted meta vectors to steer the activations of the GPT-2 attack model** when recovering the privacy spans, rather than directly concatenating the meta vector with the sanitized question as inputs to the GPT-2 model. Here, we refer to this attack method as **Simulating Activation Steering Attack**.
> > 2. Then since directly concatenating the meta vector with the sanitized question as inputs may lead to inconsistency between the meta vector and the text inputs, we propose a second attack method called **Concatenated Text Attack**.
> > For Concatenated Text Attack, attack will train two attack models, where the first model is used to inverse the meta vector to the privacy spans and the second model is used to **polish the privacy spans given the additional sanitization question**.
> > Specifically, the input to the first model is the meta vector, and the output is the recovered privacy spans.
> > These recovered spans are then concatenated with the sanitized question and input into the second model to obtain the final polished privacy spans.
> >
> > The attack performance results of both attack methods are shown in the following table:
> > Simulating  Activation Steering Attack (ROUGE-L)|1|20|40|75|125|175
> > |-|-|-|-|-|-|-|
> > Pri-DDXPlus|0.0023|0.0329|0.0321|0.0329|0.0310|0.0365
> > Pri-NLICE|0.0165|0.0123|0.0118|0.0170|0.0283|0.0315
> > Pri-SLJA|0.0161|0.0818|0.0862|0.0861|0.1048|0.1059
> >
> > Concatenated Text Attack (ROUGE-L)|1|20|40|75|125|175
> > |--|--|--|-|-|-|-|
> > Pri-DDXPlus|0.0112|0.0107|0.0130|0.0093|0.0115|0.0024
> > Pri-NLICE|0.0566|0.0486|0.0427|0.0467|0.0423|0.0350
> > Pri-SLJA|0.0027|0.0011|0.0022|0.0024|0.0021| 0.0028
> >
> > As the results above show, the performance of both attack methods is weak, even weaker than the previous Embedding Inverse Method (which uses only the meta vector or the meta vector combined with the sanitized question as model inputs).
> > The weak performance of the Simulating Activation Steering Attack may be due to the fact that **the meta vector was offline trained for the server LLM model** and **is not compatible with steering the activation of the attack model (GPT-2)**, which differs from the server LLM in its architectural design.
> > As for the Concatenated Text Attack, **the initial performance of using the given meta vector to recover the privacy spans is poor** and then prompting another attack model with the sanitized question to refine the recovered spans still results in weak performance.
> >
> > So far, we have tried **all known possible strong attack models** (Attribute Inference Attack, Embedding Inverse Attack with only meta vector, Embedding Inverse Attack with meta vector and sanitized question, Simulating Activation Steering Attack, Concatenated Text Attack),
> > and our methods have demonstrated strong privacy protection capabilities against all these attack methods.
> > Even if there may be more and more advanced attack methods in the future, we believe that our method will still have good privacy protection performance.
> > If you have any further ideas on how to attack our method, we would be happy to discuss it here!
> >
> > ---
> > We sincerely thank you once again for your valuable and timely feedback, which has greatly strengthened our paper. Your increased score has truly inspired us. We are deeply grateful.
> > We hope our additional responses address your remaining concerns and are sufficient to earn your positive score, which would be both helpful and inspiring for us.
> >
> > **We very look forward to your further feedback on our paper or any other concerns. If you have additional questions, we would be happy to discuss them.**
> > **Thank you once again!**

---

> ### Comment · Reviewer_MqBb · 2024-11-27
>
> Given that one of the main issues I was having regarding the correctness of the total privacy budget (Thm 5.2) was explained in the author responses, I have raised my score of soundness and the overall rating, under the assumption that the authors will incorporate these and other (crucial) explanations in the next version of their paper. Note that it would have been good practice if the submitted paper already had a clearer presentation at the time of (final) submission (e.g., that you in fact perturb the concatenated meta vectors, not the per-head meta vectors as explicitly written in the formula in the paper, and other questions that were addressed in the author comments). In my opinion, this could likely be the reason for the symptomatic large number of questions and long list of author responses also for other reviewers.
>
> Some doubts still remain about the usability of the proposed method w.r.t. the definition and sufficiency of privacy spans, the use of average distances for determining the privacy budget, and evaluation using potentially stronger attacker models. However, I am confident that the paper can make it to acceptance if the authors put in the necessary work.

---

> ### Author Response · Authors · 2024-11-29
> **Further Authors' Response (1/3)**
>
> We sincerely thank you for your timely response and your efforts in reviewing our paper.
> It is truly inspiring to see you raise your score!
> Below, we have addressed all of your remaining concerns by adding more experiments and explanations.
> We hope all these more experiments and explanations can solve your remaining concerns.
>
> >**The use of average distances for determining privacy budget.**
> W2.2: Differential privacy in general is understood as a worst-case guarantee that is related to the sensitivity of its inputs, which here would be the worst-case (i.e., maximum) distance between embedding vectors. ...
>
> Thank you for your helpful advice about the general definition of Differential Privacy.
> In fact, the reason why we consider using average distance between embedding vectors/meta vectors rather than the maximum distance is that **we have observed occasional outliers with large distance values, though their frequency is low**.
> Using the maximum distance directly could result in an overly **loose estimation of the privacy budget**.
> The average distance, used here for calculating the total privacy budget, may be more effective in mitigating the impact of these outliers.
> However, as proposed by you, the general definition of Differential Privacy (DP) should account for the worst-case scenario, even though such situations may occur infrequently.
> Therefore, we **re-compute the privacy hyperparameter $\epsilon$** using the general definition of Differential Privacy, which **employs the maximum distance** to calculate the privacy budget,
> as the following table shows.
> |$d_\chi$-privacy| |$d_\chi$-privacy on privacy spans| |Paraphrase| |PrivacyRestore |  |
> |-|-|-|-|-|-|-|-|
> |$n \cdot d_e$ (avg)|$\epsilon$ |$n_{sp} \cdot d_{e}$ (avg)| $\epsilon$| $n$ (avg)| $\tau$| $d_{z}$ (avg)| $\epsilon$ |
> 69.63 $\cdot$ 1.12|0.87 |20.28 $\cdot$ 1.12| 3.00| 69.63 | 2.04 | 0.91 | 75.00 |
> $n \cdot d_e$ (max)|$\epsilon$|$n_{sp} \cdot d_{e}$ (max)|$\epsilon$|$n$ (max)|$\tau$|$d_{z}$ (max)|$\epsilon$ |
> 106 $\cdot$ 1.64|0.86 $(\downarrow 0.01)$ |49 $\cdot$ 1.64| 1.87 $(\downarrow1.13)$ | 106 | 1.42 $( \downarrow 0.62)$| 2 | 75.00 |
>
> Here we still **keep the privacy hyperparameter $\epsilon$ of PrivacyRestore as 75** and **the total privacy budget increase to 150** because we use the maximum value of $d_z$ for computing the total privacy budget.
> To ensure a fair comparison between our method and the baselines, we re-compute the privacy hyperparameters for these baseline methods also based on the maximum values of  $n,n_{sp},d_e, d_z$ ,
> while **keeping the total privacy budget at 150**.
> And we can observe the smaller $\epsilon$ values for these baselines, indicating a **stricter privacy guarantee**, because we must ensure the same privacy protection for the outlier samples.
> And then, we re-evaluate the performance of these baselines under the new privacy hyperparameters $\epsilon$ and the experiment results are as follows, with the best performances highlighted in bold:
>
> |Pri-DDXPlus | MC1 | MC2 | ROUGE-L | LLM-J |
> |-|-|-|-|-|
> |$d_\chi$-privacy | 28.79 $_{\pm 0.02}$ | 30.26 $_{\pm 0.01}$ | 18.03 $_{\pm 0.00}$ | 1.14 $_{\pm 0.00}$ |
> | $d_\chi$-privacy on privacy spans | 44.74 $_{\pm 0.21}$ | 42.28 $_{\pm 0.01}$ | **29.02** $_{\pm 0.06}$ | 3.31 $_{\pm 0.00}$ |
> |Paraphrase | 27.96 $_{\pm 0.14}$ | 28.54 $_{\pm 0.06}$ | 18.08 $_{\pm 0.02}$ | 1.20 $_{\pm 0.02}$ |
> |PrivacyRestore | **62.97** $_{\pm 0.00}$ | **60.19** $_{\pm 0.00}$ | 27.24 $_{\pm 0.26}$ | **4.47** $_{\pm 0.00}$ |
>
> As the above results show, **our method still outperforms all these privacy-preserving baseline methods**, even using the privacy hyperparameter $\epsilon$ computed according to the wrost guarantee of DP.
> For the reason why the ROUGE-L score of $d_\chi$-privacy on privacy spans is higher than our method has been explained in Lines 424-427 of our paper.
> Breifly, it is because ROUGE-L scores **focus solely on the lexical similarity** between the generated and reference outputs, without accounting for the **actual quality of the generated outputs**.

---

### Official Review · Reviewer_GKPj · 2024-11-03

**Soundness:** 1
**Presentation:** 2
**Contribution:** 1
**Rating:** 3
**Confidence:** 3

**Summary:**

The paper proposes a method for a privacy-preserving inference in LLMs, satisfying $d_\mathcal{X}$ privacy, a notion originally proposed as an adaptation of local DP to location data. Authors define *privacy spans* - a portions of text containing sensitive information. They then define a method to obtain a prediction from an API-accessed LLM, while sending a query with stripped privacy spans, plus private information encoded in a *restoration vector*. The server will have trained restoration vectors for existing privacy spans (of which authors assume a limited number of), to later use restoration vectors for activation steering. The authors created 3 privacy-focused datasets and presents experiments showing their method to outperform naive application of $d_\mathcal{X}$ privacy, as well as paraphrasing, for a task at hand.

**Strengths:**

The paper raises an important question of the appropriate privacy unit for text processing. While DP for language models has been studied extensively, the field lacks consensus on what threat model should be applied to text, and whether sequence-level or document-level approaches appropriately mitigate privacy risks.

**Weaknesses:**

## List of privacy spans.
The paper assumes a closed list of available privacy spans - less than 200 words/phrases used in the experimental settings. First, this introduces a privacy leakage from the mere fact of removing privacy spans - it reveals to the attacker that an original text contained one of the few "sensitive" items. Second, privacy is contextual, and defining an exhaustive list of sensitive phrases for a real-world use might be tricky, even if limited to a narrow domain, e.g. medical. What is considered safe for some people, will be sensitive for others.

## Privacy guarantees
I did not fully understand the threat model and the privacy guarantees provided. Throughout the paper, authors refer to attackers *intercepting* user queries (lines 41, 168), which suggests they are not concerned with protecting the sensitive queries from model developers. If that's the case, however, the standard encryption protocols should suffice to protect the traffic in-flight. Further, if an LLM developer is not considered a threat, an adversary should be considered that has access to the released restoration vectors and can perform the same computations as the server in the proposed setup.

Moreover, I did not follow how privacy guarantees are comparable across methods. On line 404 authors claim to ensure that all privacy methods are under the same privacy budget. Table 5, however, shows $\epsilon=75$ for PrivacyRestore, and significantly lower values for other methods.

In Theorem 5.2 authors claim to provide $d_\mathcal{X}$-privacy with a privacy budget depending on the distance between two restoration vectors. While I'm not very familiar with $d_\mathcal{X}$-privacy, typically data-dependent privacy budget merits a very careful proof, as the privacy budget itself could potentially leak private information.

**Questions:**

* The paper introduces SMPC in the "Related Works" section. How is SMPC relevant for the technique proposed in the paper?
* The metrics MC1 and MC2 (lines 1167-1176) used in the paper seem to simply refer to accuracy and entropy. The cited paper Lin et al., 2021 does not seem to define these metrics either.
* If MC1 is indeed an accuracy of selecting one out of 4 options, how is the performance of some methods drops consistently and statistically significant below 25% random guess baseline?
* In Figure 2, the attack success rate does not seem to grow significantly with increasing privacy budget. Do you have an explanation for that?
* In table 1, I would like to see non-DP baselines. First, a method that does not provide any privacy guarantees. And second, a method that simply removes privacy spans and asks LLM to provide an output based on the incomplete information. That would help put the performance of the methods considered into a proper context.

---

> ### Author Response · Authors · 2024-11-20
> **Authors' Response (1/5)**
>
> We sincerely thank you for your feedback and suggestions! We address all your concerns below:
>
> **To be mentioned, we have updated our submitted paper, please download the latest one, thank you.**
>
> >W1.1. List of privacy spans. The paper assumes a closed list of available privacy spans ... this introduces a privacy leakage from the mere fact of removing privacy spans - it reveals to the attacker that an original text contained one of the few "sensitive" items.
>
> Firstly, the predefined set of privacy spans does not pose a privacy risk.
> You have overlooked an important fact: the number of privacy spans in the original user text is not fixed and impossible just "one of", as you said.
> In reality, **the number of privacy spans is random and can range from 0 to any value less than the size of the predefined set**.
> Even if attackers gain access to the predefined set, it is nearly impossible to determine how many privacy spans are present in the original text and what these privacy spans are.
> Ideally, the possible combinations of privacy spans in the original text would be $2^N$, where N is the size of predefined set.
> With the size of our predefined set nearly 150, it is virtually impossible for attackers to determine the correct combination from the $2^{150}$ possible choices. Therefore, assuming a predefined set will not pose any privacy risk in fact.
>
>
>
> >W1.2. List of privacy spans. Second, privacy is contextual, and defining an exhaustive list of sensitive phrases for a real-world use might be tricky, even if limited to a narrow domain, e.g. medical. What is considered safe for some people, will be sensitive for others.
>
> Defining an exhaustive list of sensitive privacy spans is not tricky and is realistic.
> You have misunderstood the application of our method.
> The realistic application of our method is that **the user can determine the what is privacy by themselves and find the most relevant privacy spans in the predefined list to construct the restoraion vectors**.
> Previous 'text sanitization' methods [4,5] automatically identify and sanitize privacy, without giving users the right to determine what is privacy for themselves.
> This approach violates the principle of 'informational self-determination' [1,2,3] and overlooks the fact that privacy is contextual.
> However, our method assumes that users can determine privacy on their own, and the **predefined privacy spans list serves merely as a scope** within which most privacy spans fall.
> Even if an item from the predefined set appears in the user input, it will not be removed if the user deems it safe. The right to determine and remove privacy spans rests with the user.
> You may raise the question of **what happens if those privacy spans determined by the users do not correspond to any privacy spans in the predefined set**.
> Indeed, we have demonstrated in Q2 of General Response that the distribution of privacy is long-tailed and most privacy will focus on the majority category.
> And even encountering the privacy that falls out of our predefined set, the user can simply select the semantically closest one from the predefined set as the substitute.
> And the performance of our method can still be superior compared to other privacy preserving methods, as shown in Q2 of General Response. For more details, please refer to the Q2 of General Response.
> Moreover, we also consider the situation when **the user can not determine the privacy by themselves**.
> In this situation, even though text sanitization may not be able to identify contextual privacy for different users, it still serves as an efficient approach to identifying privacy and removing it from user inputs. Since our method is orthogonal to existing text sanitization approaches [4,5], it can easily be combined with these approaches to automatically identify privacy spans and do restoration during LLM inference. The detailed implementations are shown in Q3 of General Response. As the experiment results in Q3 of General Response shows, the performance of combining our method with text sannitization techniques can still outperform other privacy-perserving methods and effectively handle this situation.
>
> [1] "Informational self-determination". Wikipedia. https://en.wikipedia.org/wiki/Informational_self-determination.
> [2] Jasper, M. C. "Privacy and the Internet: Your Expectations and Rights Under the Law". Oxford University Press 2009.
> [3] Van Alsenoy, et al. "Privacy notices versus informational self-determination: Minding the gap". International Review of Law, Computers & Technology 2014.
> [4] Chen, et al. "Hide and seek (HAS): A lightweight framework for prompt privacy protection".
> [5] Kan, et al. "Protecting user privacy in remote conversational systems: A privacy-preserving framework based on text sanitization".

---

> ### Author Response · Authors · 2024-11-20
> **Authors' Response (2/5)**
>
> >W2.1 Privacy guarantees. I did not fully understand the threat model and the privacy guarantees provided. Throughout the paper, authors refer to attackers intercepting user queries (lines 41, 168), which suggests they are not concerned with protecting the sensitive queries from model developers. If that's the case, however, the standard encryption protocols should suffice to protect the traffic in-flight. Further, if an LLM developer is not considered a threat, an adversary should be considered that has access to the released restoration vectors and can perform the same computations as the server in the proposed setup.
>
> Indeed, the actual threat model of 'attacker intercepting user queries', considered by us, not only occurs during data transmission, but **attackers can also attack the server to steal queries**. The encryption protocols can not be applicable to the situation when the attacker attacks the server because the encrypted input will be decrypted on the server and attack can easily steal the decrypted results from the server.
> However, our method can effectively protect privacy when attackers directly attack the server. What attackers can steal only be the restoration vectors and the incomplete inputs. An important point you overlook is that the server applies the restoration vectors only to specific attention heads, not all heads, and keeps the edited heads set confidential. Additionally, in most real scenarios, the server ensures the confidentiality of LLM weights, such as GPT-4 of OpenAI, which we have illustrated in the Threat Model Section. **For the lack of the set of edited heads and the weights of LLM, the attacker cannot perform the same computations**.
> Another question raised by you is interesting: What if the server itself is malicious? Indeed, if the server becomes malicious, in addition to exposing the restoration vectors and incomplete input, the generated output may also be obtained by the malicious server. **However, the generated outputs will not leak any private information.** Here we should consider two situations:
> 1. **The LLM on the server generates outputs using sampling based methods** (not directly select the highest probability token), which is controlled by the generation temperature. We have demonstrate that the sampling-based method will not leak any privacy information because the sampling generation is protected by the Exponential Mechanism [1]. The detail explations and proof refer to Q4 of General Response.
> 2. **The LLM generates outputs using the greedy search (direcly select the highest probability)**. Although the server can ideally obtain outputs generated by greedy search (though our method restricts the server to using sampling-based generation), it remains difficult to attack privacy through these outputs. The highest probability token will not leak privacy.
> We have implemented both Attribution Inference Attack and Embedding Inverse Attack to attack the generated output, as shown in the following table. We also count the occurrence of privacy spans in the outputs. The poor attack performance indicates that it is also impossible to attack privacy from the generated outputs, whether using greedy search or sampling generation. Therefore, our method can still protect privacy even when the server is malicious.
>
> Generation Mode|Embedding Inverse Attack|Attribute Inference Attack|Occurrence of privacy spans in outputs
> |--|--|--|--|
> Greedy searching |0.0385|0.1190|0.0310
> Sampling, Temperature=1.0 (We mainly use)|0.0970|0.0375|0.0299
>
> [1] Utpala, Saiteja, et al. "Locally Differentially Private Document Generation Using Zero Shot Prompting." EMNLP 2023.
> [2] Mattern, J., et al. "The limits of word-level differential privacy". NAACL 2022.

---

> ### Author Response · Authors · 2024-11-20
> **Authors' Response (3/5)**
>
> >W2.2 Privacy guarantees. Moreover, I did not follow how privacy guarantees are comparable across methods. On line 404 authors claim to ensure that all privacy methods are under the same privacy budget. Table 5, however, shows 𝜖=75 for PrivacyRestore, and significantly lower values for other methods.
>
> You may misunderstand the relationship between privacy budget and $\epsilon$ (Here we call it privacy hyperparameter, as shown in Line 407).
> **The privacy budget is not equal to** $\epsilon$. The privacy budget is to measure the cost of privacy protection.
> It can also be viewed as the degree of privacy protection strength. **The privacy budget is determined by many factors and the privacy hyperparameter $\epsilon$ is one of the factors**.
> For example, the privacy budget of $d_\chi$-privacy, which is mainly used in our method, is determined simultaneously by **the distance between embeddings, the input length and the $\epsilon$ value** [1,2].
> We have included the illustration of what contributes to the privacy budget of $d_\chi$-privacy in Line 1109-1110 of Appendix H.
> Therefore, for different privacy-preserving methods, the distances between embeddings and the input length to the $d_\chi$-privacy may be different.
> For example, traditional $d_\chi$-privacy applies to the token embeddings and our method applies $d_\chi$-privacy to the meta vectors.
> However, the distance between token embeddings usually differs from the distance between the meta vectors, as shown in the Figure 4 of Appendix G.
> **So to control the total privacy budget the same, we should use different privacy hyperparameters $\epsilon$ for different privacy-preserving methods**.
> The detailed $\epsilon$ values used for different privacy-preserving methods and corresponding calculation processes have been illustrated in Appendix G.
>
> [1] Qu, C., et al. "Natural language understanding with privacy-preserving BERT". CIKM 2021.
> [2] Chatzikokolakis, K., et al. "Broadening the scope of differential privacy using metrics". PETS 2013.
>
>
> >W2.3 Privacy guarantees. In Theorem 5.2 authors claim to provide $d_\chi$-privacy with a privacy budget depending on the distance between two restoration vectors. While I'm not very familiar with $d_\chi$-privacy, typically data-dependent privacy budget merits a very careful proof, as the privacy budget itself could potentially leak private information.
>
> There are some misunderstandings regarding $d_{\chi}$-privacy [1,2], which are causing your confusion about our methods. The dependence of the privacy budget on distance is a common sense concept [1] and not a new proposal from our work. As introduced in prior research [2], the formal definition of $d_{\chi}$-privacy involves assigning different privacy budgets based on the distance between two restoration vectors.
> Furthermore, the privacy budget itself only indicates the strength of the injected noise on the restoration vector to make attacker indistinguishable about the initial vector. Although the privacy budget (noise strength) is related to the distance between restoraion vectors, **the attacker can only obtained the outcome of adding restoration vector and noise, and can not infer the noise strength (privacy budget) from the noisy outcome**. The privacy budget (noise strength) will not be obtained by the attacker in fact and so no private information, distances or other information, will be leaked from the privacy budget indeed.
>
> [1] Qu, C., et al. "Natural language understanding with privacy-preserving BERT". CIKM 2021.
> [2] Chatzikokolakis, K., et al. "Broadening the scope of differential privacy using metrics". PETS 2013.

---

> ### Author Response · Authors · 2024-11-20
> **Authors' Response (4/5)**
>
> >Q1. The paper introduces SMPC in the "Related Works" section. How is SMPC relevant for the technique proposed in the paper?
>
> Our method aims to protect user privacy when using cloud LLM service, which focuses on the inference stage. The SMPC methods enable to collaborative computation without revealing original data to others. And then during cloud LLM inference, **SMPC can be used to enable cloud LLM inference while not reveal raw user input to the cloud**, which can achieve privacy protection for users [1,2]. To be more specific, the primary approach of SMPC-based methods involves fine-tuning a secure LLM that performs inference on encrypted user inputs and generates encrypted outputs. The user can then decrypt the output to obtain the final result. Although SMPC methods require significant resources to fine-tune the LLM and perform encryption and decryption, they remain a viable approach for protecting privacy during inference.
>
> [1] Liang, Z., et al. "Merge: Fast private text generation". AAAI 2024.
> [2] Li, D., et al. "MPCFORMER: Fast, performant, and private transformer inference with MPC". ICLR 2024.
>
>
>
>
> >Q2. The metrics $MC1$ and $MC2$ (lines 1167-1176) used in the paper seem to simply refer to accuracy and entropy. The cited paper Lin et al., 2021 does not seem to define these metrics either.
>
> Both $MC1$ and $MC2$ refer to accuracy, with $MC1$ focusing on the option with the highest probability, while $MC2$ considers all options. **For detailed implementations of $MC1$ and $MC2$, please refer to the released code in our cited paper** [1]. The cited paper utilizes $MC1$/$MC2$ for benchmark evaluation in the released source code, though it is not mentioned in the paper itself. Moreover, $MC1$/$MC2$ options are widely used metrics for test model performance in various previous works [2,3,4].
>
> [1] Lin, S., Hilton, J., and Evans, O. "TruthfulQA". ACL 2022. Available at GitHub: https://github.com/sylinrl/TruthfulQA.
> [2] Zhang, S., Yu, T., and Feng, Y. "Truthx: Alleviating hallucinations by editing large language models in truthful space". ACL 2024.
> [3] Chuang, Y. S., et al. "Dola: Decoding by contrasting layers improves factuality in large language models". ICLR 2024.
> [4] Chen, Z., et al. "Truth forest: Toward multi-scale truthfulness in large language models through intervention without tuning". AAAI 2024.

---

> ### Author Response · Authors · 2024-11-20
> **Authors' Response (5/5)**
>
> >Q3. If $MC1$ is indeed an accuracy of selecting one out of 4 options, how is the performance of some methods drops consistently and statistically significant below 25% random guess baseline?
>
> You may skip some process of how we construct our privacy-preserving dataset which is introduced in Appendix B.1. We do not use the whole source dataset (DDXPlus/NLICE/SLJA), because the initial datasets are extensive and some questions of initial datasets are too simple and can not be used to reflect the model performance. And then we **exclude these easy questions** and only keep those hard questions in our datasets. Moreover, we further **filter the questions to select those where privacy information plays a crucial role in LLM reasoning**. It is meaningful to benchmark the performance of different privacy-preserving methods. Therefore, the final privacy-preserving dataset is challenging for the LLM, and in some cases, the $MC1$ choice drops below 25% (equivalent to random guessing).
>
> ---
>
> >Q4. In Figure 2, the attack success rate does not seem to grow significantly with increasing privacy budget. Do you have an explanation for that?
>
> As shown in Figure 2, we provide the upper bound of attack success rate of No Protection, which can be regarded as the result when the $\epsilon = \infty$. **So the attack success rate definitely increases with the privacy budget if we include more larger $\epsilon$ values and finally approaching the upper bound**.
> Here we include more $\epsilon$ values of 200, 1500, 2000, 3000 and 5000. And the results are as follows, and we can observe more significant improvement of attack success rate.
>
> Embedding Inverse Attack on DDXplus|$\epsilon$=1|$\epsilon$=200|$\epsilon$=1500|$\epsilon$=2000
> |--|--|--|--|--|
> |$d_\chi$- privacy |0.28|0.39|0.41|0.44 $({\uparrow 0.16})$
> |$d_\chi$- privacy on privacy spans|0.36|0.39|0.46|0.46 $({\uparrow 0.10})$|
> |Paraphrase|0.19|0.31|0.26|0.44 $({\uparrow 0.25})$|
> |PrivacyRestore|0.11|0.17|0.22|0.23 $({\uparrow 0.12})$
>
>
> Attribute Inference Attack on DDXplus|$\epsilon$=1|$\epsilon$=200|$\epsilon$=1500|$\epsilon$=2000
> |--|--|--|--|--|
> |$d_\chi$- privacy |0.14|0.38|0.40|0.41 $({\uparrow 0.27})$
> |$d_\chi$- privacy on privacy spans|0.13|0.13|0.28|0.31 $({\uparrow0.18})$|
> |Paraphrase|0.10|0.11|0.12|0.26 $({\uparrow0.16})$|
> |PrivacyRestore|0.09|0.11|0.15|0.20 $({\uparrow0.11})$|
>
> ---
>
> >Q5. In table 1, I would like to see non-DP baselines. First, a method that does not provide any privacy guarantees. And second, a method that simply removes privacy spans and asks LLM to provide an output based on the incomplete information. That would help put the performance of the methods considered into a proper context.
>
> The No Protection method (non-DP baselines) serves as the **upper bound** of these privacy-preserving methods and No Restoration (simply remove privacy spans) serves as the **lower bound** of these privacy-preserving methods. We have added these experiment results in Q1 of General Response and the results demonstrate that our method shows superior performance compared to No Restoration (lower bound).
>
> ---
> We sincerely thank the reviewer for their efforts in review our paper.
>
> **We are looking forward to your reply. If you have any further questions, we are happy to dicuss them!**

---

> > ### Comment · Reviewer_GKPj · 2024-11-25
> >
> > I thanks the authors for the extensive and detailed response.
> > However, my main concerns remain.
> >
> > >  Therefore, assuming a predefined set will not pose any privacy risk in fact.
> >
> > I claim that in the threat model where the list of privacy spans is known to all parties, having *any* text removed from the query implies the presence of a sensitive information - thus revealing private information. In a world where privacy spans are all serious illness diagnosis, protecting against an adversary knowing which specific one does not fully address privacy risk.
> >
> > > attackers can also attack the server to steal queries
> >
> > Can you clarify which specific attacks or threat models would give an attacker access to the query (post-decryption), but not model weights?
> >
> > > We have demonstrate that the sampling-based method will not leak any privacy information because the sampling generation is protected by the Exponential Mechanism [1].
> >
> > I don't believe naive sampling from a token distribution would provide the DP protections of the exponential mechanism. Can you elaborate how do you guarantee DP here?

---

> > > ### Author Response · Authors · 2024-11-26
> > > **Further Authors' Response (2/2)**
> > >
> > > >I don't believe naive sampling from a token distribution would provide the DP protections of the exponential mechanism. Can you elaborate how do you guarantee DP here?
> > >
> > > Previous works have thoroughly demonstrated in **Appendix A of [1]** and **Section 4.2 of [2]** that sampling from a token distribution provides Differential Privacy protection through the Exponential Mechanism.
> > >
> > > To address your concern and clarify the misunderstanding, we will restate the proof here:
> > > The proof attempt to **demonstrate that the sampling based generation satisfy the definition of Exponential Mechanism as follows**:
> > > 1. Let we define some notation:
> > > $Q$ is the user query, $\mathcal{V}$ is the whole token vocabulary, $u \in \mathbb{R}^{|\mathcal{V}|}$ is the output logit and $u_t$ is the logit for the token $t$ in $\mathcal{V}$.
> > > $\mathcal{M}$ denotes the sampling based generation.
> > > 2. Recall that the logit $u$ should be processed by the softmax layer and then be sampled to obtain the output.
> > > Assume $T$ is the sampling temperature and $Pr[\mathcal{M}(Q) = t]$ indicates the probability of generating the token $t$,
> > > then the softmax layer can be expressed by:
> > > $$
> > >  Pr[\mathcal{M}(Q) = t] = \frac{\exp(u_t / T)}{\sum_{j=1}^{|\mathcal{V}|}\exp(u_j / T)}
> > > $$
> > > 3. Let recall the Exponential Mechanism, assuming $u$ is the utility function and $\Delta u$ is the sensitivity of $u$, then $\mathcal{M}$ satisfy the Exponential Mechanism if and only if
> > > $$
> > > Pr[\mathcal{M}(Q) = t] = \frac{\exp (\epsilon u(Q, t) / 2\Delta u)}{\sum_{j=1}^{|\mathcal{V}|} \exp (\epsilon u(Q, j) / 2\Delta u)}
> > > \propto \exp (\epsilon u(Q, t) / 2\Delta u)
> > > $$
> > >
> > > 4. By Comparing 2 and 3, we can find that the sampling from softmax layer follows the definition of Exponential Mechanism, where $u(Q, t)$ and $u_t$ are different expressions of the same thing.
> > > Furthermore, according to the fact that the privacy budget of Exponential Mechanism is $\epsilon$, we can conclude that the privacy budget of sampling based generation is $2 \Delta u / T$.
> > > The privacy budget decreases with the increasing temperature, indicating that higher temperature will bring better privacy protection.
> > >
> > > We also **provide another proof there, which attempt to directly prove that sampling based generation satisfy the definition of DP mechanism**:
> > >
> > > 1. Let recall the definition of DP mechanism is that for any two queries $Q$ and $Q^{\prime}$, the attack can not infer which one is the initial query given the output $t$, which can be expressed as:
> > > $$
> > > \frac{Pr[\mathcal{M}(Q) = t]}{Pr[\mathcal{M}(Q^{\prime}) = t]} \leq \exp (\epsilon)
> > > $$
> > >
> > > 2. Then for sampling based generation, we consider two neighboring user queries, $Q$ and $Q^{\prime}$, with corresponding output logit $u$, $u^\prime$.
> > > We assume that if the logits $u/u^{\prime} \in \mathbb{R}^{|\mathcal{V}|}$ satisfy the condition $b_1 \leq u_i/u_i^{\prime} \leq b_2, \forall i \in [|\mathcal{V}|]$.
> > > Then we have that:
> > > $$
> > > \frac{Pr[\mathcal{M}(Q) = t]}{Pr[\mathcal{M}(Q^{\prime}) = t]} = \frac{\exp(u_t / T)}{\sum_{j=1}^{|\mathcal{V}|}\exp(u_j / T)} * \frac{\sum_{j=1}^{|\mathcal{V}|}\exp(u_j^{\prime} / T)}{\exp(u_t^{\prime} / T)}
> > >  = \exp(\frac{u_t - u_t^{\prime}}{T}) * \frac{\sum_{j=1}^{|\mathcal{V}|}\exp(u_j^{\prime} / T)}{\sum_{j=1}^{|\mathcal{V}|}\exp(u_j / T)}
> > > \leq \exp (\frac{2(b_2 - b_1)}{T})
> > > $$
> > >
> > > 4. By compare 1 and 2, we can easily find that sampling based generation satisfy the definition of differential privacy.
> > > Furthermore, given the fact the privacy budget of differential privacy is $\epsilon$, then the privacy budget of sampling based generation is $\frac{2(b_2 - b_1)}{T}$.
> > >
> > > [1] Utpala, Saiteja, et al. "Locally Differentially Private Document Generation Using Zero Shot Prompting." EMNLP 2023.
> > > [2] Mattern, J., et al. "The limits of word-level differential privacy". NAACL 2022.
> > >
> > > ---
> > > We sincerely appreciate your efforts.
> > >
> > > **We look forward to your further feedback on our paper. If you have any additional questions, we would be happy to discuss them!**

---

> > > > ### Comment · Reviewer_MqBb · 2024-11-27
> > > >
> > > > I agree that softmax sampling is indeed a form of the Exponential Mechanism, where the sampling temperature is inversely proportional to the privacy budget. Only additional thing to consider is the distances between the inputs to the softmax (logits), which must be bounded to obtain a finite sensitivity.

---

> > > ### Author Response · Authors · 2024-12-01
> > > **Kind Remind (Again)**
> > >
> > > We sincerely appreciate the time and effort you’ve dedicated to reviewing our paper. As the rebuttal period deadline approaches, we would like to kindly remind you that **we are still awaiting your feedback** on our responses.
> > > We hope our responses address all of your concerns and **if we have solved your concerns, can you consider to raise your score?** Your timely feedback would be very helpful and valuable to us.
> > >
> > > Once again, thank you for your thoughtful contributions and valuable advice.

---

> ### Author Response · Authors · 2024-11-26
> **Further Authors' Response (1/2)**
>
> We sincerely appreciate your thoughtful reply and feedback! Below, we have addressed all your concerns:
>
> > I claim that in the threat model where the list of privacy spans is known to all parties, having any text removed from the query implies the presence of a sensitive information - thus revealing private information. In a world where privacy spans are all serious illness diagnosis, protecting against an adversary knowing which specific one does not fully address privacy risk.
>
> Your claim overlooks a critical aspect in our approach.
> It is nearly impossible for the attacker to dertermine whether any text has been removed from the query.
> **Our method is designed to function robustly regardless of whether the input contains any privacy spans.**
> When user query do not contain any privacy spans, the client also constructs the meta restoration vector, which only consist of DP noise and send the meta vector with the query.
> The whole process is the same as the situation when the query contains privacy spans.
> So even if attackers intercept both the transmitted query and the meta restoration vector, **they cannot determine whether the query is the result of privacy spans being removed or if the original query never contained any privacy spans.**
> Therefore, our method effectively prevents attackers from determining whether any privacy spans have been removed from the query, making it impossible for them to infer the presence of sensitive information.
> **The premise of situation (threat model) proposed by you is not realistic at all**, because our method can prevent the attacker knowing whether the client has removed any text from the query and then the attacker also can not infer the presence of sensitive information..
>
> >Can you clarify which specific attacks or threat models would give an attacker access to the query (post-decryption), but not model weights?
>
> The common threat model focuses on attacking the encrypted query rather than the large model weights,
> as **in reality, attacking the query (post-decryption) is much easier than attacking the model weights**.
> This is because the query typically contains **a small number of tokens** compared to the parameter amount of LLM, and various attacks on different encryption algorithms have been thoroughly studied.
> The further explations are as follows:
>
> 1. **Difficulty of stealing Model Weights**:
> First of all, here the considered model is Large Languge Model (LLM) and one of its significant features is the huge amount of parameters (175 billion parameters for GPT-4).
> It is nearly impossible for attacker to steal 175 billion (1,750,000,000,000) parameters from the server.
> On the contrary, **stealing the decrypted query is much easier because even the longest query just has 1024 tokens.**
> **And 1024 is just 0.000000058% of 175 billion.**
> Secondly, in nowadays LLM systems/application (the server), particularly those involving third-party services, **LLM weights are typically kept by the third-party (OpenAI) and is closed-source and well-protected.**
> These LLM systems/application (the server) itself do not own the model weights by themselves and they just call the interface by the third-party (OpenAI).
> For the third party (OpenAI), they usually stored the model weights on secure servers and the weights are often inaccessible even to the developers or publishers of the model themselves.
> As a result, gaining access to these weights is not typically feasible for attackers, particularly in a distributed setting where the model is hosted on cloud services or within proprietary infrastructures.
>
> 2. **Feasibility to attack Encrypted Queries**:
> Indeed, attacks on encrypted queries have been studied extensively over time in the information security field [1,2].
> **While encryption schemes are designed to protect sensitive information, they are not immune to attacks.**
> In reality, the attacker is able to steal the encrypted query and reverse the encryption (for example, through **cryptographic weaknesses or vulnerabilities** in the decryption process),
> Moreover, attackers may exploit reverse engineering methods or use advanced cryptographic attacks, such as **side-channel attacks**, to break the encryption scheme and retrieve the original query.[3,4]
>
> [1] Boneh, D. "Cryptography and Network Security: Principles and Practice". Pearson Education 2008.
> [2] Shamir, A. "How to Share a Secret". Communications of the ACM 1984.
> [3] Max Panoff, et al. A review and comparison of ai-enhanced side channel analysis. JETC 2022.
> [4] Weiss, Roy. “What Was Your Prompt? A Remote Keylogging Attack on AI Assistants” USENIX Security Symposium, 2024.

---

> ### Author Response · Authors · 2024-11-28
> **Kind Remind**
>
> Dear Reviewer,
>
> Thank you again for your feedback. This is a polite reminder that we are awaiting your feedback on our rebuttal. We hope that our recent responses have adequately addressed your concerns. To ensure we have fully resolved the issues, we would appreciate it if you could kindly confirm whether any remaining concerns persist or if further clarification is needed. We would greatly appreciate it if you could discuss them with us at your earliest convenience. To assist you, we have highlighted the key points for your convenience.

---

### Official Review · Reviewer_FuEE · 2024-11-04

**Soundness:** 3
**Presentation:** 3
**Contribution:** 2
**Rating:** 5
**Confidence:** 4

**Summary:**

This paper proposes PrivacyRestore to protect the privacy of user inputs during LLM inference. The user sends text with privacy spans removed along with the aggregated restoration vector to the server. The experimental results demonstrate that PrivacyRestore achieve a good balance between utility and privacy.

**Strengths:**

- The paper proposes a plug-and-play method.
- The authors conduct extensive experiments to validate the effectiveness of their methods.

**Weaknesses:**

- **Identification of privacy spans**. The authors didn't discuss in detail how to identify the privacy span in their paper. As is observed in the dataset they provided, there's still sensitive information contained in the input after the privacy span removed. For example, for the raw query "A 31-year-old male has a history of antipsychotic medication usage, nausea, stimulant drug use. The 31-year-old male presents the symptoms of involuntary eye movement, jaw pain, muscle spasms, ptosis, shortness of breath. What is the likely diagnosis?", the mask input is "A 31-year-old male has a history of nausea. The 31-year-old male presents the symptoms of muscle spasms. What is the likely diagnosis?". We can see that some key information, such as the age, gender, and symptom of the person, is directly reveal to the server. From this perspectively, the protection level of this method is limited, compared with applying DP or SMPC on the full sentence.

- **Reliability of restoration vector**. According to my understanding, the restoration vector for each head is the same for all privacy spans. Then how can the restoration vector help the server obtain better prediction for each specific privacy span? Even if the weights on each head are varied, the usability of the method is not persuasive enough. Furthermore, in the ablation study, the authors are recommended to compare the utility of PrivacyRestore, and the case without restoration vector.

- **Unavoidable privacy leakage from response**. As the framework directly utilizes the response generated by the LLM, it's inevitable that the server could infer the user's information from the plaintext response. Specifically, a high accurate response may contain sensitive attributes of the user, leading to a inherent trade-off between utility and privacy.

**Questions:**

- How do the users determine the privacy span in reality? What if privacy information spans the majority of the prompt?

---

> ### Author Response · Authors · 2024-11-20
> **Authors' Response (1/4)**
>
> We sincerely thank you for your feedback and suggestions! We address all your concerns below:
>
> **To be mentioned, we have updated our submitted paper, please download the latest one, thank you.**
>
> >W1.1. The authors didn't discuss in detail how to identify the privacy span in their paper.
>
> Actually, in our main setting, we assume the users should have the right to determine the privacy spans by themselves.
> And the input of our method is the input with all privacy spans have been identified by users, as shown in the Figure 1.
> In reality, the privacy should be determined by users themselves, which is called "information self-determination right" [1] and this setting is widely advocated by the public [2,3].
> However, our method can still work when the privacy spans in user input can not be identified by users themselves.
> As illustrated in Q3 in General Reponse, our method is orthognal to the existing text sanitization technique [4, 5] and we can use the text sanitization technique to identify all privacy spans before applying our method for restoration. And the experiment results of combining our method and text sanitization, as shown in Q3 in General Reponse, our method can still demostrate superior performance by combining with the text sanitization technique to identified privacy spans. Please refer to Q3 in General Reponse for details explantion and additional experiment results.
>
>
> [1] "Informational self-determination". Wikipedia. https://en.wikipedia.org/wiki/Informational_self-determination.
> [2] Jasper, M. C. "Privacy and the Internet: Your Expectations and Rights Under the Law". Oxford University Press, 2009.
> [3] Van Alsenoy, B., et al. "Privacy notices versus informational self-determination: Minding the gap". International Review of Law, Computers & Technology, 28(2), 185–203, 2014.
> [4] Chen, Y., et al. "Hide and seek (HAS): A lightweight framework for prompt privacy protection".
> [5] Kan, Z., et al. "Protecting user privacy in remote conversational systems: A privacy-preserving framework based on text sanitization".
>
>
>
> >W1.2. As is observed in the dataset they provided, there's still sensitive information contained in the input after the privacy span removed. For example，.... We can see that some key information, such as the age, gender, and symptom of the person, is directly reveal to the server.
>
> Firstly, **the age and gender information of patients in our dataset are all faked**. The original age and gender values are replaced with randomly selected fake values.
> Additionally, the influences of age and gender information on model generation are limited.
> To demonstrate this, we directly remove all age and gender information in user question, rather than replace it by fake one.
> Here is an example, "A 31-year-old male has a history of nausea. The 31-year-old male presents the symptoms of muscle spasms. What is the likely diagnosis?"->"A patient has a history of nausea. The patient presents the symptoms of muscle spasms. What is the likely diagnosis?" The performance of our method on Pri-DDXPlus dataset is as follows:
> | | MC1 | MC2 | RL |  LLM-J|
> |------------|------------|--------------|--------------|--------------|
> | PrivacyRestore(remains fake age/gender)| 62.97 $_{\pm0.00}$|60.19 $_{\pm0.00}$|27.24 $_{\pm0.26}$|4.47 $_{\pm0.00}$|
> | PrivacyRestore(removes all age/gender)| 61.20 $_{\pm0.00}$|58.39 $_{\pm0.00}$|27.63 $_{\pm0.16}$|4.42 $_{\pm0.01}$|
>
> As shown in the table, remaining fake age/gender or removing age/gender will obtain the similar results.
> Secondly, why there are some remaining symptoms is that **all the remaining symptoms in user questions are non-sensitive symptoms and will not cause privacy risks**.
> Indeed, for realistic situations, some symptoms, such as pain, are less likely to be linked to personal privacy, while others, like a history of HIV, are highly sensitive.
> We classify all symptoms into sensitive and non-sensitive categories using powerful ChatGPT, as shown in Appendix B.1 and lines 356–357 of the main text. Applying protection to non-sensitive symptoms offers minimal privacy improvement (nealy no improvement), but incurs additional performance costs. This is why our method outperforms the application of DP or SMPC—it allows for a more reasonable allocation of privacy costs.

---

> ### Author Response · Authors · 2024-11-20
> **Authors' Response (2/4)**
>
> >W2. The restoration vector for each head is the same for all privacy spans. Then how can the restoration vector help the server obtain better prediction for each specific privacy span? ... Furthermore, in the ablation study, the authors are recommended to compare the utility of PrivacyRestore, and the case without restoration vector.
>
> You have some misunderstandings about our methods. As shown in our definition in Line 256-257, we train different restoration vectors $r^h_s$ for different privacy spans and different heads. So **different heads $h$ and different privacy spans $s
> $will cause different restoration vectors**. Then obviously, the restoration vector can help prediction, even lacking specific privacy spans in the inputs. Also, the weights are used to reflect the importance of different privacy spans, as shown Line 287-288, which is varied for different privacy spans $s$ rather than heads $h$.
> Furthermore, we add the comparions between PrivacyRestore and the case without restoration vectors in Q1 in General Reponse. The experiment results can also demonstrate the effectiveness of our method.
>
>
> >W3. Unavoidable privacy leakage from response. As the framework directly utilizes the response generated by the LLM, it's inevitable that the server could infer the user's information from the plaintext response. Specifically, a high accurate response may contain sensitive attributes of the user, leading to a inherent trade-off between utility and privacy.
>
> Actually, as demonstrated by prior works [1, 2], if we use sampling based LLM generation rather than greedy searching based LLM generation (directly select the highest probability token), the outputs from LLM is protected by the **Exponential Mechanism** [3], which is a variant of Differential Privacy. In all experiments of our method, **we use sampling based generation and set the generation temperature at 1.0**. The higher temperature means better privacy protection and the commonly used ranges of temperature is [0.75, 1.75], as shown in the Implementation Details part of [2]. For empirical proof,  we provide more experimental results by implement Attribute Inference Attack and Embedding Inverse Attack to attack the generated outputs of our method to recover the privacy spans in user input. We also count the frequency of occurrence of privacy spans of user inputs in model generated outputs. The details experiment results and explations please refer to Q4 in General Reponse.
>
> [1] Mattern, J., et al. "The limits of word-level differential privacy". NAACL 2022.
> [2] Utpala, S., et al. "Locally differentially private document generation using zero-shot prompting". EMNLP 2023.
> [3] McSherry, F., et al. "Mechanism design via differential privacy". FOCS 2007.

---

> ### Author Response · Authors · 2024-11-20
> **Authors' Response (3/4)**
>
> >Q1. How do the users determine the privacy span in reality?
>
> In reality, privacy span is personalized and contextual. Users should have the ability to personally select which spans are treated as privacy. [1,2,3] So in our main setting, the users should determine which parts of the question are considered private by themselves. And then they should select the closest privacy spans from the predefined privacy spans set for constructing the restoration vector and remove the private part from the question. Finally, the restoration vector and sanitized question are sent to the server. Two potential problems may exist that you might raise.
> **1. What if the users can not find the exact privacy spans in the predefined privacy spans set?**
> In fact, most privacy spans are concentrated in a small set, while others outside this set occur less frequently. As shown in Q2 in General Reponse, we counted the occurrence frequency of different privacy spans in user questions of Pri-DDXPlus dataset and found that the distribution is long-tailed. This means that if the predefined privacy spans set includes the high-frequency spans, most potential privacy spans in user questions will be covered.
> Even if certain privacy spans in a user’s query fall outside the predefined set, users can still replace them with the most similar spans from the predefined set. We have added additional experiments under the setting that only Top100 privacy spans are included in the predefined set and the other 49 privacy spans are out of the predefined set. The experiment results in Q2 in General Reponse demonstrate that our method still work well, **with the improvement of 21.36 in MC1, 19.66 in MC2, 1.18 in ROUGE-L and 0.98 in LLM-J**, compared to the No Restoraion baseline. And these results are comparable to the results when the predefined privacy spans cover all potiential privacy spans.
> **2. What if the users can not determine the privacy spans of their question by themselves and how to find the privacy spans in the questions?**
> It is worth emphasizing that the main contribution of our work is to restore the lacking privacy spans during model inference and ensure the privacy security. How to identify and remove the privacy spans automatically  has been well-studied, which is call "text sanitization" [1, 2]. Our method is orthogonal to these text sanitization methods and can be easily combined with these methods to handle this situation when the users can not dertermine privacy spans of their questions by themselve. To demonstrate the effectiveness, we combine our method with prior text sanitization method [1]. The detailed implementations refer to Q3 in General Reponse.
> As shown in Q3 in General Reponse, combining with the existing text sanitization method, our method can effectively identify and remove the privacy spans in user questions while maintaining the inference utility. Moreover, the inference performance of our method when the privacy spans is unknown (users can not determine privacy spans) is comparable to the performance when the privacy spans is known (user can determine the privacy spans by themselves).
> For more detailed information about “how to identify privacy spans in reality", please refer to Q3 in General Reponse.
>
> [1] "Informational self-determination". Wikipedia. https://en.wikipedia.org/wiki/Informational_self-determination.
> [2] Jasper, M. C. "Privacy and the Internet: Your Expectations and Rights Under the Law". Oxford University Press, 2009.
> [3] Van Alsenoy, B., et al. "Privacy notices versus informational self-determination: Minding the gap". International Review of Law, Computers & Technology, 28(2), 185–203, 2014.
> [4] Chen, Y., et al. "Hide and seek (HAS): A lightweight framework for prompt privacy protection".
> [5] Kan, Z., et al. "Protecting user privacy in remote conversational systems: A privacy-preserving framework based on text sanitization".

---

> ### Author Response · Authors · 2024-11-20
> **Authors' Response (4/4)**
>
> >Q2. What if privacy information spans the majority of the prompt?
>
> If the privacy information spans the majority of the prompt, all privacy information will be removed and the model inference is mainly influenced by the restoration vector. The restoration vector can effectively improve the utility of the generated output. Here we provide an example where the count of initial symptoms is 10 and the count of private ones is 8 (**80% of symptoms are private and occupy the majority of the prompt**):
> ```
> [User Question]:
> I'm a 68-year-old woman with a history of taking antipsychotic medications and using stimulant drugs. Lately, I've been experiencing some troubling symptoms. My eyes are moving involuntarily, and I'm having pain in my jaw. I've also been having muscle spasms in my neck, and my eyelids are drooping. On top of that, I've been feeling short of breath. Given my medical history and these symptoms, what could be causing this?
>
> [Privacy Spans]:
> ptosis, involuntary eye movement, tongue protrusion, muscle spasms in neck, shortness of breath, jaw pain, stimulant drug use, antipsychotic medication usage
>
> [Sanitized Question]:
> A patient presents the symptoms of nausea, muscle spasms. What is the likely diagnosis?
>
> [Response under the initial question]:
> Based on the symptoms you've provided, the likely diagnosis for this 80-year-old woman is a drug-induced movement disorder, specifically tardive dyskinesia (TD).
> Tardive dyskinesia is a serious side effect of long-term use of antipsychotic medications, such as haloperidol, risperidone, and olanzapine. It is characterized by involuntary movements of the face, tongue, and limbs, as well as jaw pain, muscle spasms, and neck stiffness. The symptoms can be mild, moderate, or severe and can persist even after the medication is discontinued.
> The patient's history of antipsychotic medication usage and stimulant drug use increase the likelihood of TD. The patient's age also plays a role, as TD is more common in older adults due to the accumulation of toxic metabolites in the brain.
>
> [Response under sanitized question without PrivacyRestore]:
> (ROUGE-L: 21.92; LLM-J: 2)
> Thank you for presenting this case scenario. Based on the symptoms provided, the likely diagnosis for the 80-year-old female patient is a metabolic disorder, specifically hypokalemia (low potassium levels in the blood).
> Hypokalemia can cause a range of symptoms, including nausea, muscle spasms, and fatigue. As the patient is 80 years old, it is important to consider the possibility of a chronic condition such as hypokalemia, which can be caused by a variety of factors including kidney disease, diuretic use, and certain medications.
>
> [Response under sanitized question using PrivacyRestore]:
> (ROUGE-L: 42.71; LLM-J: 8)
> Thank you for presenting this case scenario. Based on the symptoms provided, the likely diagnosis for the 80-year-old female patient is a drug-induced movement disorder, specifically tardive dyskinesia (TD).
> TD is a serious side effect of long-term use of certain medications, particularly antipsychotics, antidepressants, and benzodiazepines. The symptoms of TD include involuntary movements of the face, tongue, and limbs, as well as grimacing, protrusion of the tongue, and repetitive movements of the arms and legs.
> The patient's age and the presence of muscle spasms also suggest the possibility of another movement disorder, such as dystonia or chorea. However, given the patient's history of long-term medication use, TD is the most likely diagnosis.
> ```
>
> ---
> We deeply appreciate the reviewer’s efforts.
>
> **We are looking forward to your reply. If you have any further questions, we are happy to dicuss them!**

---

> ### Comment · Reviewer_FuEE · 2024-11-25
>
> I appreciate the authors' effort during the rebuttal. However, I still have some concerns:
>
> W1.1. Self-determination is a good idea. However, various definition may cause different length and proportion of privacy span in the sentence, I think it's important to discuss how that affect the utility.
>
> W2. That requires to pre-define and mantain an exhaustive list of sensitive privacy spans. Though the privacy spans follow a long-tailed distribution, it is very likely that new ones could appear continuously. Maintaining a huge table of privacy spans and their corresponding restoration vector could be a burden for both the server and user.

---

> > ### Author Response · Authors · 2024-11-28
> > **Kind Remind**
> >
> > Dear Reviewer,
> >
> > Thank you again for your feedback. This is a polite reminder that we are awaiting your feedback on our rebuttal. We hope that our recent responses have adequately addressed your concerns. To ensure we have fully resolved the issues, we would appreciate it if you could kindly confirm whether any remaining concerns persist or if further clarification is needed. We would greatly appreciate it if you could discuss them with us at your earliest convenience. To assist you, we have highlighted the key points for your convenience.

---

> ### Author Response · Authors · 2024-11-25
> **Further Authors' Response (1/1)**
>
> We sincerely appreciate your thoughtful reply and feedback! Below, we have addressed your concerns:
>
> > W1.1. Self-determination is a good idea. However, various definition may cause different length and proportion of privacy span in the sentence, I think it's important to discuss how that affect the utility.
>
> Thank you for agreeing with our self-determination setting.
> **In fact, longer lengths and a larger proportion of privacy spans in the sentence do not impact the utility of our method**, as demonstrated in the example in the 'Authors' Response (4/4)' section.
> In this example, 80% of the user's question consists of privacy spans, yet the response generated by our method maintains significantly higher quality compared to when our method is not used.
> The ROUGE-L rises from 21.92 to **42.71** and LLM-J rises from 2 to **8**.
>
> We also give more evidence to prove that our method is robust to the length and proportion of privacy spans.
> We divide the test set of Pri-DDXPlus according to the proportion of privacy spans and evaluate our methods under different proportion setting.
> We consider a proportion of privacy spans greater than 0.5 as indicating that privacy spans dominate the question.
> The experiment results are as follows:
>
> | | MC1 | MC2 | RL |  LLM-J|
> |------------|------------|--------------|--------------|--------------|
> |Proportion<0.5(without Privacy Restore) | 58.32 |55.15 |26.22 |3.42 |
> |Proportion<0.5(with Privacy Restore) |79.97 $(\uparrow 21.65)$ |74.38 $(\uparrow 19.23)$ |28.86 $(\uparrow 2.64)$ |4.7 $(\uparrow 1.28)$ |
> |Proportion>0.5(without Privacy Restore) | 10.98 |11.81 |24.25 | 3.02 |
> |Proportion>0.5(with Privacy Restore) | 47.53 $(\uparrow 36.55)$ |47.14 $(\uparrow 35.33)$ |25.76 $(\uparrow 1.51)$ |4.26 $(\uparrow 1.24)$ |
>
> As the table shows,
> **although the performance of our method decreases when privacy spans dominate the question (proportion > 0.5) compared to when the proportion is less than 0.5,
> our method still outperforms the baseline (No restoration)**, demonstrating its effectiveness.
>
> > W2. That requires to pre-define and mantain an exhaustive list of sensitive privacy spans. Though the privacy spans follow a long-tailed distribution, it is very likely that new ones could appear continuously. Maintaining a huge table of privacy spans and their corresponding restoration vector could be a burden for both the server and user.
>
> Firstly, **we have added additional experiments and analysis to examine the impact of new privacy spans emerging in General Q2 of the General Response.**
> If new privacy spans continue to emerge, our predefined set of privacy spans may not cover all of them.
> However, our method remains effective due to the long-tailed distribution characteristic.
> As demonstrated in General Q2 of the General Response, we conducted experiments where our predefined privacy span set only covered **the top 5, 40, 80, 100, or 120 spans**.
> **Despite this limitation, our method still outperforms the 'No restoration' baseline (lower bound)**, showcasing its robustness.
>
> Secondly, **the cost of storing the predefined privacy spans and restoration vectors is very low because the restoration vectors are stored only for the common Top-K heads set $\mathcal{H}_c$ , which consists of just 175 or 125 heads, rather than all the heads in the LLM.**
> The initial Llama-2-7B model has 1024 heads, along with additional embedding and FFN parameters, totaling 7 billion parameters.
> In contrast, our method uses only 175 or 125 heads, with the dimension of each head is 128 and corresponding parameter counts are **175 $\times$ 128 or 125 $\times$ 128 (22.4K or 16K parameters)**.
> **This amounts to just 0.0003%/0.0002% of the total parameters in the original LLM.**
> Therefore, the storage burden on both the server and user is negligible, and in fact, this low burden is one of the key advantages of our approach.
>
> ---
> We sincerely appreciate your efforts.
>
> **We look forward to your further feedback on our paper. If you have any additional questions, we would be happy to discuss them!**

---

> ### Author Response · Authors · 2024-12-01
> **Kind Remind (Again)**
>
> We sincerely appreciate the time and effort you’ve dedicated to reviewing our paper. As the rebuttal period deadline approaches, we would like to kindly remind you that **we are still awaiting your feedback** on our responses.
> We hope our responses address all of your concerns and **if we have solved your concerns, can you consider to raise your score?** Your timely feedback would be very helpful and valuable to us.
>
> Once again, thank you for your thoughtful contributions and valuable advice.

---

### Author Response · Authors · 2024-11-20
**Part Ⅰ of General Response**

We sincerely appreciate the time and effort of all reviewers in evaluating our paper. Below, we address the common questions raised by the reviewers:
### **General Q1. More results about No Protection (performance upper bound)and No Restoration (performance lower bound)**
We have added more experiment results of No Protection (directly sending the initial query to the server) and No Restoration (directly sending the incomplete query with privacy spans removed). No Protection method provides the performance upper bound but the complete query will be attacked very easily. No Restoration method serves as the lower bound, as incomplete queries will inevitably degrade the performance of LLM inference.
**The results show that our method can outperform the lower bound across all three datasets, effectively demonstrating its effectiveness**.
(It is worth mentioning that the RL of No Protection is 100.00 because the reference of RL computation is just the output generated without any protection method.)


Pri-DDXPlus|MC1|MC2|RL|LLM-J
|--|--|--|--|--|
PrivacyRestore|62.97 $_{\pm0.00}(\uparrow 29.4)$|60.19 $_{\pm0.00}(\uparrow 27.7)$|27.24 $_{\pm0.26}(\uparrow 2.05)$| 4.47 $_{\pm0.00}(\uparrow 1.26)$
No protection(upper bound)|64.88 $_{\pm0.01}$|61.48 $_{\pm0.03}$| 100.00 $_{\pm0.00}$| 5.58 $_{\pm0.03}$
No restoration(lower bound)|33.57 $_{\pm0.00}$ |32.49 $_{\pm0.01}$ |25.19 $_{\pm0.43}$ |3.21 $_{\pm0.01}$

Pri-NLICE|MC1|MC2|RL|LLM-J
|--|--|--|--|--|
PrivacyRestore|62.23 $_{\pm1.70}(\uparrow 35.16)$ |57.94 $_{\pm0.09}(\uparrow 29.31)$ |24.42 $_{\pm0.81}(\uparrow 7.52)$ |3.67 $_{\pm0.01}(\uparrow 2.06)$
No protection(upper bound)|80.30 $_{\pm0.38}$ |77.60 $_{\pm1.23}$ |100.00 $_{\pm0.00}$ | 5.90 $_{\pm0.04}$
No restoration(lower bound)|27.07 $_{\pm1.98}$ |28.63 $_{\pm2.23}$ |16.90 $_{\pm0.51}$ |1.61 $_{\pm0.03}$

Pri-SLJA|MC1|MC2|RL|LLM-J
|--|--|--|--|--|
PrivacyRestore|35.47 $_{\pm1.48}(\uparrow 10.55)$ |35.41 $_{\pm0.64}(\uparrow 9.44)$ | 37.56 $_{\pm0.06}(\uparrow 6.54)$ | 5.25 $_{\pm0.00}(\uparrow 0.82)$
No protection(upper bound)| 69.57 $_{\pm0.61}$ | 67.58 $_{\pm0.43}$ | 100.00 $_{\pm0.00}$ | 5.44 $_{\pm0.03}$
No restoration(lower bound)|24.92 $_{\pm0.98}$ | 25.97 $_{\pm1.12}$ | 31.02 $_{\pm0.16}$ | 4.43 $_{\pm0.01}$

---

> ### Author Response · Authors · 2024-11-20
> **Part Ⅱ of General Response**
>
> ### **General Q2. What if new privacy spans (symptoms/diseases) appear?**
>
> Firstly, the privacy spans proposed by us can be regarded as some attribute of users (e.g. symptoms/disease). These attributes are particularly relevant to privacy concerns in specific contexts, such as medical diagnosis and legal judgment. In reality, the number of potential types of privacy spans (e.g. symptoms/diseases) is **finite and follows a long-tailed distribution**. For example,the World Health Organization (WHO) has proposed the classification of all diseases and symptoms in International Classification of Diseases (ICD) in [1] and the famous Mayo Clinic also list all existing symptoms in [2]. So, it is possible to include these finite symptoms/diseases in our predefined privacy spans.
> Another characteristic of the privacy spans is that most privacy spans focus on several majority categories, which indicates that the distribution is long-tailed.
> We recorded the occurrence frequencies of the top n privacy spans, where "top n" refers to the n most frequently occurring privacy spans.
> **We observed that certain types of privacy spans (majority types) dominate the occurrence of privacy spans**, as shown in the following tables:
>
> Total privacy spans type of Pri-DDXPlus | The occurrence ratio of Top 5 types |The occurrence ratio of Top 40 types |The occurrence ratio of Top 80 types |The occurrence ratio of Top 100 types |The occurrence ratio of Top 120 types
> |-|-|-|-|-|-|
> 149 |19.52\% |50.02\% |76.35\%| **84.11\%** |90.36\%
>
> As shown in the table above, the top 100 privacy span types cover nearly all possible occurrences of privacy spans.
> Therefore, if our predefined set of privacy spans includes these major categories, it will be sufficient to handle most user queries.
> Even if the privacy spans in user queries fall outside our predefined set, simply replacing them with the most similar spans from the predefined set can still be an effective approach.
> And our method continues to demonstrate superior performance, as shown below.
> **Even when only the top 100 types are included, our method still significantly outperforms the No Restoration approach**, with the improvement marked in parentheses.
>
>
> Methods(Pri-DDXPlus)|MC1|MC2|RL|LLM-J|
> |-- |--|--|--|--|
> No restoration(lower bound)|33.57| 32.49| 25.19| 3.21
> predefine only top 5 |38.21 |36.17| 25.82 |3.57
> Predefine only top 40 |44.28 |42.00 |25.89| 3.83
> Predefine only top 80 |45.83| 43.53|26.59 |3.95
> Predefine only top 100| 54.93 $(\uparrow 21.36)$| 52.15 $(\uparrow 19.66)$ |26.37 $(\uparrow 1.18)$| 4.19 $(\uparrow 0.98)$
> Predefine only top120 |58.42 | 55.40 |26.87 |4.27|
> Predefine ALL(top149) |62.97 $(\uparrow 29.4)$| 60.19 $(\uparrow 27.70)$| 27.24 $(\uparrow 2.05)$| 4.47 $(\uparrow 1.26)$
>
>
>
> [1] WHO. "International Classification of Diseases 11th Revision." https://icd.who.int/browse/2024-01/mms/en.
> [2] Mayo Clinic. "Symptoms A-Z." https://www.mayoclinic.org/symptoms/index?letter=C.

---

> ### Author Response · Authors · 2024-11-20
> **Part Ⅲ of General Response**
>
> ### **General Q3. How to identify privacy spans in reality?**
>
> First of all, in reality, privacy spans are personalized because someone regards the particular spans as private while others do not. Therefore, we claimed to give users the right to decide what privacy is by themselves.
> In reality, **the privacy should be determined by users themselves**, which is called "information self-determination right" [1] and this setting is widely advocated by the public [2,3].
> So in our main experiment, we assume that **users are responsible for identifying the parts of the question they consider private and matching them to predefined privacy spans** to construct restoration vectors.
> They can then remove the private portions of the question directly to obtain the santinized question. The sanitized question can even be formatted into a fixed structure, such as: 'A patient has a history of xxx. The patient presents with symptoms of xxx. What is the likely diagnosis?'
> This is because in reality users can and should handle sanitization and formatting by themselves.
> However, some reviewers expressed concerns that **users might struggle with sanitization and formatting on their own**.
> Under this setting, users' questions may vary in format, and it may be unclear which parts are considered private.
> However, our methods can still work for this situation by just easily combining our work with previous well-studied "text sanitization" work [4,5].
> Text sanitization focuses on identifying privacy and removing privacy automatically.
> **Our work is completely orthogonal to existing text sanitization techniques and can be easily combined with them** to handle situations where users cannot determine privacy.
> We have implemented this by using the text sanitization technique to identify privacy spans and remove these privacy spans.
> And then, based on the identified privacy spans, restoration vectors are constructed to restore LLM inference utility, after removing the privacy spans. The implementation details are as follows:
>
> **Step1**: Utilizing GPT-4 to generate potential user questions, varying in format and description. Here is an example:
> ```
> [Privacy Spans]:
> involuntary eye movement, jaw pain, muscle spasms in neck, ptosis, shortness of breath, antipsychotic medication usage, stimulant drug use
> [Question before transforming]:
> A 10-year-old woman has a history of antipsychotic medication usage, nausea, stimulant drug use. The 10-year-old woman presents the symptoms of involuntary eye movement, jaw pain, muscle spasms, muscle spasms in neck, ptosis, shortness of breath. What is the likely diagnosis?
> [Question after transforming (Varying in format)]:
> I'm a 10-year-old girl and I've been taking antipsychotic medication for a while now. I've been feeling really nauseous, and I also used a stimulant drug recently. Lately, I've been having these strange symptoms like my eyes moving involuntarily, and my jaw hurts a lot. I've also been getting muscle spasms in my neck and elsewhere, and my eyelids droop sometimes. On top of that, I've been feeling short of breath. What could be causing all of these symptoms?
> ```
>
> **Step2**: Following the text sanitization work [4], we finetuned a best-base-uncased model to identify which privacy spans appear in the user questions, and the performance of identification is :
>
> | |Precision|Recall| F1|
> |--|--|--|--|
> Privacy Spans Identification|99.16 $_{\pm 0.21}$|98.78 $_{\pm 0.31}$|99.66 $_{\pm 0.27}$
>
> **Step3**: Based on the identified results, we finetuned a Qwen-2.5-0.5B model to paraphrase the user question to achieve the text sanitization (remove all privacy spans).
> We also use Attribute Inference Attack and Embedding Inverse attack to attack the sanitized question to evaluate the performance of privacy protection, which is shown as follows:
>
> ||Embedding Inverse Attack(ROUGE-L )|Attribute Inference Attack (F1)
> |-|-|-
> No ProtectionResults|0.40|0.70
> Sanitized Results|0.06 $(\downarrow 0.34)$|0.07 $(\downarrow 0.63)$

---

> ### Author Response · Authors · 2024-11-20
> **Part Ⅳ of General Response**
>
> **Step4**: The following is our method, which mainly consists of constructing the meta restoration vector and restoring information during LLM inference. The utility performance results are as follows:
>
>
> Pri-DDXPlus|MC1|MC2|RL|LLM-J
> |-|-|-|-|-|
> PrivacyRestore(Unknown privacy spans)|62.87 $_{\pm 0.02}$|59.97 $_{\pm 0.01}$|26.47 $_{\pm 0.31}$|4.28 $_{\pm 0.01}$
> PrivacyRestore(Known privacy spans)|62.97 $_{\pm 0.00}$|60.19 $_{\pm 0.00}$|27.24 $_{\pm 0.26}$|4.47 $_{\pm 0.00}$
> No protection(upper bound)|64.88 $_{\pm0.01}$|60.48 $_{\pm 0.03}$|100.00 $_{\pm 0.00}$|5.58 $_{\pm 0.03}$
> No restoration(lower bound)|33.57 $_{\pm 0.00}$|32.49 $_{\pm 0.01}$|25.19 $_{\pm 0.43}$|3.21 $_{\pm 0.01}$
>
> As the above table shows, **even the users can not identify privacy spans (unkown privacy spans), our method can be combined with the text sanitization technique and achieve superior performance improvement compared to No Restoraion (lower bound)**.
> The performance results are even comparable to our initial setting where the user can identify privacy by themselves.
>
> [1] "Informational self-determination". Wikipedia. https://en.wikipedia.org/wiki/Informational_self-determination.
> [2] Jasper, M. C. "Privacy and the Internet: Your Expectations and Rights Under the Law". Oxford University Press, 2009.
> [3] Van Alsenoy, B., et al. "Privacy notices versus informational self-determination: Minding the gap". International Review of Law, Computers & Technology, 2014.
> [4] Chen, Y., et al. "Hide and seek (HAS): A lightweight framework for prompt privacy protection".
> [5] Kan, Z., et al. "Protecting user privacy in remote conversational systems: A privacy-preserving framework based on text sanitization".
>
> ### **General Q4. Do the outputs from LLM leak privacy spans in our method?**
> First of all, the answer to the question is No because our method generates outputs by sampling rather than gready searching.
> The detailed explanations/proofs are as follows:
> As pointed by prior work [1], [2], the reason why paraphrasing can protect user privacy is that the last step in generation of LLM is sampling from token probability distribution rather than greedy searching (directly selecting the highest probability token).
> The sampling mechanism follows the definition of Exponential Mechanism [3], which is a variant of Differential Privacy.
> Therefore, even attackers can obtain the generated outputs from LLM (paraphrased text), they can not infer any privacy containing in user inputs.
> The detailed proof of why the sampling mechanism satisfies the exponential mechanism can be found in [2]. Due to space limitations, we do not repeat it here.
> **In all of experiments of our method, the outputs from LLM are also samping from token probability distribution and the temperature are set at 1.0.**
> So the outputs from LLM using our method will not leak privacy spans.
> Our method can protect privacy protection for both user inputs and user outputs, although the protection of output through sampling is not our main contribution.
> All of the above are theoretical proof of why the outputs from LLM will not leak privacy spans in our method.
> We also conduct experiments to empirically demonstrate that our method can protect privacy-sensitive spans, even when the attacker has access to the outputs.
> Here we implement Attribute Inference Attack and Embedding Inverse Attack to attack the generated outputs, and we also report the occurrence of privacy spans in model generated outputs under different generation temperatures.
> The results demonstrate that when output is generated through sampling, no privacy information is leaked.
>
> Temperature(Pri-DDXPlus)|0.75|1.0|1.25|1.5|1.75
> |-|-|-|-|-|-|
> Attribute Inference Attack (F1)|0.0963|0.0970|0.0922|0.0922|0.0970
> Embedding Inverse Attack(ROUGE-L)|0.0367|0.0375|0.0353|0.0353|0.0373
> Occurrence of privacy spans in generated outpouts|0.0306|0.0299|0.0301|0.0293|0.0307
>
> [1] Mattern, J., et al. "The Limits of Word Level Differential Privacy". Findings of the Association for Computational Linguistics: NAACL 2022.
> [2] Utpala, S., et al. "Locally Differentially Private Document Generation Using Zero Shot Prompting". Findings of the Association for Computational Linguistics: EMNLP 2023.
> [3] McSherry, F., and Talwar, K. "Mechanism Design via Differential Privacy". FOCS 2007.
>
> ---
>
> We sincerely thank all reviewers again for their valuable suggestions, which have greatly helped strengthen our paper.
>
> **If you have any further questions, we would be happy to discuss them!**

---

### Note · Authors · 2024-12-15

I have read and agree with the venue's withdrawal policy on behalf of myself and my co-authors.